# Cryo-EM structures reveal native GABA$_A$ receptor assemblies and pharmacology

Chang Sun[1], Hongtao Zhu[1,3], Sarah Clark[1,4] & Eric Gouaux[1,2 ✉]

Type A γ-aminobutyric acid receptors (GABA$_A$Rs) are the principal inhibitory receptors in the brain and the target of a wide range of clinical agents, including anaesthetics, sedatives, hypnotics and antidepressants[1–3]. However, our understanding of GABA$_A$R pharmacology has been hindered by the vast number of pentameric assemblies that can be derived from 19 different subunits[4] and the lack of structural knowledge of clinically relevant receptors. Here, we isolate native murine GABA$_A$R assemblies containing the widely expressed α1 subunit and elucidate their structures in complex with drugs used to treat insomnia (zolpidem (ZOL) and flurazepam) and postpartum depression (the neurosteroid allopregnanolone (APG)). Using cryo-electron microscopy (cryo-EM) analysis and single-molecule photobleaching experiments, we uncover three major structural populations in the brain: the canonical α1β2γ2 receptor containing two α1 subunits, and two assemblies containing one α1 and either an α2 or α3 subunit, in which the single α1-containing receptors feature a more compact arrangement between the transmembrane and extracellular domains. Interestingly, APG is bound at the transmembrane α/β subunit interface, even when not added to the sample, revealing an important role for endogenous neurosteroids in modulating native GABA$_A$Rs. Together with structurally engaged lipids, neurosteroids produce global conformational changes throughout the receptor that modify the ion channel pore and the binding sites for GABA and insomnia medications. Our data reveal the major α1-containing GABA$_A$R assemblies, bound with endogenous neurosteroid, thus defining a structural landscape from which subtype-specific drugs can be developed.

Regulation of brain excitability by activation of neuronal GABA$_A$Rs is essential for normal brain development and function[5]. Deficits in GABA$_A$R activity are associated with health problems, ranging from epilepsy to intellectual disability[6]. A large number of ions and small molecules modulate GABA$_A$R activity, including $Zn^{2+}$ (ref. 7) and picrotoxin[8], as well as therapeutic agents, such as benzodiazepines[8–10], barbiturates[10] and propofol[10]. Indeed, the GABA$_A$R modulators flurazepam and ZOL are widely used to treat insomnia. Lipophilic neurosteroids are also potent endogenous modulators of GABA$_A$Rs. APG (3α-hydroxy-5α-pregnan-20-one), synthesized chiefly in the brain[11], potentiates GABA$_A$R activity in a subunit-dependent manner[12–14], and its anxiolytic and sedative effects have proved to be effective for the treatment of postpartum depression[15]. In addition, ganaxolone, a synthetic derivative of APG, has recently entered the clinic as an anticonvulsive agent.

Because modulation of receptor function is dependent on subunit composition and arrangement, a knowledge of native GABA$_A$R architecture is crucial to understand how these different molecules elicit distinct physiological responses. However, the potential diversity of pentameric GABA$_A$Rs is vast due to the existence of 19 different receptor subunits (α1–6, β1–3, γ1–3, ρ1–3, δ, ε, π and θ). Studies in vitro suggest that variations in subunit expression levels can modify subunit stoichiometry. Despite progress in resolving the architecture of recombinant di- and tri-heteromeric GABA$_A$Rs[4,7,9,16,17], there is no structural understanding of the various GABA$_A$Rs that are present in the brain. Moreover, although decades of biochemical and biophysical studies have thoroughly investigated the subunit identities of GABA$_A$R complexes, including major and minor assemblies, the arrangements of subunits in their respective complexes remain less well explored[2,3].

To elucidate the ensemble of GABA$_A$Rs that define the molecular action of endogenous and therapeutic modulators, we isolated native α1 subunit-containing GABA$_A$Rs (nα1-GABA$_A$Rs) from mouse brains using an engineered high-affinity, subunit-specific antigen-binding antibody fragment (Fab)[9]. Because the α1 subunit is ubiquitously expressed throughout the brain, and is a subunit of both synaptic and extrasynaptic receptors, this approach enabled us to analyse 60%–80% of native GABA$_A$Rs[18,19]. Furthermore, it permitted an investigation of three clinically relevant molecules, APG, didesethylflurazepam (DID) and ZOL, in the context of native receptors. With isolated nα1-GABA$_A$Rs in hand, we were able to count the number of α1 subunits in these

[1]Vollum Institute, Oregon Health and Science University, Portland, OR, USA. [2]Howard Hughes Medical Institute, Oregon Health and Science University, Portland, OR, USA. [3]Present address: Laboratory of Soft Matter Physics, Institute of Physics, Chinese Academy of Sciences, Beijing, China. [4]Present address: Department of Biochemistry and Biophysics, Oregon State University, Corvallis, OR, USA. ✉e-mail: gouauxe@ohsu.edu

complexes by single-molecule fluorescence bleaching experiments, investigate protein composition by mass spectrometry and elucidate high-resolution structures of nα1-GABA$_A$R assemblies by single-particle cryo-EM.

## Isolation of functional nα1-GABA$_A$Rs

We engineered the 8E3 α1 subunit-specific Fab fragment[9] to include a green fluorescent protein (GFP) fluorophore, affinity tag and 3C protease site to enable release from the affinity resin (8E3-GFP Fab; dissociation constant, $K_d$ = 0.5 nM; Extended Data Fig. 1). The 8E3-GFP Fab was then used to isolate nα1-GABA$_A$Rs from solubilized mouse brain tissue (excluding the cerebellum) while monitoring the purification workflow via GFP fluorescence. Detergent (lauryl maltose neopentyl glycol (LMNG)) treatment routinely solubilized the majority of nα1-GABA$_A$Rs, accompanied by the inhibitory synapse marker neuroligin 2 (Extended Data Fig. 1). Following nearly complete capture of receptors on affinity resin (Extended Data Fig. 1), nα1-GABA$_A$Rs complexes were reconstituted into lipid-filled nanodiscs and eluted by 3C protease treatment. Further purification by size-exclusion chromatography yielded an ensemble of nα1-GABA$_A$R–Fab complexes. Radioligand binding assays showed that the purified pentameric preparations were functional and retained high-affinity flunitrazepam binding ($K_d$ = 6.0 ± 0.2 nM (mean ± s.e.m.); Extended Data Fig. 1).

To validate that the 8E3-GFP Fab captured GABA$_A$R complexes, before carrying out structural studies, we performed mass spectrometry analysis of the purified native receptor complexes, cognisant of the experimental fact that mass spectrometry is a highly sensitive method that enables the identification of both the most abundant receptor subunits as well as those subunits that consist only of a small fraction of the total population. Indeed, in the subsequent mass spectrometric analysis we identified all α, β and γ subunits, as well as the δ subunits (Extended Data Fig. 1 and Supplementary Tables 1–3), demonstrating that α1-dependent isolation captured receptors containing most of the 19 GABA$_A$R subunits, which is consistent with decades of incisive experimental studies showing that the α1 subunit assembles with all other α subunits, and with the β, γ and δ subunits[19–24]. Our mass spectrometry studies also detected peptides unique to the proposed short-splicing isoform of the γ2 subunit[25] (γ2-S) (Supplementary Table 3).

## Three major populations of nα1-GABA$_A$Rs

To elucidate the composition and arrangement of native receptors, we collected cryo-EM data from nα1-GABA$_A$R–Fab complexes in the presence of DID, ZOL plus GABA (ZOL/GABA), and APG plus GABA (APG/GABA) (Extended Data Table 1), and carried out single-particle analysis. The two-dimensional (2D) class averages derived from all three datasets showed prominent Fab features at the periphery of the receptors. In contrast to a previous study on recombinant α1-containing tri-heteromeric GABA$_A$R complexes, in which all receptors contained two α1 subunits[9], we observed class averages with only one Fab bound (Extended Data Figs. 2–4), demonstrating the presence of receptors with a single α1 subunit.

We subsequently used extensive three-dimensional (3D) classification to investigate subunit composition and arrangement of nα1-GABA$_A$R–Fab complexes. An inverse mask of the entire transmembrane domain (TMD) enabled us to exclude structural heterogeneity in the region of the pore and enabled classification to be driven by the α1-specific Fab and N-glycosylation patterns unique to each α, β and γ subunit (Supplementary Fig. 1). After combining classes with the same Fab and N-glycosylation features, we consistently obtained three different 3D classes: a single class with two Fabs (two-Fab) and two classes with one Fab (one-Fab) (Extended Data Figs. 2–4). We defined the two one-Fab classes as meta-one-Fab and ortho-one-Fab according to the relative position of their α1 and γ subunits. In all three classes from all

three datasets we observed two α subunits, two β subunits and one γ subunit arranged in an α*-β-α-β*-γ clockwise order when viewed from the extracellular side of the membrane (asterisks denote subunits adjacent to the γ subunit). This pentameric configuration therefore represents the dominant form of nα1-GABA$_A$Rs.

In the APG/GABA dataset, 3D reconstructions at resolutions of 2.5 Å, 2.6 Å and 2.6 Å were achieved for the two-Fab, ortho-one-Fab and meta-one-Fab assemblies, respectively. This resolution was sufficient for subunit identification, small-molecule positioning and model building (Fig. 1a, Extended Data Figs. 5 and 6, Extended Data Table 1 and Supplementary Discussion). The identities of β and γ subunits were determined from a combination of glycosylation patterns and side chain densities. Receptors in the two-Fab class had a tri-heteromeric α1*-β2-α1-β2*-γ2 arrangement, consistent with pharmacological[1], immunohistochemical[26] and electrophysiological[27] data suggesting that this is the most abundant subtype in the brain.

By contrast, each of the meta-one-Fab and ortho-one-Fab classes contained mixed receptor ensembles that we categorized as α2/3*-β1/2-α1-β1/2*-γ2 and α1*-β1/2-α2/3-β1/2*-γ2 (Extended Data Fig. 6 and Supplementary Discussion), respectively. A single α1 subunit at one position and either α2 or α3 subunits at the second α position, together with either β1 or β2 subunits at the β position, yielded receptors with at least four, and as many as five, unique subunits in the pentameric assembly. The ambiguities of the α2/α3 and β1/β2 subunit identifications are simply because there are not a sufficient number of differences in amino acid sequence or posttranslational modification to distinguish them from each other. Moreover, these subunit assignments only represent the most abundant subunits based on the cryo-EM maps, with the potential presence of other subunits, such as β3 at the β positions or γ1 at the γ position, if their relative abundance is approximately 20% or less. Both one-Fab classes exhibit ordered N-glycosylation of α subunits in the extracellular vestibule that includes a polysaccharide bridge between the γ subunit and the non-adjacent α subunit, akin to the two-Fab class[9]. Intriguingly, the α2/3 subunit may have a fucose sugar attached to the asparagine-linked N-acetylglucosamine, which is absent in the α1 subunit (Extended Data Fig. 6).

The two one-Fab classes comprise 45% of particles in the APG/GABA dataset and 38% of particles in the ZOL/GABA dataset (Fig. 1 and Extended Data Figs. 2–4), demonstrating that receptors containing one α1 subunit are relatively abundant, consistent with previous studies[20–23,28]. To independently measure the α1 subunit stoichiometry within nα1-GABA$_A$Rs we measured photobleaching of the GFP fluorophore in purified 8E3-GFP complexes using total internal reflection fluorescence (TIRF) microscopy. Roughly 50% of photobleaching events consisted of a single step (Fig. 1b), indicating that about one-half of the purified receptors have one α1 subunit, in agreement with our cryo-EM data.

We used the two-Fab and ortho-one-Fab structures from the APG/GABA dataset as models to compare interdomain arrangements in receptors containing one or two α1 subunits. Despite containing homologous subunits, in which there are 80%/75% amino acid sequence identities between α1 and α2/3 subunits, respectively, we observed conformational differences between the two-Fab and one-Fab complexes. The extracellular domains (ECDs) and TMDs are almost identical in α1 and α2/3 subunits in equivalent positions, having backbone root mean squared deviations (r.m.s.d.) of 0.45 Å and 0.35 Å, respectively. However, when aligned by the TMD, the r.m.s.d. of the ECD increases to 1.05 Å (Supplementary Fig. 2), suggesting notable interdomain displacement between these α1 and α2/3 subunits. Furthermore, both meta-one-Fab and ortho-one-Fab exhibit markedly 'shorter' separations between the ECD and TMD centre of masses than the two-Fab complexes, which is also apparent as a reduction of angles between the primary axes of the ECDs and the TMDs in one-Fab receptors (Extended Data Fig. 7).

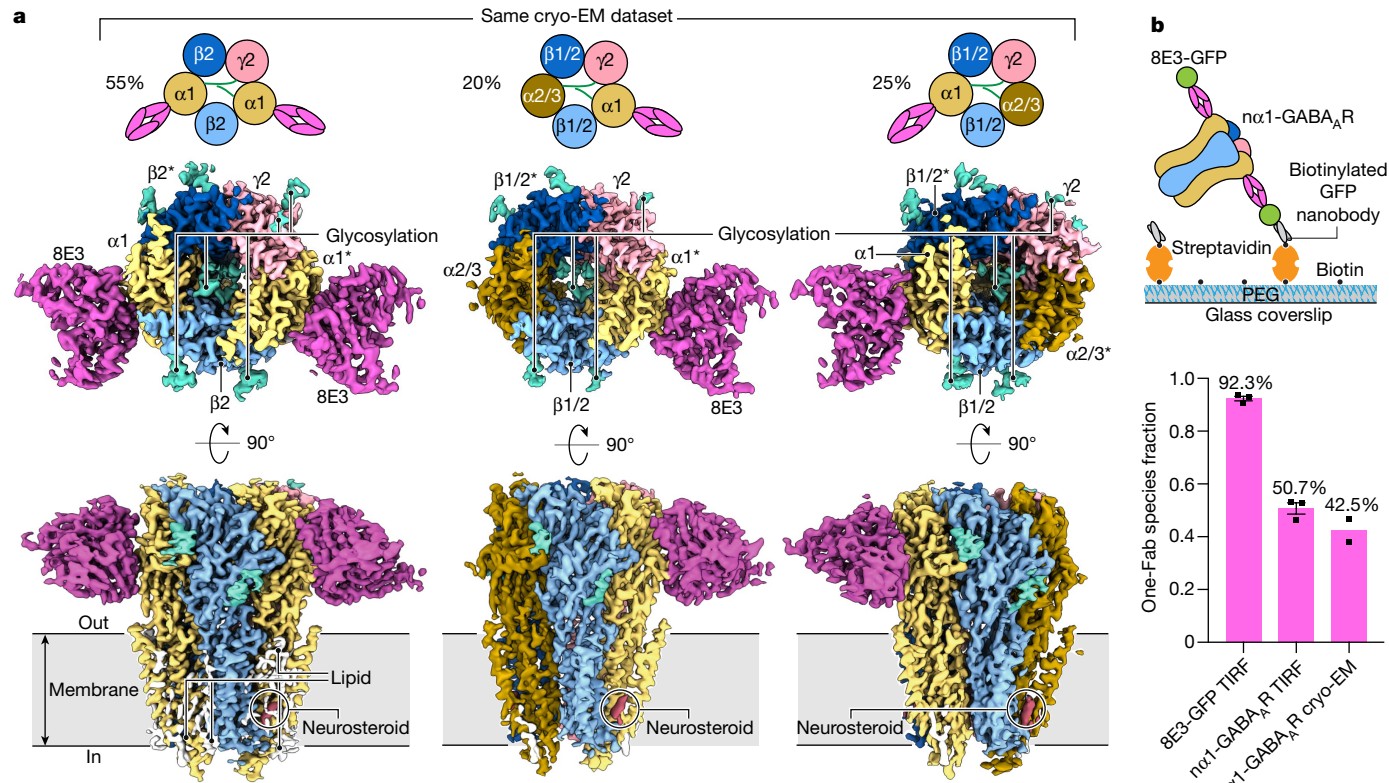

**Fig. 1 | The three major nα1-GABA_AR complexes. a**, Cryo-EM reconstruction of two-Fab α1*-β2-α1-β2*-γ2 (*denotes the subunit that is next to the γ2 subunit, subunits are counted clockwise when viewed from the extracellular space), ortho-one-Fab α1*-β1/2-α2/3-β1/2*-γ2 and meta-one-Fab α2/3*-β1/2-α1-β1/2*-γ2 receptor complexes (from left to right), purified from mouse brains (cerebellum excluded) using an α1-specific Fab. All reconstructions were processed from the APG/GABA dataset. Percentages calculated from particles separated by 3D classification using an inverse TMD mask (Extended Data Fig. 4). **b**, Single-molecule TIRF photobleaching of purified nα1-GABA_AR–GFP-Fab complexes. Top, experimental design; bottom, distribution of photobleaching events for nα1-GABA_AR–GFP-Fab complexes and isolated GFP-Fab (control). Individual data points are presented as squares whereas standard errors of the mean (s.e.m.) are shown as error bars (*n* = 3 photobleaching movies examined over one independent isolated GFP-Fab sample, *n* = 3 photobleaching movies examined over one independent nGABA_AR–GFP-Fab sample, *n* = 2 independent cryo-EM samples). The DID cryo-EM dataset was excluded from the analysis because it was not purified the same way as the TIRF samples and contained a smaller particle count than the other two cryo-EM datasets. PEG, polyethylene glycol.

## Neurosteroid APG modulates nα1-GABA_ARs

Neurosteroids, such as APG (Fig. 2a) and allotetrahydrodeoxycorticosterone, are endogenous ligands that confer anxiolytic, sedative, hypnotic and anaesthetic properties by potentiating the activity of GABA_ARs, and by direct activation at higher concentrations (greater than or equal to 100 nM)[29–32]. To investigate the molecular basis of neurosteroid modulation we compared the structures of two-Fab, meta-one-Fab and ortho-one-Fab assemblies in complex with GABA and APG. In the two-Fab structure, two APG molecules are bound in the TMD region, each approximately 60 Å 'below' one of the two GABA-binding pockets in the ECD (Fig. 2b). The APG pockets are at the interface between transmembrane helices 1 and 4 (M1 and M4) of an α1 subunit, and M3 of the adjacent β2 subunit, which form an almost rectangular box lined by primarily aromatic and hydrophobic residues: α1-W245 on one side, β2-Y304 and β2-L301 at the base, and β2-L297, α1-V242 and α1-I238 on another side. These key residues identified are consistent with previous studies on recombinant GABA_ARs[33,34] and GABA_AR chimeras[35–37]. Remarkably, lipid acyl chains are present on the other two long sides and the box is capped by α1-P400 and α1-Q241, the amide oxygen of the latter forming a hydrogen bond with the 3′-OH of APG (Fig. 2c).

Incorporation of APG remodels the conformation of TMD helices, enlarging the channel pore compared to a recombinant α1β3γ2 structure without neurosteroid (Protein Data Bank (PDB) 6I53;

Supplementary Fig. 3). Local alignment of the β2 and α1* TMDs in the two structures reveals a 2.7° rotation of the line connecting the Cα atoms at the base and top of the APG box (β2-Y304 and α1*-Q241), whereas the length of this line remains constant. In addition, the α1* TMD rotates by 2.8° around an axis between the centre of mass of the entire TMD and the centre of mass of the α1* TMD (Fig. 2d). We observed a similar but smaller effect at the β2*/α1 APG box, with an α1 TMD rotation of 1.8°, suggesting that the two APG pockets in the pentamer have a different molecular pharmacology. Structural comparison with the GABA-bound recombinant α1β2γ2 structure (PDB 6X3Z) revealed similar TMD rearrangements near the APG-binding pockets (Extended Data Fig. 8). Global TMD alignment, on the other hand, highlights a greater tilt of the M2 helices with respect to the pore axis, collectively yielding an enlarged and more symmetric ion channel pore in our APG-bound structure compared to that without APG (Fig. 2e and Extended Data Fig. 9). In particular, the side chains of the 9′-Leu residues, which are crucial for channel gating, are rotated out of the pore in the presence of APG (Fig. 2e).

Neurosteroids achieve GABA_AR potentiation by enhancing the ability of agonists to gate the channel[12,38,39]. Such enhancement must be due to allosteric rearrangements in the GABA-binding ECDs, which we indeed observe in our structure. Specifically, global TMD alignment reveals a concerted, approximately 2° (between 1.5° and 2.5°) anticlockwise rotation of individual ECDs compared to the APG-free structure when viewed from the extracellular side (Extended Data

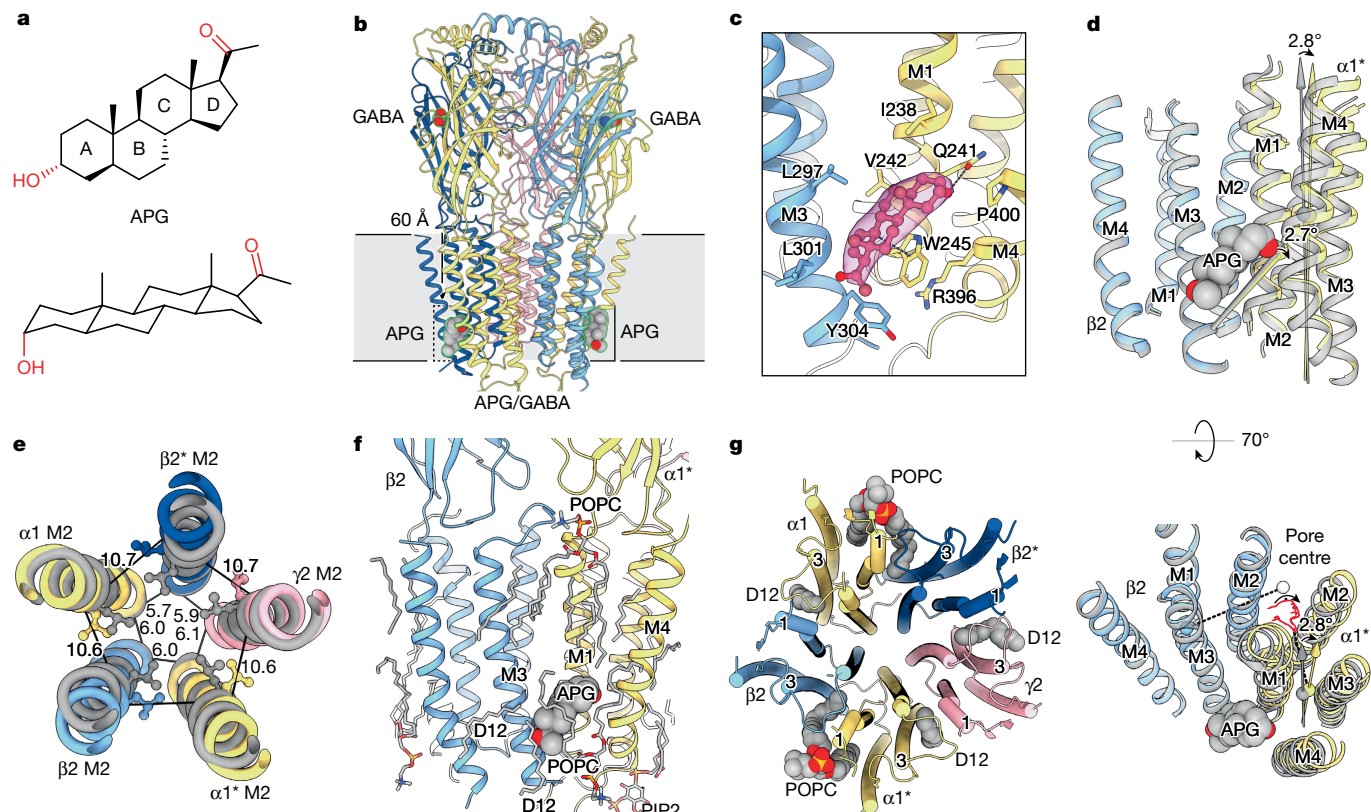

**Fig. 2 | APG sculpts the conformation of the TMD. a**, Chemical structure of APG. **b**, Structural overview of α1*-β2-α1-β2*-γ2 nα1-GABA_AR (*denotes the subunit is next to the γ2 subunit, subunits are counted clockwise when viewed from the extracellular space) in complex with APG and GABA. Bound Fabs hidden for clarity. **c**, Binding pose of APG and ligand density in the binding pocket at the β2^+/α1^*− TMD interface. Cryo-EM density around APG is contoured at 6.6σ. **d**, Local conformational changes induced by APG binding. Coordinates of grey structure of a full-length α1β3γ2 recombinant receptor in complex with GABA are from PDB 6I53. Structural alignment based on the β subunit TMD. **e**, Global structural rearrangements induced by APG binding. Structural alignment based on global TMD. Distances (in Å) are between 9′ gate Cα and the −2′ gate. **f**, Lipid molecules resolved in the APG/GABA structure. **g**, Lipids molecules with acyl tails inserted between M1 and M3 of adjacent subunits.

Fig. 9). This probably accommodates expansion of the TMDs via interactions between the ECD Cys loops and TMD M2−M3 loops. Although GABA binding remains largely unchanged (Extended Data Figs. 8 and 9), concerted ECD rotations may pose an additional energy barrier to GABA release, thus slowing its unbinding and increasing channel gating. Furthermore, because agonist-induced gating is known to be accompanied by anticlockwise rotation of the ECDs[8], our observed conformational changes are compatible with allosteric potentiation of nα1-GABA_ARs by APG. Thus, despite both molecular models in this structural comparison being in a desensitized state with the 2′ gate closed, APG-induced remodelling of TMDs and ECDs suggests how the receptor opens more readily in the presence of APG. Furthermore, our data suggest that direct activation by neurosteroids is mediated via the same two binding pockets, as no additional APG molecules were resolved in samples prepared with APG concentrations as high as 5 μM.

Neurosteroids have unusually slow on- and off-rates compared to more hydrophilic ligands[40,41]. This behaviour has been attributed to their lipophilic nature and tendency to be enriched in the membrane[42], but consideration of lipids in our APG-bound structure offers an additional explanation for this phenomenon. An annulus of lipids with distorted acyl tails completely buries APGs in their binding sites. In total, we resolve nine lipid-like molecules at the β2^+/α1^− interface (^+ denoting the principal face and ^− denoting the complementary face), three being less than 5 Å from APG (Fig. 2f and Supplementary Fig. 4). As a consequence, APG molecules must coordinate with the motions of these annular lipids to secure an exit pathway from the pocket, and partially

dissociated APG molecules may effectively re-engage the receptor without leaving the pocket via the trapping provided by these lipids.

In addition to their prevalence in the APG-binding pockets, lipids structurally engage the receptor at other sites. The greater TMD tilt in our APG-bound structure creates five intersubunit pockets near the centre of the plane of the membrane. All five pockets, including two general anaesthetic-binding sites, are occupied with lipid tails bent like a snorkel (Fig. 2g). Collectively, these lipids serve as small wedges that stabilize the expanded conformation of the TMD.

Both meta-one-Fab and ortho-one-Fab have APG bound at their β2^+/α3^*− or β2^*+/α3^− pockets, demonstrating that neurosteroid binding at the β^+/α^− interface is independent of subunit identity and arrangement within the pentamer (Extended Data Fig. 7). Consistent with this notion, residues involved in binding APG are conserved in all β and α subunits. Thus, we propose that neurosteroid potentiation of all nα1-GABA_ARs with a β^+/α^− interface involves a mechanism similar to the one we have described for APG binding to the native tri-heteromeric α1β2γ2 receptor. Nevertheless, our structures suggest there may be differences in potency or efficacy at each of the neurosteroid-binding sites. Although the sequences forming the immediate APG pockets are identical, the W245 residue (α1 numbering) in other α subunits adopts a different side chain conformer, and can serve as a longer and more effective 'lever' for APG to reshape the TMD and potentiate receptor activity, consistent with previous electrophysiology experiments[43,44].

Strikingly, we observed similar neurosteroid densities in the ZOL/ GABA dataset in the absence of added neurosteroid, which are best modelled as APG molecules (Extended Data Fig. 9). Although we did

not locate any distinct neurosteroid densities in the DID dataset, this is probably due to the lower map resolution. Analysis of our purified ZOL/GABA sample by high-performance liquid chromatography and mass spectrometry confirmed the absence of neurosteroid in the buffer and lipids used for protein purification. However, the same analysis uncovered 115 ng ml$^{-1}$ (362 nM) of neurosteroid in the ZOL/GABA cryo-EM sample (containing approximately 250 nM pentameric receptor), with more than 95% being APG rather than the other three possible stereoisomers (Supplementary Fig. 5). The nearly identical TMD configuration of the APG/GABA and ZOL/GABA structures supports this chemical assignment (Extended Data Fig. 9). Thus, endogenous APG copurified with nα1-GABA$_A$Rs and its stoichiometric presence in our native receptor structures highlights its abundance in the brain and high affinity for nα1-GABA$_A$Rs relative to other endogenous neurosteroids.

## Binding of insomnia drugs to nα1-GABA$_A$Rs

GABA$_A$Rs are the target of a range of insomnia medicines, including flurazepam and ZOL. To investigate the molecular effects of insomnia treatments on nα1-GABA$_A$Rs, we examined the interactions with either DID (inhibition constant, $K_i$ 16.9 ± 1.7 nM), one of the major metabolites of flurazepam[45], or ZOL ($K_i$ 22.9 ± 2.7 nM) and nα1-GABA$_A$Rs (Fig. 3a,b). Both compounds engage the receptor ECD at the α1$^{*+}$/γ2$^-$ interface, which is spatially equivalent to the GABA pockets, each sandwiched at a β$^+$/α$^-$ interface. The binding of DID in the two-Fab dataset is reminiscent of recombinant GABA$_A$R structures in complex with diazepam or alprazolam[8]. DID makes extensive interactions with the receptor, including a hydrogen bond between its carbonyl and the α1-S204 side chain; two hydrogen bonds between its A ring chloride and the α1-H101 and γ2-N60 side chains; and several π−π/CH interactions with α1-F99, α1-Y159, α1-Y209 and γ2-Y58 (Fig. 3c).

ZOL binds to the α1$^{*+}$/γ2$^-$ ECD interface in tri-heteromeric α1β2γ2 receptors at roughly the same position as DID, but engages α1-H101 via π−CH interactions rather than a hydrogen bond. In addition, its amide oxygen forms a hydrogen bond with the α1-S204 side chain, and the imidazo nitrogen forms a separate hydrogen bond with the α1-T206 side chain (Fig. 3d). We hypothesize that this hydrogen bond duet is preserved in interactions with α2 and α3 subunits but not the α5 subunit, in which a threonine residue substitutes for S204. This difference would provide an explanation for the greater than tenfold weaker affinity of ZOL for α5-containing receptors[46]. As is the case for DID, ZOL forms π−π interactions with α1-Y159, α1-Y209 and γ2-Y58, as well as with γ2-F77. The latter interaction explains why ZOL is more sensitive to the γ2-F77I mutation than diazepam[47]. During the preparation of this manuscript, a recombinant GABA$_A$R structure in complex with ZOL was published[48], revealing a similar binding pose for ZOL in the ECD. This structure also captured ZOL in the general anaesthetic-binding pockets at the β2$^+$/α1$^-$ TMD interface using a similar ZOL concentration to that used in our study. We hypothesize that remodelling of the TMDs by endogenous APG prevented ZOL from binding to the general anaesthetic pockets in nα1-GABA$_A$Rs.

Binding of DID or ZOL causes only moderate conformational changes in their binding pockets. We observed a slight opening of loop C due to a 1.2 Å displacement of the γ2-S205 Cα, as well as side chain reorganization of α1-H101, γ2-Y58, γ2-N60 and γ2-F77, which enlarges the pocket to accommodate the ligand. These subtle changes suggest that ZOL-like medications potentiate GABA$_A$Rs via a benzodiazepine-like mechanism, namely strengthening of the α1$^{*+}$/γ2$^-$ interface and facilitating GABA-induced ECD rotation[8]. Indeed, when the TMDs of the ZOL/GABA structure were aligned to the picrotoxin-bound, closed resting structure[8], the ECDs showed concerted anticlockwise rotations ranging from 2° to 5° for individual ECD centres of mass (Fig. 3e).

We also observed ZOL binding to the α1$^{*+}$/γ2$^-$ (ortho-one-Fab) and α2/3$^{*+}$/γ2 (meta-one-Fab) ECD interfaces. Despite sequence differences,

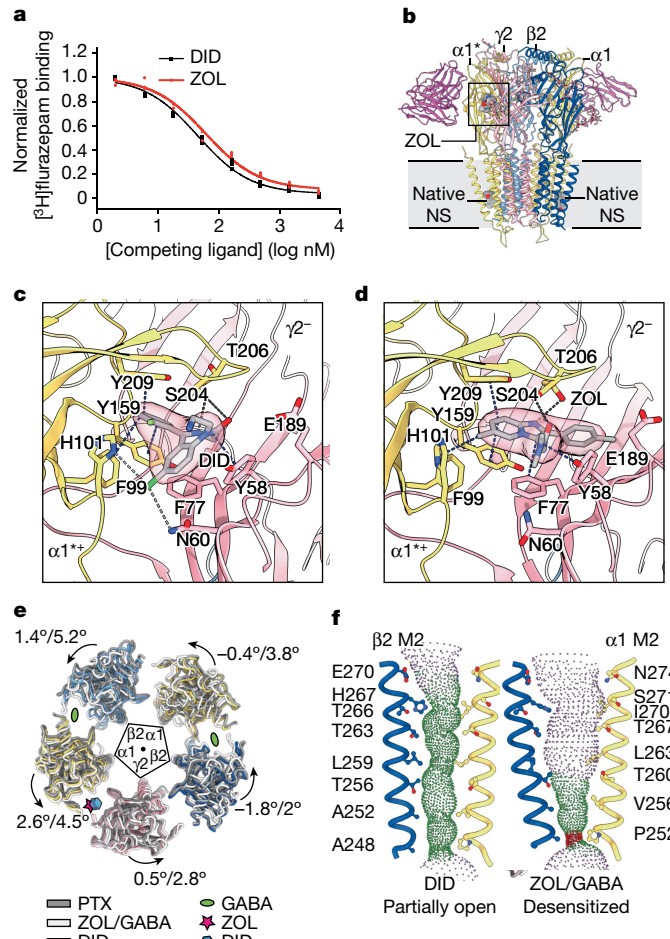

**Fig. 3 | DID and ZOL binding propagates conformational changes to the TMD. a**, Competitive radioligand binding assay for purified nα1-GABA$_A$Rs in complex with ZOL or DID. Individual data points are plotted, along with the curve fitted with the one-site model. **b**, Structural overview of the two-Fab receptor α1$^*$-β2-α1-β2$^*$-γ2 nα1-GABA$_A$R ($^*$denotes the subunit is next to the γ2 subunit) in complex with ZOL, GABA and endogenous neurosteroid (NS). **c,d**, Binding poses of DID (**c**) and ZOL (**d**) and cryo-EM ligand density. Blue dashed lines, π−π/CH interactions; black dashed lines, hydrogen bonds less than 3.5 Å (acceptor to donor); grey dashed lines, weak hydrogen bonds between 3.5 Å and 4 Å. Cryo-EM density around DID and ZOL is contoured at 5.7σ and 5.1σ, respectively. **e**, Effect of ligand binding on ECD arrangement. Coordinates of the closed resting receptor in complex with picrotoxin (PTX) are from PDB 6HUG. Structures superimposed on the basis of global TMD. Individual ECDs displaced from centre of pore by 15 Å for clarity. **f**, Pore profiles in DID structure and ZOL/GABA structure. Pore-delineating dots coloured according to pore radius at that position: red, less than 1.8 Å; green, between 1.8 Å and 4 Å; blue, more than 4 Å.

the immediate α2/3$^{*+}$/γ2 pocket shares the same chemical environment as the α1$^{*+}$/γ2$^-$ pocket, but the side chains adopt different conformations. Accordingly, we observed notably different structural consequences of ZOL binding in the α2/3$^{*+}$/γ2 pocket, including a binding pose closer to loop C on the α2/3 subunit and concerted shifts of the ligand and the protein (Extended Data Fig. 7). As mentioned above, our data suggest that α2/3 subunits have a greater intrinsic bend between their ECD and TMD than α1 subunits, causing different global rearrangements when incorporated into the pentamer. Although this variation in ECD/TMD coupling causes relatively small structural perturbations to orthosteric and allosteric ligand binding, it has the potential to affect channel gating and ligand modulation, which depend on interdomain crosstalk.

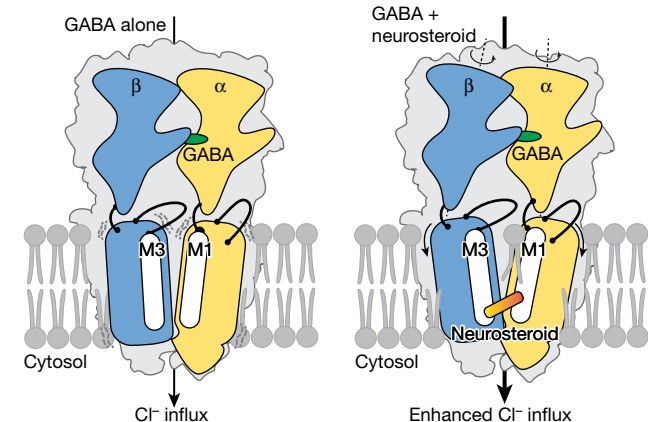

**Fig. 4 | Mechanism of neurosteroid potentiation.** The schematic on the left is based on the full-length recombinant receptor structure (PDB 6I53), whereas the schematic on the right is based on the two-Fab-APG structure from this study. GABA interacts with the receptor at the ECD β[+]/α[−] pocket, inducing anticlockwise rotations of the ECD that reorganize the TMD for Cl[−] ion conduction. Neurosteroids bind to the receptor at the TMD β[+]/α[−] pocket, causing additional anticlockwise ECD rotation and acting as a diagonal brace to stabilize the TMD open-channel conformation. Annular lipids insert their acyl tails between the M3 and M1 helices of adjacent subunits, offering further stabilization of an open-channel conformation.

Intriguingly, the minimum pore radius in the DID structure is 2 Å (Fig. 3f and Supplementary Fig. 6), possibly representing a partially open state of GABA$_A$R that has not been observed before. We speculate this could be due to partial occupancy of the orthosteric binding site by GABA that copurified with the receptor, and by the presence of DID, which may potentiate GABA$_A$R activity similarly to diazepam, as previously reported[49,50]. Accordingly, we observed incomplete loop C closure—the structural hallmark of GABA-dependent allostery—in the GABA-binding pockets in the DID structure (Fig. 3e and Supplementary Fig. 7).

## Conclusion

In summary, the cryo-EM analysis reveals three major structural populations of α1-containing receptors and the single-molecule TIRF experiments are consistent with a substantial fraction of receptors harbouring a single α1 subunit, yet the mass spectrometry studies show that nearly all of the 19 GABA$_A$R subunits are present in the α1-purified preparation, in harmony with the long-standing observation that multiple other subunits assemble with the α1 subunit[19–24]. The apparent discrepancy between the cryo-EM and mass spectrometric experiments is simply grounded in their relative sensitivities. Although the cryo-EM experiments enable the resolution of the major receptor species and their subunit arrangements, they only enable the detection of the most highly populated receptor species. By contrast, mass spectrometry studies enable the detection of less abundant receptor subunits, but do not enable determination of which subunits are in specific receptor assemblies. In combination, the two approaches define the major α1 receptor species present in the brain and at the same time support the presence of multiple, yet less abundant, crucial α1-containing receptor complexes[18,20–23,28]. Future experiments will be required to obtain structural information on these less abundant species.

The molecular structures of the ligand-bound nα1-GABA$_A$R complexes have revealed the binding poses of APG, an endogenous neurosteroid used for treating postpartum depression, and two insomnia drugs, DID and ZOL. Our work highlights the conformational changes induced by neurosteroid binding to native receptors and thus the structural basis for neurosteroid-dependent positive modulation (Fig. 4).

Finally, the serendipitous finding that endogenous neurosteroids remain bound to nα1-GABA$_A$Rs after isolation and purification emphasizes the importance of considering intrinsic neurosteroid modulation when investigating the pharmacology of nα1-GABA$_A$Rs.

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

## Methods

### Expression and purification of the α1-specific 8E3-GFP Fab

The α1-specific mouse monoclonal antibody 8E3 was generated and screened as previously described[9]. The coding sequences of 8E3 Fab light and heavy chains were determined from hybridoma mRNA, and a construct to express the Fab portion of the antibody was designed by including sequences to encode an N-terminal GP64 signal peptide. Codons were optimized for expression in insect cells. To facilitate recombinant antibody detection and purification, a 3C cleavage sequence, an EGFP gene and a twin-strep II tag were added to the C terminus of the heavy chain. Synthetic genes for both chains were then cloned into the pFastBac-Dual vector under the polyhedrin promoter. The recombinant baculovirus was prepared as previously described[51]. Sf9 cells at a density of 3 million per ml were infected with the recombinant baculovirus, with a multiplicity of infection of 2, and further cultured for 96 h at 20 °C. The antibody-containing supernatant was collected by a 20-min centrifugation at 5,000$g$ and then the pH was adjusted to 8 with 30 mM Tris base, incubated in the cold room overnight to enable precipitation of non-Fab protein and clarified by another 20-min centrifugation at 5,000$g$. The supernatant was concentrated and buffer exchanged three times with TBS (20 mM Tris, 150 mM NaCl, pH 8) using a tangential-flow concentrator equipped with a 15 kDa filter. The concentrated supernatant was then loaded onto a 15 ml streptactin column, which was washed with at least 20 column volumes (CV) of TBS and eluted with 5 mM desthiobiotin in TBS. Selected fractions were pooled, concentrated and buffer exchanged to TBS using microconcentrators with a 50 kDa cutoff. Concentrated 8E3-GFP Fab (approximately 100 μM) was aliquoted and stored at −80 °C until use.

### Purification of nα1-GABA$_A$Rs from mouse brains

One-month-old BL/6 mice of mixed sex (approximately 50 mice per preparation) were used for native receptor isolation. The mice were first euthanized and decapitated. The whole brain was isolated from the skull using a laboratory micro spatula and stored in ice-cold TBS. Cerebella were removed from the whole brain, frozen in liquid nitrogen and stored at −80 °C for a separate study. After being washed twice with ice-cold TBS, brain tissue was resuspended with ice-cold TBS (1 ml per brain) supplemented with 0.2 mM phenylmethyl sulfonyl fluoride (PMSF). The suspension was processed with a loose-fit Potter–Elvehjem homogenizer for 20 full up-and-down strokes and further sonicated (1 min per 50 ml) at a setting of 6, typically at a 40 W output. The suspension was centrifuged at 10,000$g$ for 10 min, resulting in a hard pellet of mainly the nuclear fraction and a 'runny' soft pellet containing a substantial amount of nα1-GABA$_A$Rs. The supernatant was further centrifuged at 200,000$g$ for 45 min to pellet the membranes. About 0.1 g of hard pellet and 0.2 g of soft pellet were obtained per mouse brain, on average. These membrane pellets were resuspended with an equal volume of TBS buffer containing protease inhibitors (aprotinin/leupeptin/pepstatin A/PMSF). If not used straightaway, the 50% membrane suspension was supplemented with 10% glycerol and snap frozen in liquid nitrogen.

The following membrane solubilization and affinity chromatography were all carried out at 4 °C. First, LMNG/CHS (10:1 w/w) stock (10% w/v) was diluted to 2.5% in TBS buffer containing protease inhibitors. Then, one volume of the 50% membrane suspension was mixed with two volumes of the diluted detergent stock and incubated for 1 h on a platform rocker, which routinely resulted in the solubilization of about 60% of the α1 subunit present in the tissue, estimated on the basis of western blot (Millipore, catalogue no. 06-868, 1:1,000 dilution) (Extended Data Fig. 1). Next, BioLock solution was added at 0.1 ml per brain to quench the naturally biotinylated proteins, and the mixture was clarified by centrifugation at 200,000$g$ for 1 h. Finally, the 8E3-GFP Fab was added to the solubilized membrane to a concentration between 60 nM and 100 nM. After 1 h incubation, 3 ml of pre-equilibrated streptactin resin

was added to bind the 8E3-GFP Fab and associated nα1-GABA$_A$Rs for 2 h in batch mode.

### On-column nanodisc reconstitution

MSP2N2 (ref. 52) or a recently engineered MSP1E3D1 variant, CSE3 (ref. 53), was used for on-column MSP nanodisc reconstitution. The affinity resin, bound with receptor complexes, was washed in batches, first with 20 CV of ice-cold TBS, then with 20 CV of TBS containing 0.05% LMNG and 0.01% brain polar lipid (Avanti). During this wash, 40 nmol MSP2N2 and 3.2 μmol POPC:bovine brain extract (Sigma) (85:15) lipids, or 40 nmol CSE3 and 4.8 μmol lipids, were mixed to a final volume of 1 ml in TBS and incubated at room temperature for 30 min. The beads were transferred to an empty Econo-Pac gravity flow column (Bio-Rad) to drain the buffer. Then, the 1 ml pre-incubated MSP:lipids were added and incubated for 1.5 h. Next, biobeads were added to a 20× weight excess to the LMNG detergent. The mixture was incubated with a rotator in the cold room for at least 4 h. The biobead/resin mixture was washed with 20 CV of ice-cold buffer to remove unbound empty nanodiscs.

Two approaches were used to elute reconstituted nanodiscs: competitive ligand elution and protease cleavage. For ligand elution, 0.5 CV 5 mM desthiobiotin dissolved in TBS was incubated with the streptactin superflow resin for 10 min before gravity elution, which was repeated a total of six times. In the case of 3C cleavage, 0.1 mg 3C protease was first diluted to 50 μg ml$^{-1}$ with 2 ml TBS and added to the resin. After a 2 h incubation in the cold room, the elution was collected and the column was further washed three times with 2 ml TBS to improve the protein yield. 3C protease cleavage offered better protein purity and was used for the ZOL and the APG samples. Pooled elution was concentrated to about 0.5 ml using a 50 kDa cutoff centricon, regardless of the elution methods. The concentrated sample was then injected into a Superose 6 increase 10/300 GL column (Cytiva) pre-equilibrated with TBS supplemented with 1 mM GABA and other ligands. Selected fractions corresponding to the nα1-GABA$_A$R–Fab complex were combined and concentrated to about 0.1 mg ml$^{-1}$ using a centricon with a 50 kDa cutoff.

### Mass spectroscopy protein identification

**Method 1.** The protein mass spectroscopy analysis was carried out with native receptor samples (approximately 5 μg protein) as previously described[54]. The proteins identified are summarized in Extended Data Fig. 1 and provided in Supplementary Table 1.

**Method 2.** The protein mass spectroscopy analysis was carried out with native receptor samples (approximately 5 μg protein) as previously described[55], except that the canonical Uniprot[56] protein sequences from *Mus musculus* were used during data analysis. Both isoforms of γ2 (short and long) were included to probe their presence with mass spectroscopy. Identified proteins and peptides are provided in Supplementary Tables 2 and 3, respectively.

### Single-molecule photobleaching of nα1-GABA$_A$R–Fab complexes

Coverslips and glass slides were extensively cleaned, passivated and coated with methoxy polyethylene glycol (mPEG) and 2% biotinylated PEG as previously described[57]. A flow chamber was created by drilling 0.75 mm holes in the glass slide and placing double-sided tape between the holes. A coverslip was placed on top of the slide, and the edges were sealed with epoxy, creating small flow chambers. A concentration of 0.25 mg ml$^{-1}$ streptavidin was then applied to the slide, incubated for 5 min and washed off with buffer consisting of 50 mM Tris, 50 mM NaCl and 0.25 mg ml$^{-1}$ bovine serum albumin (BSA), pH 8.0. Anti-GFP nanobody (plasmid of the GFP nanobody was a gift from B. Collins) was expressed and purified according to the published protocol[58]. Biotinylation was carried out with a maleimide-PEG2-Biotin kit (ThermoFisher). Biotinylated nanobody at 7.5 μg ml$^{-1}$ was applied to the slide, incubated

for 10 min and washed off with 30 µl buffer A (20 mM Tris, 150 mM NaCl, pH 8) supplemented with 0.2 mg ml$^{-1}$ BSA. nα1-GABA$_A$R–Fab complexes in nanodiscs were eluted from the streptactin-XT resin, with biotin instead of 3C protease cleavage to preserve the GFP moiety. The sample was further purified by fluorescence-detection size-exclusion chromatography, and the peak corresponding to the complex was hand collected, which separated the native receptor from free Fab. The sample was diluted 1:30 to about 50 pM on the basis of fluorescence quantitation, applied to the chamber and incubated for 5 min before being washed off with 30 µl of buffer A. The chamber was immediately imaged using a Leica DMi8 TIRF microscope with an oil-immersion ×100 objective. Images were captured using a back-illuminated EMCCD camera (Andor iXon Ultra 888) with a 133 µm × 133 µm imaging area and a 13 µm pixel size. This 13 µm pixel size corresponds to 130 nm on the sample due to the ×100 objective.

Photobleaching movies were acquired by exposing the imaging area for 180 s. Single-molecule fluorescence time traces of nα1-GABA$_A$R–Fab were generated using a custom Python script. Each trace was manually scored as having one to three bleaching steps, or was discarded if no clean bleaching steps could be identified. A total of about 450 molecules were evaluated from three separate movies. Scoring was verified by assessing the intensity of the spot; on average, the molecules that bleached in two steps were twice as bright as those that bleached in one step.

## Scintillation proximity assay

YSI copper SPA beads from PerkinElmer were used to capture the nα1-GABA$_A$R in the nanodisc via the MSP His-tag. Tritiated flunitrazepam from PerkinElmer was used as the radioligand, and clorazepate was used as the competing ligand to estimate background. During the ligand-binding assay setup, nα1-GABA$_A$R in the nanodisc was first mixed with SPA beads and radioligand (2× beads), whereas the ligand of different concentrations (2× ligand) and competing ligand (2× background) were prepared using serial dilution. Then, an equal volume of 2× bead was mixed with 2× ligand (in triplicate) or 2× background in a 96-well plate. The final concentrations were 0.5 mg ml$^{-1}$ SPA beads, approximately 1 nM native receptors, 10 nM [$^3$H]flunitrazepam and 0.5 mM clorazepate in the background wells only. The plate was then read with a MicroBeta TriLux after a 2 h incubation. Specific counts were then imported into GraphPad Prism and analysed using a one-site competition model.

## Negative-stain electron microscopy

Purified nα1-GABA$_A$R–Fab complex in nanodiscs was first diluted with TBS to a concentration of approximately 0.05 mg ml$^{-1}$. Continuous carbon grids (Ted Pella, catalogue no. 01844-F) were glow discharged with a PELCO easiGlow unit (Ted Pella) for 60 s at a current of 15 mA. A protein sample (5 µl) was applied to the carbon side of the grid held with a fine-tip tweezer and incubated for 10–30 s. The excessive sample was then wicked away from the side with a small piece of filter paper. The grid was quickly washed with 5 µl deionized water, followed by side wicking, which was repeated a total of three times. Immediately afterwards the grid was incubated with 5 µl 0.75% uranium formate for 45 s, wicked several times from the side and dried for at least 2 min at room temperature.

## Cryo-EM sample preparation and data acquisition

We used a specific setup to streamline the preparation grids under different buffer and ligand conditions. First, buffers containing 10× ligand or additive were prepared and dispensed in 0.5 µl aliquots into a strip of 200 µl microcentrifuge tubes. Then, 5 µl purified nα1-GABA$_A$R–Fab complex was added and quickly mixed by pipetting. Within 10 s, a 2.5 µl sample was applied to a glow-discharged (30 s at 15 mA) 200 mesh gold Quantifoil 2/1 grid overlaid with 2 nm continuous carbon (Ted Pella, catalogue no. 661-200-AU) and incubated for 30 s. The grid was blotted with a Mark IV Vitrobot under 100% humidity at 16 °C and flash frozen

in liquid ethane. For the DID sample, no GABA was included during the purification, and the DID (2 µM) was added before vitrification. For the ZOL sample, 1 mM GABA was included throughout the purification, and 5 µM ZOL was added before vitrification using the above-mentioned PCR tube method. For the APG sample, 1 mM GABA and 5 µM APG were included from the membrane solubilization to the final size-exclusion chromatography.

Cryo-EM data were collected on a 300 keV Titan Krios equipped with a BioQuantum energy filter at either Pacific Northwest Cryo-EM Center or the Janelia cryo-EM facility. Data acquisition was automated using serialEM: defoci ranged between 0.9 to 2.5 µm, holes with suitable ice thickness were selected with the hole finder and combined to produce multishot–multihole targets, which enabled the acquisition of six movies per hole in each of the neighbouring nine holes. These movies were captured with a K3 direct electron detector. A total dose of 50 e$^-$/Å$^2$ was fractionated into 40 frames, with a dose rate of about 15 electrons per pixel per second for non-CDS mode or 7 electrons per pixel per second for CDS mode (Extended Data Table 1).

## Cryo-EM data analysis

Super-resolution movies were imported to cryoSPARC[59] v.3.3.1 and motion corrected using patch motion correction in cryoSPARC, with the output Fourier cropping factor set to ½. Initial contrast transfer function (CTF) parameters were then calculated using the patch CTF estimation in cryoSPARC. For each dataset, 2D class averages of particles picked by glob picker from approximately 1,000 micrographs were used as templates for the template picker. One round of 2D classification and several rounds of heterogeneous refinement seeded with ab initio models generated within cryoSPARC were used to select GABA$_A$R particles, ranging from 4 to 6 million particles for our datasets. A non-uniform refinement (NU-Refinement) was performed to align these particles to a consensus structure. Two downstream strategies were used for our datasets (strategy 1 for the DID dataset, strategy 2 for the ZOL/GABA and the APG/GABA datasets), as described below.

**Data processing strategy 1.** Bin1 GABA$_A$R particles, both images (360 × 360) and the star file converted using pyem[60], were ported into RELION[61] v.3.1. Then, a 3D autorefinement job with local search (angular sampling of 1.8°) was carried out to fine tune the particle poses in RELION. The refined structure, similar to that generated by cryoSPARC, had relatively weaker γ subunit transmembrane helices, which was reported previously[10]. To tackle this issue, we prepared a nanodisc mask in Chimera[62] and carried out 3D classification without alignment (15 classes, regularization parameter $T = 20$) using that mask. The 3D classification can robustly give classes with much stronger transmembrane helices of the γ subunit. Those selected particles were imported into cryoSPARC and further refined using NU-Refinement with both defocus refinement and per-group CTF refinement options turned on. The consensus structure was a two-Fab bound structure, but earlier data processing revealed the presence of one-Fab species. Therefore, a 3D classification job was used with a mask focusing on the two binding sites of 8E3 Fab to isolate the one-Fab species. The one-Fab and two-Fab particles were separately refined with NU-Refinement and further refined with local refinement.

**Data processing strategy 2.** In this strategy the heterogeneity in Fab binding was addressed upstream in the data processing pipeline. As for strategy 1, GABA$_A$R particles, at bin3 or 120 × 120, were imported into RELION for focused 3D classification. A reverse mask was prepared in cryoSPARC, which only excluded the TMD, to enable Fab binding at all possible positions. The 3D classification (10 classes, $T = 20$) gave clear two-Fab and one-Fab classes, and classes with incomplete Fab. Further 3D classification on these incomplete Fab particles produced only incomplete Fab classes, which led us to believe they were damaged particles and should be excluded from downstream processing. The

two-Fab particles and the one-Fab particles, on the other hand, were imported into cryoSPARC, re-extracted at bin1 and separately refined with NU-Refinement. We still saw weak transmembrane helices for the γ subunit for the one-Fab and the two-Fab populations. To tackle this issue, instead of the 3D classification in RELION, we used the 3D classification (beta) job in cryoSPARC with a nanodisc mask, which was less robust but faster. Classes with stronger transmembrane helices were then combined and refined with NU-Refinement and finished with local refinement.

Global sharpening worked suboptimally for our nα1-GABA$_A$R structures because of the local resolution variation and the lower signal-to-noise ratio for the TMD. The best method to sharpen our maps was achieved with LocScale[63], which was used to represent some of our structures in Fig. 1. DeepEMhancer[64] can yield comparable sharpening for the protein but not for the annulus lipids.

### Subunit identification, model building, refinement and validation

Due to the subunit specificity of 8E3 Fab, the subunit with 8E3 Fab bound is defined as α1. The remaining subunits can be easily classified as α, β or γ from the characteristic N-linked glycosylation patterns of each subunit. It was clear that all 3D classes obtained were α-β-α-β-γ clockwise when viewed from the extracellular side of the membrane. Given the relative subunit abundance from earlier studies, we used α1-β2-α1-β2-γ2 as the starting model of the two-Fab class. We then examined the cryo-EM density maps to test our assignment in the context of sequence information. Specifically, we carefully examined residues for which the side chain can be unambiguously assigned and in which there is a difference of more than three carbon atoms or one sulfur atom between respective residues of the different receptor subunits. Regarding the non-α1 α subunit in the one-Fab classes, we further limited our scrutiny to positions showing no notable conformational differences in the corresponding two-Fab structure, to ensure the observed density difference was caused by the chemical identity of underlying residues.

For each dataset, the two-Fab bound nα1-GABA$_A$R model was built first. The starting structures used were AlphaFold[65] models of mouse GABA$_A$R subunits and the best 8E3 Fab model generated with Rosetta[66]. These individual chains were first docked into the unsharpened cryo-EM density maps using the 'fit-in-map' tool of Chimera to assemble the full receptor–Fab complex. The full complex was then edited to remove unresolved portions and refined extensively to achieve better model–map agreement in Coot[67]. N-glycosylation was modelled using the carbohydrate module in Coot. Lipid and lipid-like molecules, including POPC, PIP2, dodecane and octane, were modelled using the CCP4 monomer library. New ligands included in this study, including their optimized geometry and constraint, were generated using phenix.elbow[68]. After the initial modelling, multiple runs of phenix. real_space_refinement[69] and editing in Coot were carried out to improve the model quality.

The optimized two-Fab GABA$_A$R structure was used as the starting model for one-Fab GABA$_A$R structures. Although the one-Fab population probably consists of a mixture of α2/3 subunits at the α positions and a mixture of β1/β2 subunits at the β positions, we decided to use the α3 subunit and the β2 subunit for the modelling and subsequent structural comparisons, on the basis of our best interpretations of the density maps. We emphasize, nevertheless, that both α2 and β1 models reasonably fit the density maps. Shown in Extended Data Fig. 6 are sequence relationships between the subunits at chosen regions. The two-Fab structure was first docked into the one-Fab cryo-EM map using the 'fit-in-map' tool of ChimeraX[70]. Then, the aligned structure was edited in Coot to remove the extra Fab, replaced and renumbered the α1 sequence with the α3 sequence. This edited structure was further fitted and refined in Coot, first with secondary structure restraints generated with ProSMART[71], and then without the restraints. Furthermore, certain residues and lipids were removed due to less clear density, and the glycosylation trees were remodelled. Similarly, this initial model was subjected to multiple runs of phenix.real_space_refinement and editing in Coot.

### Animal use statement

Mouse carcasses donated from other laboratories of the Vollum Institute were used to establish and optimize the native GABA$_A$ receptor isolation workflow. The quantity of purified native receptor from each mouse was estimated using the fluorescence from the recombinant antibody fragment, which was then extrapolated to give the minimum number required for cryo-EM and biochemical analysis. For each native GABA$_A$ receptor preparation, 50 one-month-old (4–6 weeks) C57BL/6 mice (both male and female) were ordered from Charles River Laboratories. The housing conditions were set as: temperature 20–22 °C, humidity 40–60%, dark/light cycle 12:12 h. No randomization, blinding or experimental manipulations were performed on these animals. All mice were euthanized under the OHSU Institutional Animal Care and Use Committee (IACUC) protocols, consistent with the recommendations of the Panel on Euthanasia of the American Veterinary Medical Association (AVMA) and carried out only by members of the E.G. laboratory approved on IACUC protocol TR03_IP00000905.

### Cell line statement

Sf9 cells for generation of baculovirus and expression of recombinant antibody fragment were from ThermoFisher (12659017, lot 421973). The cells were not authenticated experimentally for these studies. The cells were tested negative for mycoplasma contamination using the CELLshipper Mycoplasma Detection Kit M-100 from Bionique.

### Reporting summary

Further information on research design is available in the Nature Portfolio Reporting Summary linked to this article.

### Data availability

The cryo-EM maps and coordinates for the native GABA$_A$ receptor in complex with DID and endogenous GABA (two-Fab-DID) have been deposited in the Electron Microscopy Data Bank (EMDB) under accession number EMD-29728 and in the Protein Data Bank (PDB) under accession code 8G4O. The cryo-EM maps and coordinates for the native GABA$_A$ receptor in complex with ZOL, GABA and endogenous neurosteroids have been deposited and can be accessed via (EMDB/ PDB) codes: EMD- 39727/8G4N (two-Fab-ZOL), EMD-29743/8G5H (ortho-one-Fab-ZOL), EMD-29742/8G5G (meta-one-Fab-ZOL). The cryo-EM maps and coordinates for the native GABA$_A$ receptor in complex with GABA and APG have been deposited and can be accessed via EMD-29350/8FOI (two-Fab-APG), EMD-29741/8G5F (ortho-one-Fab-APG), EMD-29733/8G4X (meta-one-Fab-APG).

### Code availability

Custom code used for analysing single-molecule photobleaching trajectories in this study is available at Zenodo (https://doi.org/10.5281/ zenodo.8161179).

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

**Acknowledgements** We thank J. Luo and A. DeBarber for mass spectroscopy neurosteroid analysis, J. Guidry and A. Reddy for mass spectroscopy protein identification and D. Claxton and D. Cawley for the monoclonal antibody. We acknowledge use of the Oregon Health and Science University (OHSU) Bioanalytical Shared Resource/Pharmacokinetics Core Facility expertise and instrumentation (Research Resource Identifier (RRID): SCR_009963), the OHSU Multiscale Microscopy Core (MMC), the Pacific Northwest Cryo-EM Center (PNCC) and the cryo-EM facility at Janelia research campus. The OHSU Bioanalytical Shared Resource/Pharmacokinetics Core Facility is supported in part by the University Shared Resource Program at OHSU. The OHSU Proteomics Shared Resource is partially supported by NIH core grants P30EY010572, P30CA069533, S10OD012246, S10RR025571 and OHSU Emerging Technology fund. PNCC is supported by NIH grant U24GM129547 and accessed through EMSL (grid.436923.9), a DOE Office of Science User Facility sponsored by the Office of Biological and Environmental Research. This work was supported by NIH grant 5R01GM10040 to E.G. E.G. is an investigator of the Howard Hughes Medical Institute and thanks B. LaCroute and J. LaCroute for generous support.

**Author contributions** C.S. and E.G. designed the project. C.S. prepared cryo-EM samples and carried out biochemical characterizations. C.S. and S.C. performed single-molecule photobleaching experiments. C.S. and H.Z. carried out the cryo-EM data analysis and C.S. built the molecular models. C.S. and E.G. wrote the manuscript, with input from all authors.

**Competing interests** The authors declare no competing interests.

**Additional information**
**Correspondence and requests for materials** should be addressed to Eric Gouaux.

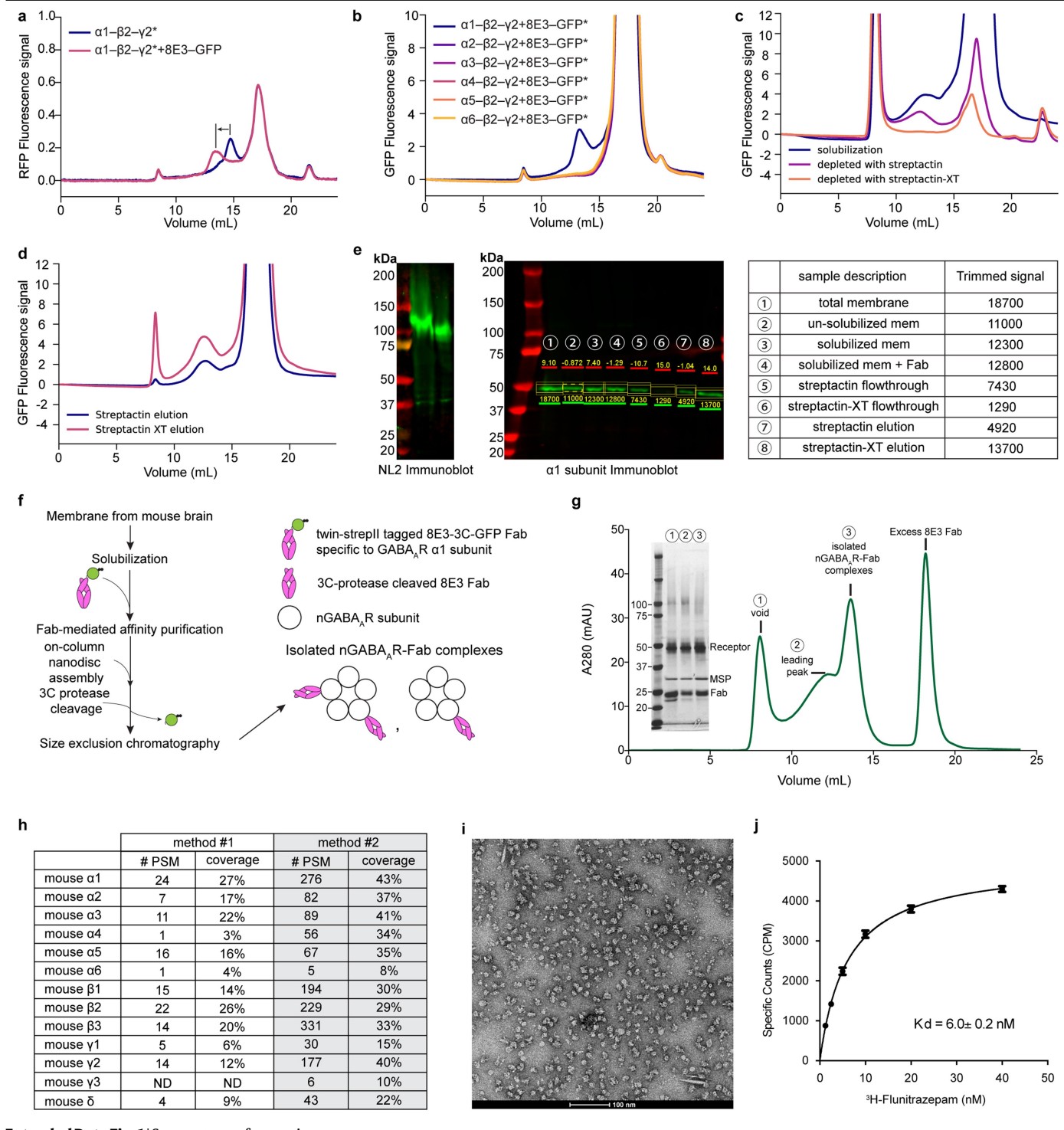

**Extended Data Fig. 1** | See next page for caption.

**Extended Data Fig. 1 | Biochemical characterization of native receptor isolation from mouse brains using an engineered Fab fragment. a** and **b**, Expression of tri-heteromeric GABA$_A$Rs with different α subunits and binding test with the engineered 8E3-GFP Fab monitored with fluorescence-detection size-exclusion chromatography (FSEC). The signal is from the fusion-red protein inserted into the intracellular loop of the γ2 subunit in panel **a** or the GFP of the 8E3-GFP Fab in panel **b**. **c** and **d**, FSEC traces demonstrating the superior capturing efficiency and protein yield of streptactin-XT resin. **e**, Western blot analysis of steps during the native receptor purification. The neuroligin 2 (NL2) immunoblot (Synaptic Systems 129 202, 1:1000 dilution) shows robust solubilization of the inhibitory synapse marker NL2. Lanes from left to right were the membrane input and the LMNG solubilized supernatant. The α1 subunit immunoblot (Millipore 06–868, 1:1000 dilution) shows quantitation of the α1 subunit during membrane solubilization, affinity capturing, and elution steps. The numbers on green lines are trimmed signal from the 800 nm channel (shown as green) used for quantitation. The numbers on red lines are trimmed signal from the 700 nm channel (shown in red), which should be close to zero. Western blot detection of the NL2 and the quantitative analysis of the α1 subunit were repeated twice with comparable results. **f**, The workflow of nα1-GABA$_A$Rs purification from mouse brains. **g**, Size-exclusion chromatography (SEC) of nα1-GABA$_A$Rs and silver-stain SDS-PAGE analysis of different SEC fractions. The SEC and SDS-PAGE analysis on nα1-GABA$_A$Rs were repeated more than 5 times with similar peak profile and band pattern. **h**, Identification of GABA$_A$R subunits from the pentameric peak using mass spectrometry. #PSM indicates the number of peptide spectrum matches, coverage refers to the sequence coverage of protein subunits from identified peptides, and ND indicates 'not detected'. **i**, Negative-staining electron microscopy images of protein samples from the pentameric peak. **j**, Scintillation proximity assay of the pentameric peak fraction with $^3$H-flunitrazepam, at each concentration the specific count is shown as mean ± s.d. (n = 3 replicates prepared from 1 independent native receptor preparation).

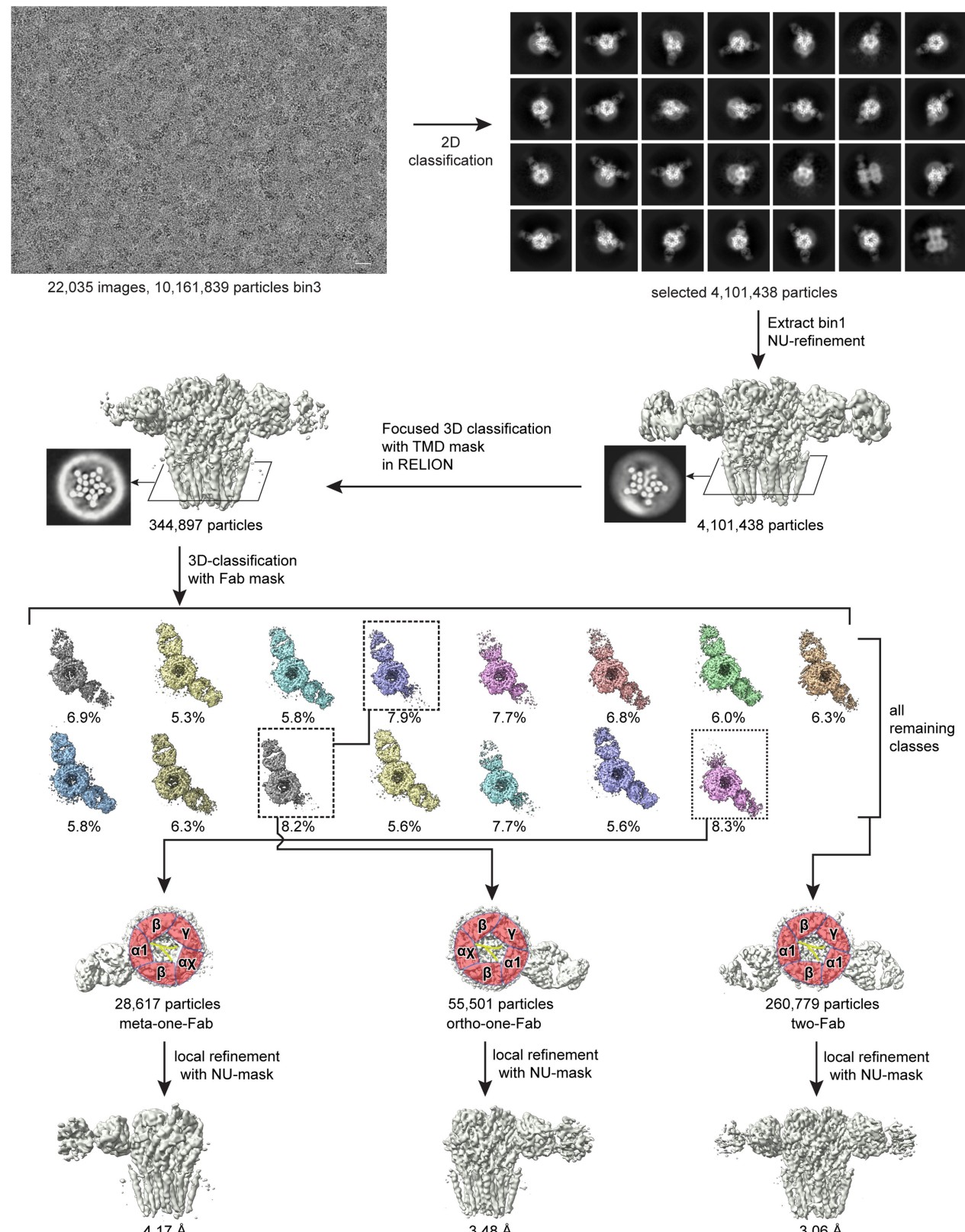

**Extended Data Fig. 2 | Cryo-EM data processing of the DID dataset.** Refer to the "cryo-EM data analysis" method section for more details. Scale bar, 20 nm.

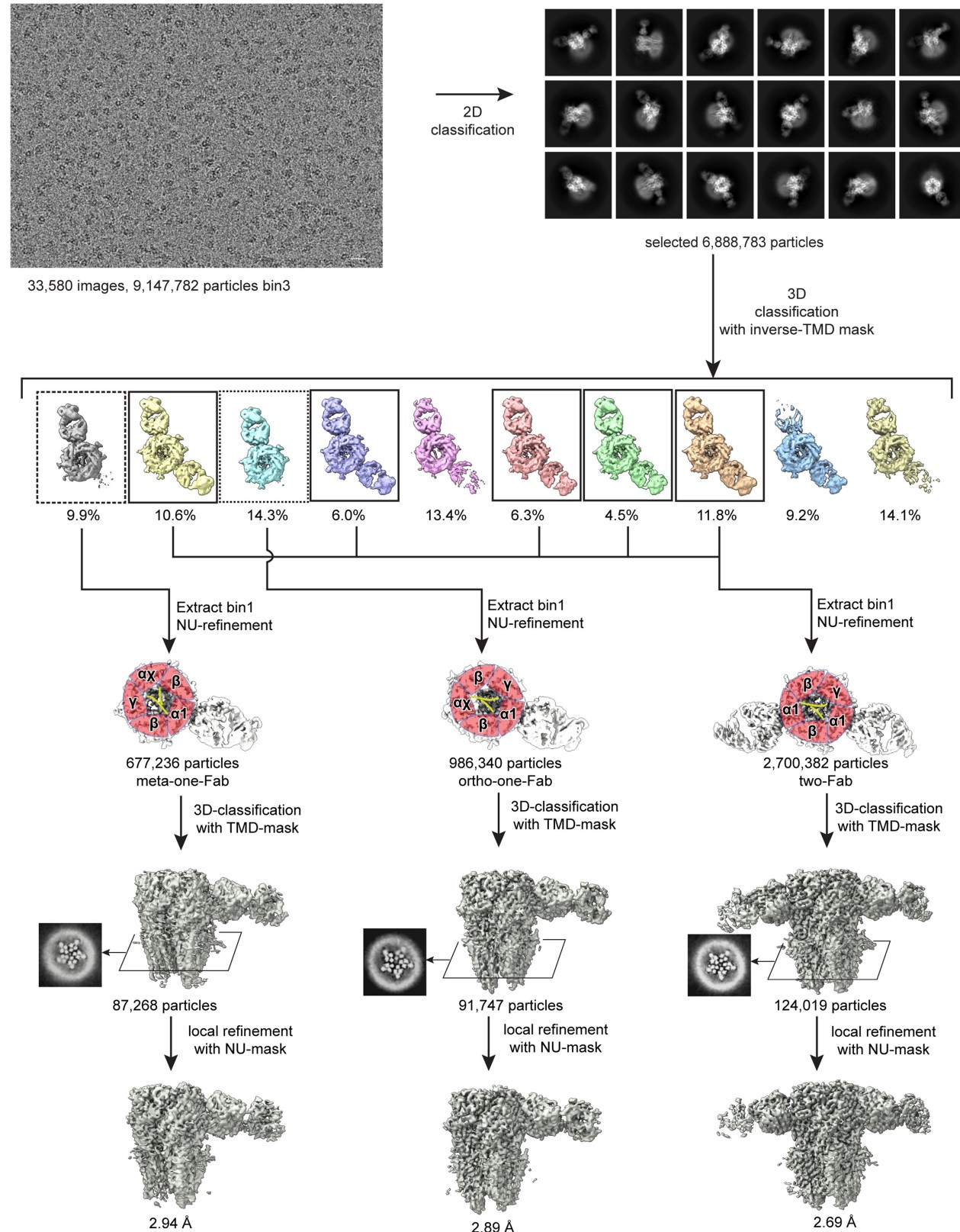

33,580 images, 9,147,782 particles bin3

2D classification

selected 6,888,783 particles

3D classification with inverse-TMD mask

9.9%   10.6%   14.3%   6.0%   13.4%   6.3%   4.5%   11.8%   9.2%   14.1%

Extract bin1 NU-refinement

Extract bin1 NU-refinement

Extract bin1 NU-refinement

677,236 particles meta-one-Fab

986,340 particles ortho-one-Fab

2,700,382 particles two-Fab

3D-classification with TMD-mask

3D-classification with TMD-mask

3D-classification with TMD-mask

87,268 particles

91,747 particles

124,019 particles

local refinement with NU-mask

local refinement with NU-mask

local refinement with NU-mask

2.94 Å

2.89 Å

2.69 Å

**Extended Data Fig. 3 | Cryo-EM data processing of the ZOL/GABA dataset.** Refer to the "cryo-EM data analysis" method section for more details. Scale bar, 20 nm.

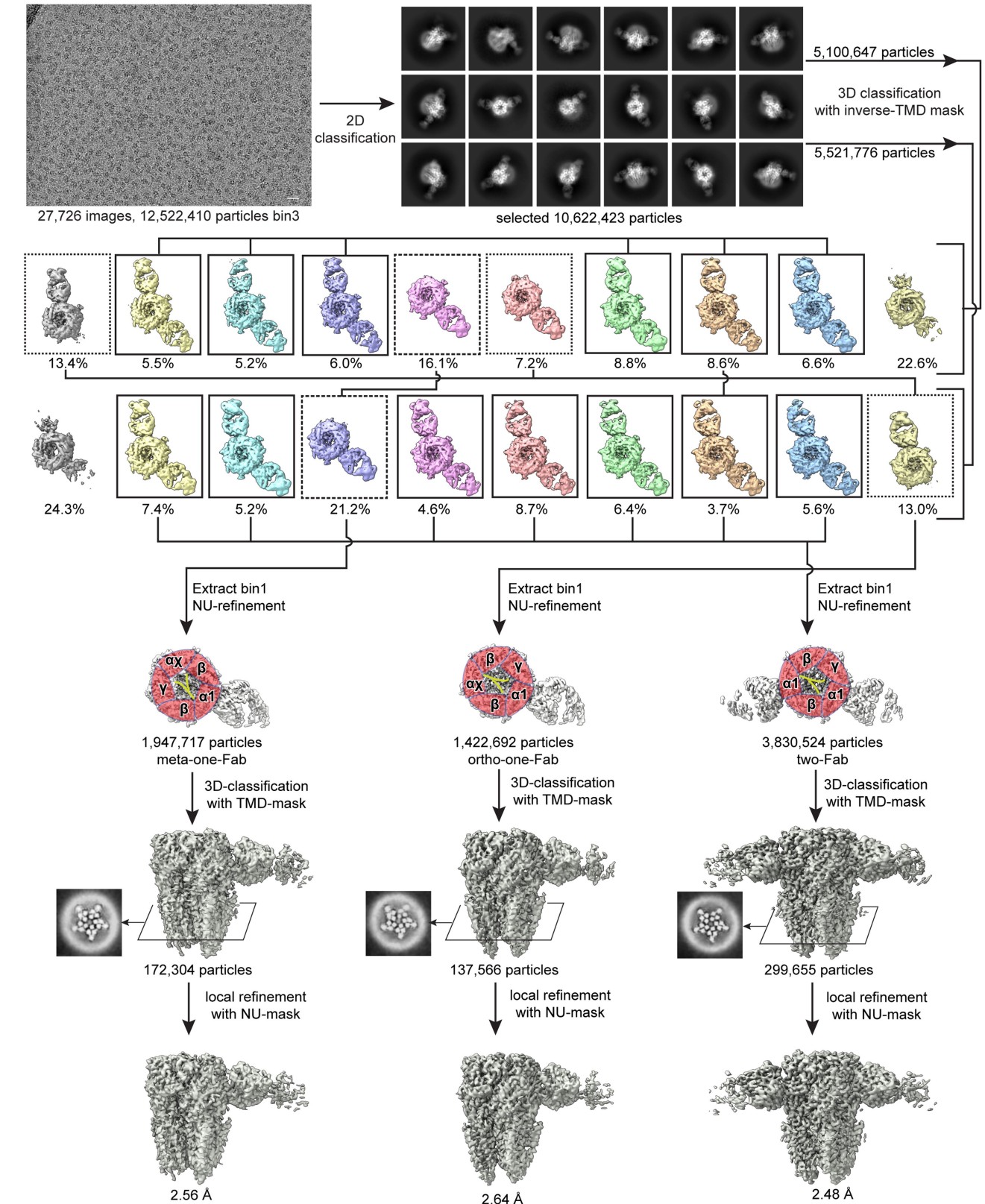

**Extended Data Fig. 4 | Cryo-EM data processing of the APG/GABA dataset.** Refer to the "cryo-EM data analysis" method section for more details. Scale bar, 20 nm.

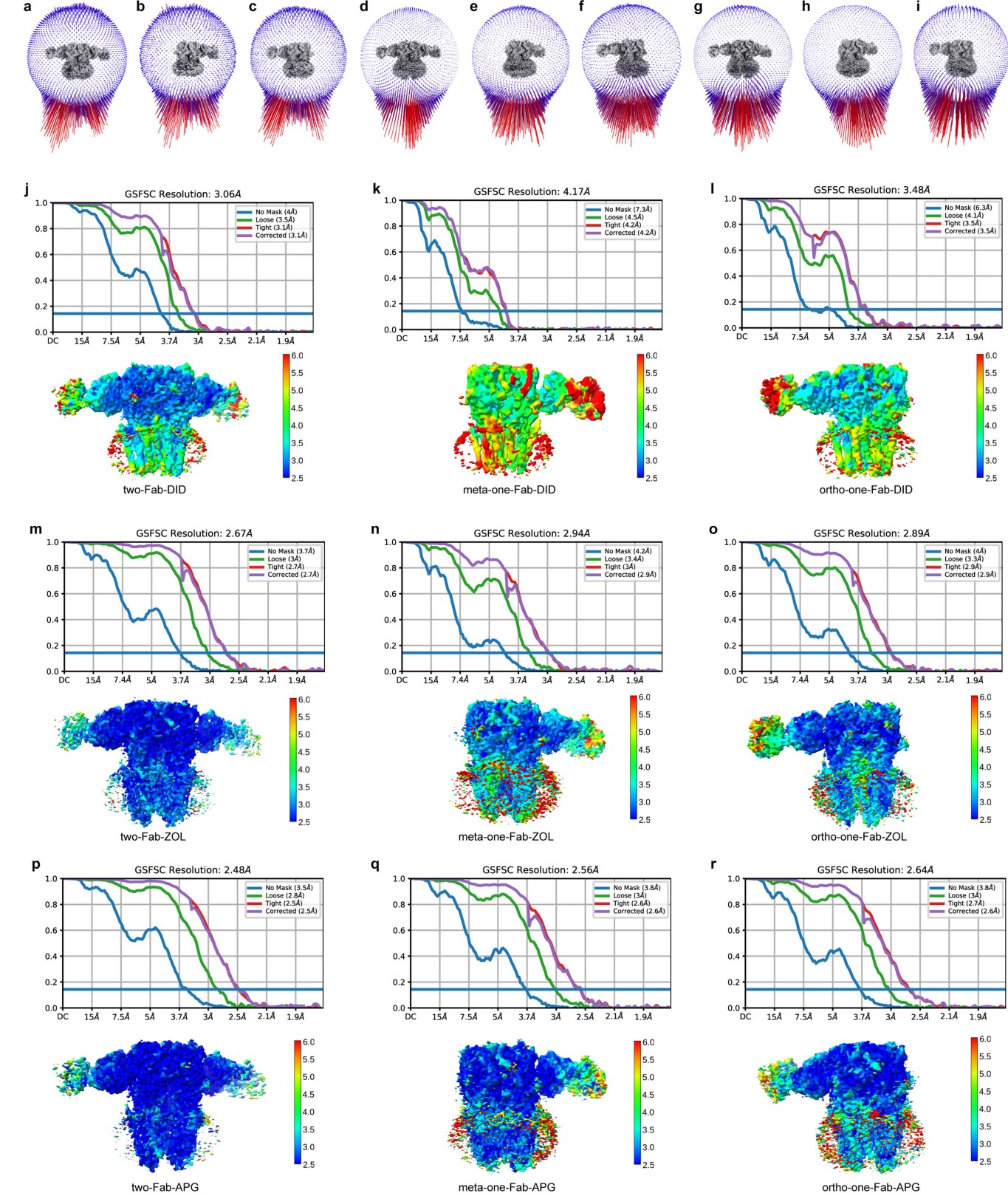

**Extended Data Fig. 5 | Statistics of final cryo-EM reconstructions. a–i**, Euler angle distributions of particles used for final cryo-EM reconstruction of two-Fab-DID (**a**), meta-one-Fab-DID (**b**), ortho-one-Fab-DID (**c**), two-Fab-ZOL (**d**), meta-one-Fab-ZOL (**e**), ortho-one-Fab-ZOL (**f**), two-Fab-APG (**g**), meta-one-Fab-APG (**h**), ortho-one-Fab-APG (**i**). **j–r**, FSC curves and local resolution plots of final cryo-EM reconstructions.

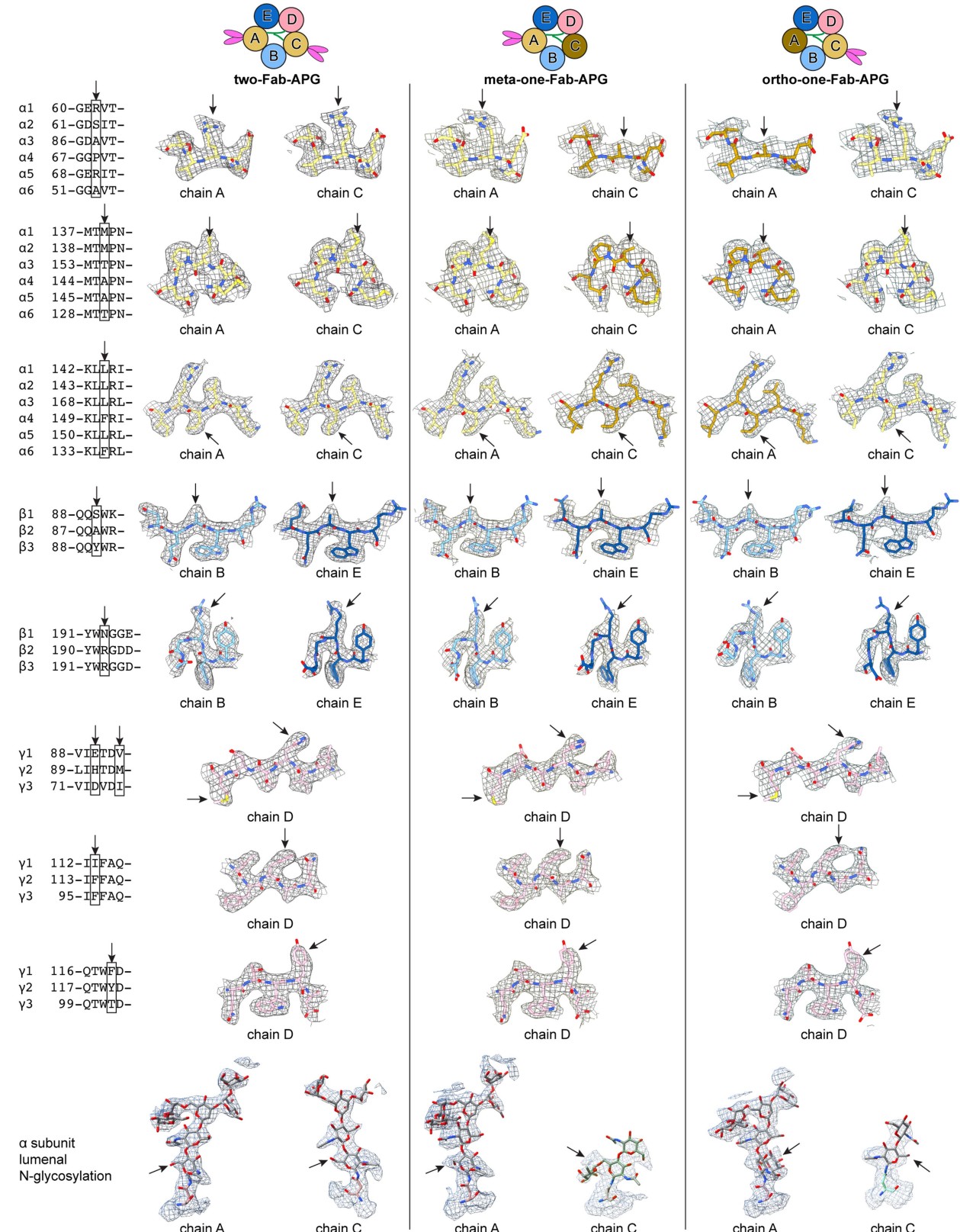

**Extended Data Fig. 6 | Cryo-EM densities of protein side chains and N-glycosylation used for subunit identification.** Cryo-EM maps from the APG/GABA dataset were analyzed alongside sequences and atomic models to identify subunits that are not bound to the antibody fragment. The figure's top section displays schematics of the three identified populations, with the five GABA$_A$R subunits labeled by chain IDs and color-coded. In each row, aligned sequence segments are presented on the left, with the distinguishing residues encased in boxes. Following this, cryo-EM densities contoured at the same level from these three populations (separated by a vertical bar) surrounding these residues are shown, with arrows pointing to the distinguishing residues. By comparing the models and cryo-EM densities within the sequence context, subunits can be assigned or excluded (detailed description in the Supplementary Information). The final row showcases the lumenal N-glycosylations from α subunits instead of amino acid residues, with arrows pointing to the positions of a fucose moiety (absent in α1 subunit) linked to the first *N*-acetylglucosamine of α2/α3 subunits.

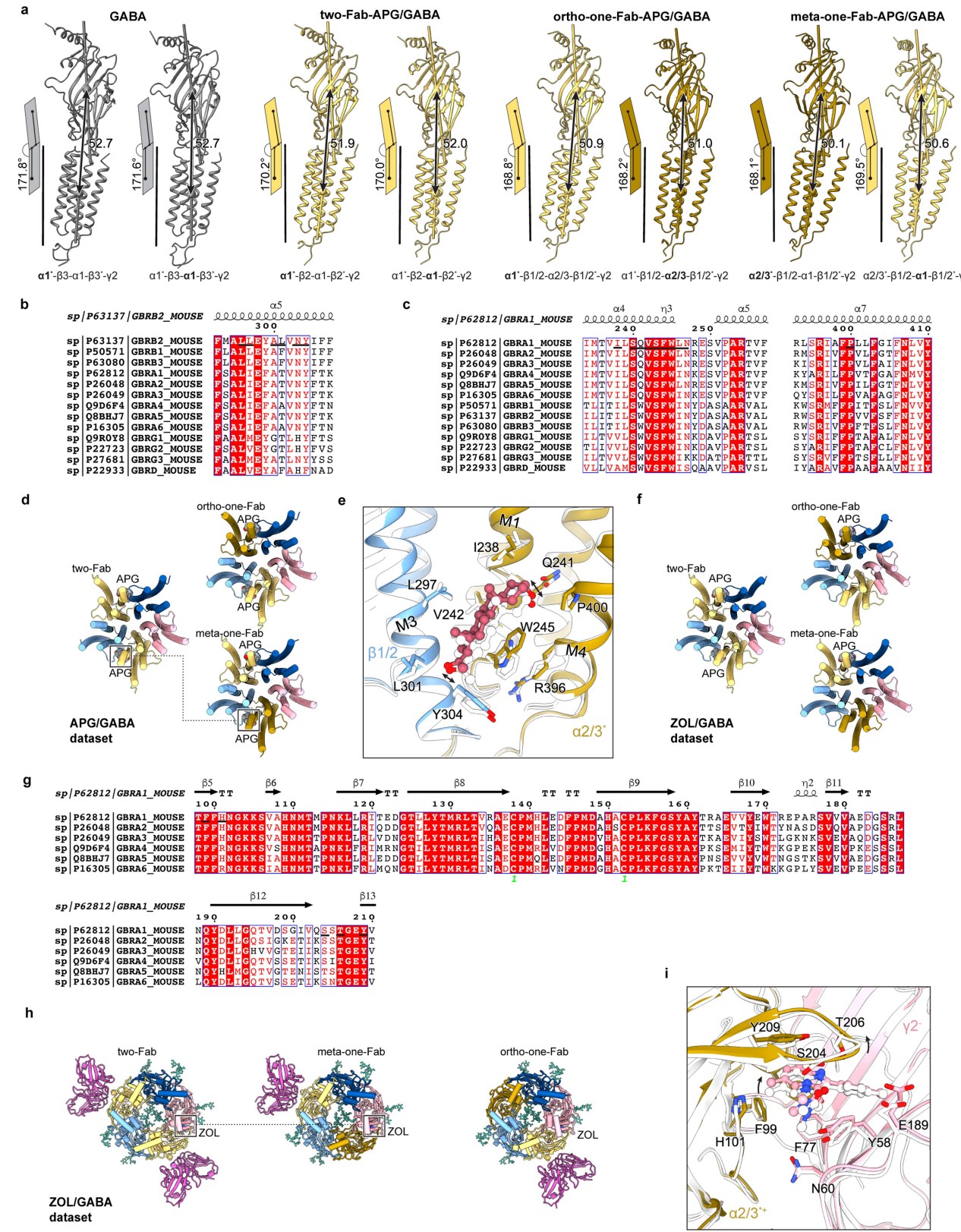

**Extended Data Fig. 7** | See next page for caption.

**Extended Data Fig. 7 | Sequence alignments of nα1-GABA$_A$R subunits and structural variations of nα1-GABA$_A$R assemblies. a**, Differential inter-domain arrangements of α subunits from the nα1-GABA$_A$Rs of the APG/GABA dataset and a previous GABA$_A$R structure (PDB code: 6I53). **b** and **c**, Sequence alignments of nα1-GABA$_A$R subunits with sequence ranges relevant to neurosteroid binding. **d**, TMD structures from the APG/GABA dataset with the allopregnanolone (APG) shown in Vdw representation. **e**, Structure comparison of the APG binding pockets between two-Fab (shown in white) and meta-one-Fab (shown in blue and yellow). The two structures are overlayed based on the TMD of adjacent β and α subunits. **f**, TMD structures from the ZOL/GABA dataset with the endogenous neurosteroid molecules shown in Vdw representation. **g**, Sequence alignments of nα1-I α subunits with sequence ranges relevant to zolpidem binding. **h**, ECD structures from the ZOL/GABA dataset with the zolpidem shown in Vdw representation. **i**, Structure comparison of the ZOL binding pockets between two-Fab (shown in white) and meta-one-Fab (shown in yellow and red). The two structures are overlayed based on the ECD of adjacent β and α subunits.

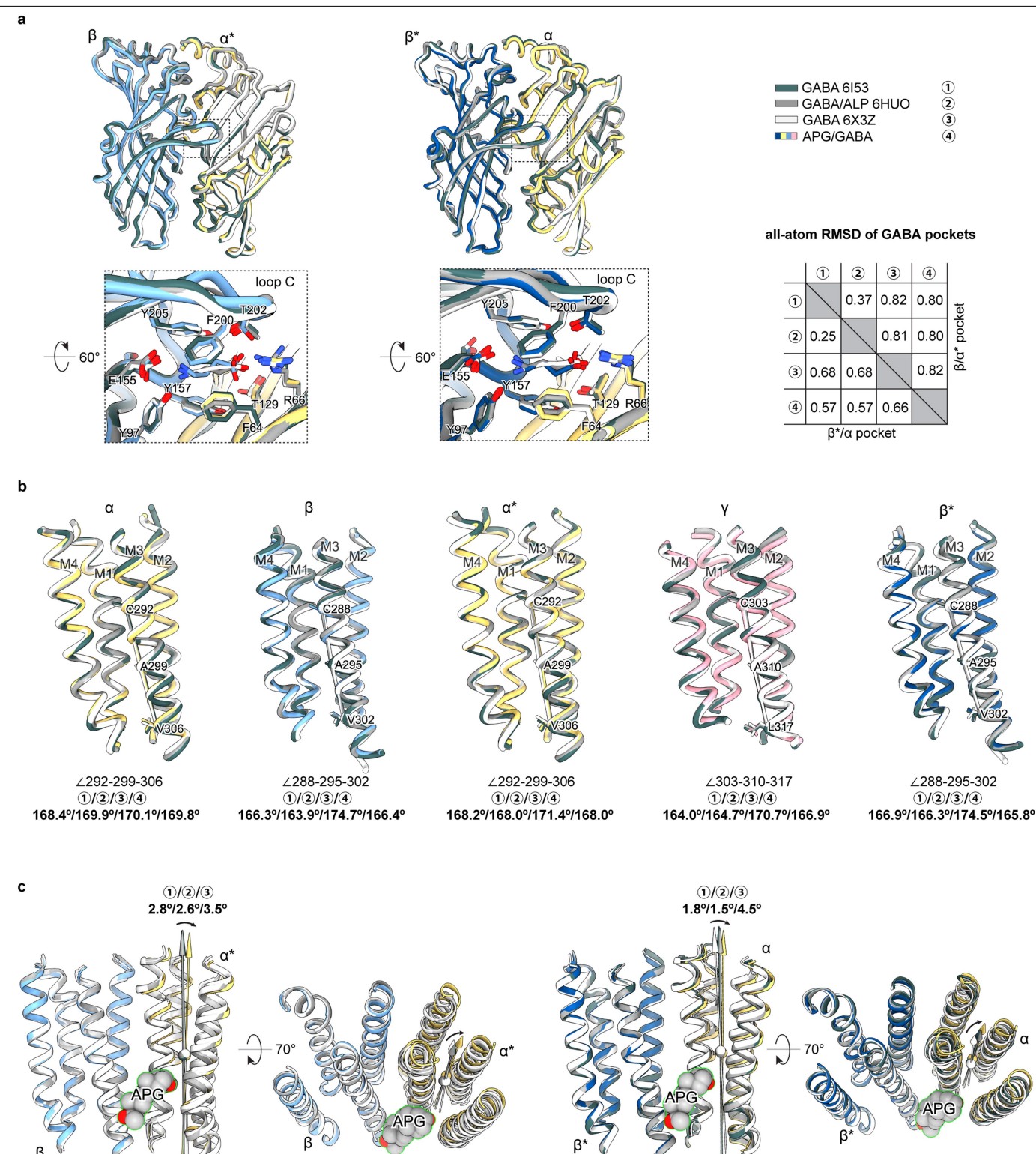

**Extended Data Fig. 8** | See next page for caption.

**Extended Data Fig. 8 | Comparative analysis of the allopregnanolone (APG) bound structure with previous apo structures. a**, Structural superposition based on the extracellular domains (ECD) of adjacent β and α subunits. The compared structures include a recombinant GABA-bound α1*-β3-α1-β3*-γ2 receptor (*denotes the subunit is next to the γ2 subunit; subunits are counted clockwise when viewed from the extracellular space; PDB code 6I53; colored dark gray), a recombinant α1*-β3-α1-β3*-γ2 receptor bound with GABA and alprazolam (PDB code 6HUO; color gray), a recombinant GABA-bound α1*-β2-α1-β2*-γ2 receptor (PDB code 6X3Z; colored white), and the native GABA/APG-bound α1*-β2-α1-β2*-γ2 receptor from this study (colored in yellow, blue, and red). These structures are presented in the same color schemes and referred to by the numbers in the color key throughout the figure. The RMSD analysis of both pockets includes the backbone and the sidechain atoms of residues F64, R66, L117, T129 from the α1 subunit and Y97, E155, S156, Y157, F200, T202, Y205 from the β2/β3 subunit (residue numberings are based on the native receptor from this work). Taken together, the superpositions and the RMSD analysis shows that the pockets, and the ECDs, adopt similar conformations. **b**, Superposition of individual subunits based on the transmembrane domain (TMD). Despite the general agreement among TMDs from these structures, the poses of the M3 helices are influenced by the substitution of the M3-M4 loop with a short peptide linker (6X3Z), as indicated by the angles formed by Cα atoms of three conserved residues that are shown below the structure overlay. Given that the M3 of the β subunit is part of the APG binding pocket, we focus the downstream structural comparison with the full-length structure bound with GABA only (6I53). **c**, Local conformational changes induced by APG binding at the β/α* pocket (left) and the β*/α pocket (right). Structural alignment is based on the β subunit TMD. The solid spheres are the centers of mass for the α1 subunits, and the axes shown are the longest axis from the inertia ellipsoid representations of the α1 subunits. The angles between these axes of unbound and APG-bound structures are measured and included in the figure.

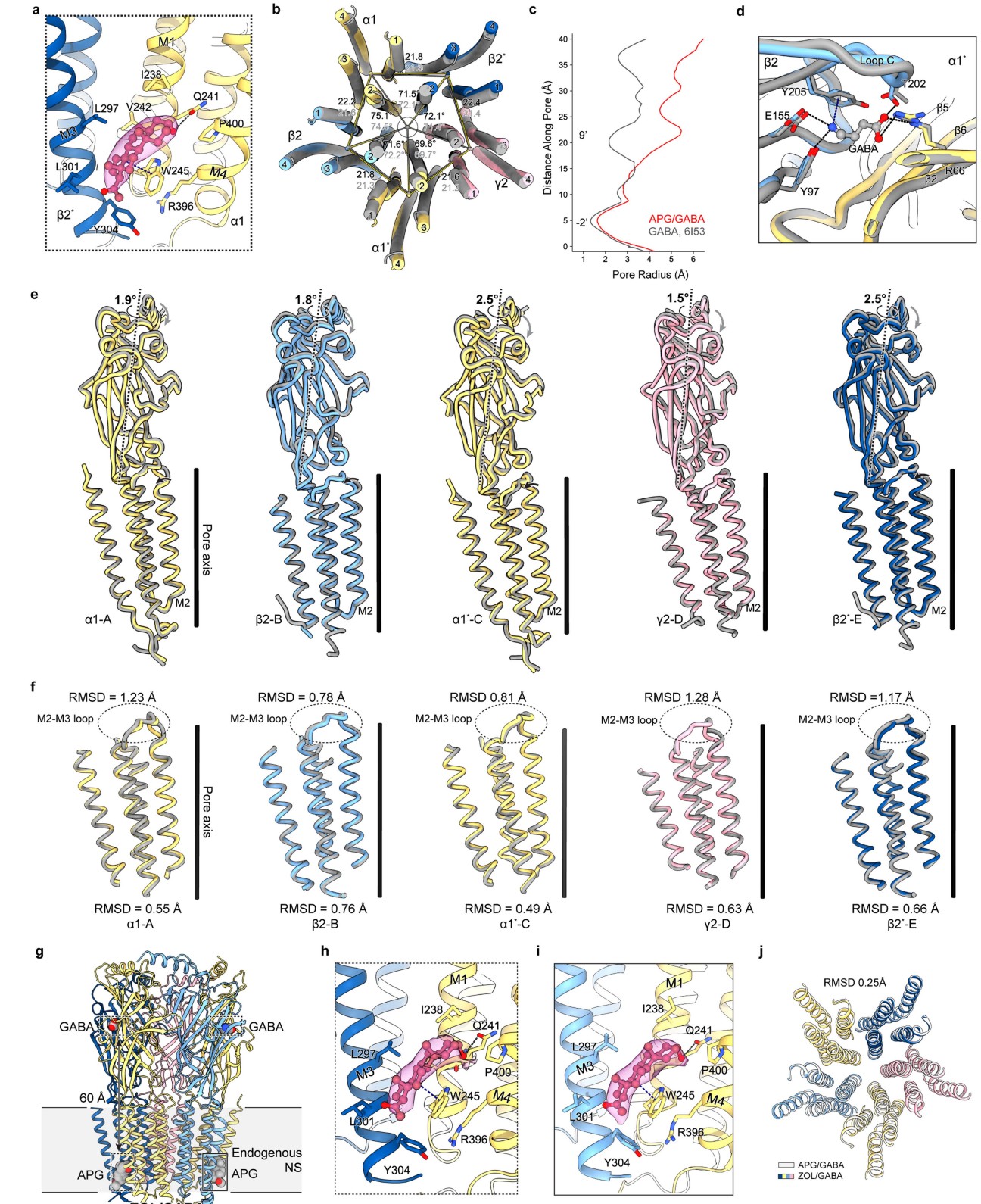

**Extended Data Fig. 9** | See next page for caption.

**Extended Data Fig. 9 | Neurosteroid binding to the nα1-GABA$_A$Rs.**
**a**, Allopregnanolone bound between the β2*⁺/α1⁻ interface of the two-Fab-APG. Cryo-EM density around APG is contoured at 6.6 σ. **b**, Structural comparison between two-Fab-APG and previous structure without APG (PDB code: 6I53, apo structure hereafter). The two structures are aligned based on the global TMD. Distances and angles formed with mass centers of TMD are also shown with those of the two-Fab-APG colored black. **c**, Comparison of pore profiles between two-Fab-APG and previous apo structure. **d**, Structural overlay of the GABA binding pocket from the two-Fab-APG and previous apo structure. The two structures are aligned based on the ECD domains of the adjacent β and α subunits. **e**, Comparison of each subunit between two-Fab-APG and previous apo structure based on global TMD structural alignment. **f**, Comparison of each TMD between two-Fab-APG and previous apo structure based on individual TMD structural alignment. RMSD values of the entire TMD domain and the M2-M3 loop are also shown. **g**, Structural overview of the two-Fab-ZOL. Two APG molecules are modeled based on the cryo-EM densities. **h** and **i**, Binding poses of APG in the two-Fab-ZOL structure. Cryo-EM density around APG is contoured at 5.1 σ. **j**, Structural overlay of the two-Fab-APG and two-Fab-ZOL based on the global TMD.

**Extended Data Table 1 | Statistics of cryo-EM data collection and analysis, and downstream model refinement and validation**

| | Native GABA$_A$R + DID (Endogenous GABA) DID | Native GABA$_A$R + GABA + ZOL ZOL/GABA | Native GABA$_A$R + GABA + APG APG/GABA |
|---|---|---|---|
| **Data collection and processing** | | | |
| Microscope | PNCC Krios | PNCC Krios | Janelia Krios |
| Electron Gun | XFEG | XFEG | XFEG |
| Voltage (kV) | 300 | 300 | 300 |
| Energy filter slit width (eV) | 20 | 20 | 20 |
| Detector | K3 | K3 | K3 |
| Operation mode | non-CDS | non-CDS | CDS |
| Flux on detector (e$^-$/pix/sec) | 16 | 15 | 7 |
| Total electron exposure on sample (e$^-$/Å$^2$) | 50 | 60 | 50 |
| Number of movie frames | 40 | 40 | 40 |
| Magnification | 105K | 105K | 105K |
| Pixel size (Å) | 0.826 | 0.826 | 0.831 |
| Targeted defocus range (μm) | -1.2 to -2.5 | -0.8 to -2.1 | -0.8 to -2.1 |
| Number of collected movies | 22,035 | 33,580 | 27,726 |
| Symmetry imposed | C1 | C1 | C1 |
| Initial particle images (no.) | 10,161,839 | 9,147,782 | 12,522,410 |
| GABA$_A$R particle images (no.) | 4,101,438 | 6,888,783 | 11,031,705 |
| Final particle images (no.) | 260,779 (two-Fab) 55,501 (ortho-one-Fab) 28,617 (meta-one-Fab) | 124,019 (two-Fab) 91,747 (ortho-one-Fab) 87,268 (meta-one-Fab) | 299,655 (two-Fab) 137,566 (ortho-one-Fab) 172,304 (meta-one-Fab) |
| Map resolution (Å) FSC=0.143 | 3.06 (two-Fab) 3.52 (ortho-one-Fab) 4.05 (meta-one-Fab) | 2.67 (two-Fab) 2.89 (ortho-one-Fab) 2.94 (meta-one-Fab) | 2.48 (two-Fab) 2.64 (ortho-one-Fab) 2.56 (meta-one-Fab) |

| | Two-Fab + DID (Endogenous GABA) | Two-Fab + GABA + ZOL (Endogenous neurosteroid) | Ortho-one-Fab + GABA + ZOL (Endogenous neurosteroid) | Meta-one-Fab + GABA + ZOL (Endogenous neurosteroid) | Two-Fab + GABA + APG | Ortho-one-Fab + GABA + APG | Meta-one-Fab + GABA + APG |
|---|---|---|---|---|---|---|---|
| **EMDB ID** | EMD-29728 | EMD-29727 | EMD-29743 | EMD-29742 | EMD-29350 | EMD-29741 | EMD-29733 |
| **PDB ID** | 8G4O | 8G4N | 8G5H | 8G5G | 8FOI | 8G5F | 8G4X |
| Model resolution (Å) FSC=0.143 | 3.06 | 2.67 | 2.89 | 2.94 | 2.48 | 2.64 | 2.56 |
| Map sharpening *B* factor (Å$^2$) | 0 | 0 | 0 | 0 | 0 | 0 | 0 |
| **Model composition** | | | | | | | |
| Non-hydrogen atoms | 17,355 | 18,311 | 15,876 | 15,861 | 18,288 | 15,753 | 15,899 |
| Protein residues | 2,094 | 2,120 | 1,887 | 1,887 | 2,120 | 1,888 | 1,895 |
| Glycans (molecules) | 441 (36) | 441 (36) | 430 (35) | 419 (34) | 441 (36) | 294 (23) | 386 (31) |
| Ligands (molecules) | 37 (3) | 83 (5) | 60 (4) | 60 (4) | 60 (4) | 60 (4) | 60 (4) |
| Lipids (molecules) | - | 694 (41) | 94 (2) | 94 (2) | 694 (41) | 94 (2) | 94 (2) |
| **B factors (Å$^2$)** | | | | | | | |
| Protein | 156 | 125 | 127 | 149 | 115 | 134 | 127 |
| Glycans | 212 | 171 | 179 | 204 | 165 | 170 | 179 |
| Ligands | 114 | 115 | 116 | 134 | 111 | 158 | 116 |
| Lipids | - | 144 | 182 | 205 | 146 | 199 | 172 |
| **R.m.s. deviations** | | | | | | | |
| Bond lengths (Å) | 0.003 | 0.003 | 0.005 | 0.006 | 0.004 | 0.003 | 0.004 |
| Bond angles (°) | 0.526 | 0.560 | 0.999 | 0.764 | 0.627 | 0.676 | 0.683 |
| **Validation** | | | | | | | |
| MolProbity score | 1.92 | 1.65 | 1.87 | 2.27 | 1.50 | 1.71 | 1.78 |
| Clashscore | 5.82 | 5.76 | 6.22 | 8.39 | 3.75 | 4.95 | 5.10 |
| Poor rotamers (%) | 3.45 | 2.24 | 2.49 | 6.23 | 1.97 | 2.37 | 2.95 |
| **Ramachandran plot** | | | | | | | |
| Favored (%) | 96.81 | 97.66 | 96.51 | 96.67 | 97.47 | 97.05 | 97.17 |
| Allowed (%) | 3.19 | 2.34 | 3.49 | 3.33 | 2.53 | 2.95 | 2.83 |
| Disallowed (%) | 0.00 | 0.00 | 0.00 | 0.00 | 0.00 | 0.00 | 0.00 |

# Reporting Summary

## Statistics

For all statistical analyses, confirm that the following items are present in the figure legend, table legend, main text, or Methods section.

| n/a | Confirmed | |
|---|---|---|
| ☐ | ☒ | The exact sample size (*n*) for each experimental group/condition, given as a discrete number and unit of measurement |
| ☐ | ☒ | A statement on whether measurements were taken from distinct samples or whether the same sample was measured repeatedly |
| ☒ | ☐ | The statistical test(s) used AND whether they are one- or two-sided *Only common tests should be described solely by name; describe more complex techniques in the Methods section.* |
| ☒ | ☐ | A description of all covariates tested |
| ☒ | ☐ | A description of any assumptions or corrections, such as tests of normality and adjustment for multiple comparisons |
| ☐ | ☒ | A full description of the statistical parameters including central tendency (e.g. means) or other basic estimates (e.g. regression coefficient) AND variation (e.g. standard deviation) or associated estimates of uncertainty (e.g. confidence intervals) |
| ☒ | ☐ | For null hypothesis testing, the test statistic (e.g. *F*, *t*, *r*) with confidence intervals, effect sizes, degrees of freedom and *P* value noted *Give P values as exact values whenever suitable.* |
| ☒ | ☐ | For Bayesian analysis, information on the choice of priors and Markov chain Monte Carlo settings |
| ☒ | ☐ | For hierarchical and complex designs, identification of the appropriate level for tests and full reporting of outcomes |
| ☒ | ☐ | Estimates of effect sizes (e.g. Cohen's *d*, Pearson's *r*), indicating how they were calculated |

*Our web collection on statistics for biologists contains articles on many of the points above.*

## Software and code

Policy information about availability of computer code

| Data collection | SerialEM 3.8, MicroBeta workstation v 4.0, Leica Application Suite X v3.7.4, Labsolutions v5.11, |
|---|---|
| Data analysis | RELION-3.1, cryosparc v3.3.1, Prism-9 v9.5.1, Coot v0.9-pre, Phenix v1.20.1, Chimera v1.16, ChimeraX v1.4, Python v3.8.13, locScale v0.1 (Github commit fe5d7e7), deepEMhancer (Github commit 2817c29), TIRF photobleaching analysis script (https://doi.org/10.5281/zenodo.8161179) |

For manuscripts utilizing custom algorithms or software that are central to the research but not yet described in published literature, software must be made available to editors and reviewers. We strongly encourage code deposition in a community repository (e.g. GitHub). See the Nature Portfolio guidelines for submitting code & software for further information.

## Data

Policy information about availability of data

All manuscripts must include a data availability statement. This statement should provide the following information, where applicable:
- Accession codes, unique identifiers, or web links for publicly available datasets
- A description of any restrictions on data availability
- For clinical datasets or third party data, please ensure that the statement adheres to our policy

The cryo-EM maps and coordinates for the native GABA receptor in complex with didesethylflurazepam and endogenous GABA (two-Fab-DID) have been deposited in the Electron Microscopy Data Bank (EMDB) under accession number EMD-29728 and in the Protein Data Bank (PDB) under accession code 8G4O. The cryo-EM

maps and coordinates for the native GABA receptor in complex with zolpidem, GABA, and endogenous neurosteroids have been deposited and accessed via EMD-39727/8G4N (two-Fab-ZOL), EMD-29743/8G5H (ortho-one-Fab-ZOL), EMD-29742/8G5G (meta-one-Fab-ZOL). The cryo-EM maps and coordinates for the native GABA receptor in complex with GABA, and allopregnanolone have been deposited and accessed via EMD-29350/8FOI (two-Fab-ALP), EMD-29741/8G5F (ortho-one-Fab-ALP), EMD-29733/8G4X (meta-one-Fab-ALP).

## Human research participants

Policy information about studies involving human research participants and Sex and Gender in Research.

| | |
|---|---|
| Reporting on sex and gender | N/A |
| Population characteristics | N/A |
| Recruitment | N/A |
| Ethics oversight | N/A |

Note that full information on the approval of the study protocol must also be provided in the manuscript.

# Field-specific reporting

Please select the one below that is the best fit for your research. If you are not sure, read the appropriate sections before making your selection.

☒ Life sciences          ☐ Behavioural & social sciences          ☐ Ecological, evolutionary & environmental sciences

For a reference copy of the document with all sections, see nature.com/documents/nr-reporting-summary-flat.pdf

# Life sciences study design

All studies must disclose on these points even when the disclosure is negative.

| | |
|---|---|
| Sample size | Sample sizes of cryo-EM data were determined by the anticipation of particle numbers within the constraints of microscope availability. No calculation of sample size was performed beforehand. We opted for a sample size of three, because it was done previously and we thought it is a good compromise between statistics and practicality. Single-molecule photobleaching experiments were carried out by collecting at least three photobleaching movies from different sample chambers. Three replicates of each condition were used for the radio-ligand binding assay. |
| Data exclusions | No data were excluded from the analyses. |
| Replication | Cryo-EM related biochemical experiments, including protein purification, SDS-PAGE analysis, and Western blot were repeated at least three times. Despite some degree of variability in protein yield, the pattern of proteins observed in SDS-PAGE and the detection of proteins with Western blot were reproducible. Radio ligand binding assay were repeated at least twice with independent samples on different days. This assay consistently yielded comparable binding affinities, reaffirming its reproducibility. The single-molecule photobleaching control experiment with isolated 8E3-Fab were repeated with two independent samples on different days with success. The single-molecule photobleaching experiments with 8E3-Fab bound GABAA receptors was conducted once. Because we obtained consistent results from the three parallel trials, which consisted of ~150 photobleaching traces each, no replication attempts have been made for this particular experiment. Finally, mass spectrometry analysis was conducted on independent samples using both the mass spectrometry core facilities at OHSU and LSU. Despite the difference in instrumentation, the identified proteins were largely consistent. |
| Randomization | For the single-molecule photobleaching experiments, photobleaching movies were acquired at random regions of the sample chamber. We did not employ randomization for sample or organism allocation across our range of other experiments, which include protein purification, gel analysis, ligand binding assay, cryo-EM data collection and analysis, as well as mass spectrometry analysis. This approach is primarily due to the fact that these specific types of experiments do not necessitate randomization or control for covariates, as they are not influenced by the sample allocation process common in other research methods. The only potential exception to this is our cryo-EM data collection where we selected squares based on subjective criteria of ice thickness and particle distribution, which can be argued as a form of non-random sample selection. However, it is important to note that this selection process is widely accepted in the field because it is a crucial step to ensure usable, high-quality data. |
| Blinding | The investigators were not blinded. In the context of our research, blinding during data collection/analysis was not feasible or relevant for a number of reasons. Specifically, for cryo-EM analysis, the goal is to describe the structure of the sample and the sample details should be given as a prior. As for the mass spectrometry analysis, single-molecule experiments, ligand binding assays, while it is technically feasible to conduct with the investigators blinded, the practicality is questionable. Introducing blinding would increase the labor cost substantially, without significantly enhance the validity of the results, as the downstream protein detection, determination of photobleaching steps, or curve fitting are reasonably streamlined. |

# Reporting for specific materials, systems and methods

We require information from authors about some types of materials, experimental systems and methods used in many studies. Here, indicate whether each material, system or method listed is relevant to your study. If you are not sure if a list item applies to your research, read the appropriate section before selecting a response.

## Materials & experimental systems

| n/a | Involved in the study |
|-----|----------------------|
| ☐ | ☒ Antibodies |
| ☐ | ☒ Eukaryotic cell lines |
| ☒ | ☐ Palaeontology and archaeology |
| ☐ | ☒ Animals and other organisms |
| ☒ | ☐ Clinical data |
| ☒ | ☐ Dual use research of concern |

## Methods

| n/a | Involved in the study |
|-----|----------------------|
| ☒ | ☐ ChIP-seq |
| ☒ | ☐ Flow cytometry |
| ☒ | ☐ MRI-based neuroimaging |

## Antibodies

**Antibodies used**

In house antibody: 8E3-GFP Fab anti-GBRA1 (produced by our lab), biotinylated anti-GFP nanobody (plasmid of the GFP nanobody was a gift from Brett Collins, and the nanobody was expressed, purified and biotinylated in our lab)
Commercial antibodies: anti-GBRA1 (Millipore, 06–868), anti-NL2 (Synaptic Systems, 129 202).

**Validation**

Validation of 8E3 mAb and its papain-digested fragment can be found in the previous paper from our lab (Phulera, S., Zhu, H., Yu, J., Claxton, D. P., Yoder, N., Yoshioka, C. & Gouaux, E. Cryo-EM structure of the benzodiazepine-sensitive α1β1γ2S tri-heteromeric GABAA receptor in complex with GABA. eLife 7, e39383, (2018))
Validation of the recombinant 8E3-GFP can be found in this study in the Extended Data Figure 1.
Validation of the anti-GFP nanobody can be found in the publication (Kubala, M. H., Kovtun, O., Alexandrov, K. & Collins, B. M. Structural and thermodynamic analysis of the GFP:GFP-nanobody complex. Protein Sci. 19, 2389-2401, (2010)). The biotinylation of the anti-GFP nanobody was carried out according to the manufacturer instructions (ThermoFisher A39259) and not validated.
The validation of commercial antibodies for western blot can be found in vendors' websites: https://www.emdmillipore.com/US/en/product/Anti-GABAA-Receptor-1-Antibody,MM_NF-06-868?ReferrerURL=https%3A%2F%2Fwww.google.com%2F (anti-GBRA1), https://sysy.com/product/129202 (anti-NL2).

## Eukaryotic cell lines

Policy information about cell lines and Sex and Gender in Research

**Cell line source(s)**

Sf9 cells for generation of baculovirus and expression of recombinant antibody fragment are from Thermo Fisher (12659017, lot 421973).

**Authentication**

The cells were not authenticated experimentally for these studies.

**Mycoplasma contamination**

The cells were tested negative for mycoplasma contamination using the CELLshipper Mycoplasma Detection Kit M-100 from Bionique.

**Commonly misidentified lines**
(See ICLAC register)

No commonly misidentified lines

## Animals and other research organisms

Policy information about studies involving animals; ARRIVE guidelines recommended for reporting animal research, and Sex and Gender in Research

**Laboratory animals**

One-month-old C57BL/6 mice (both male and female) were order from Charles River Laboratories. The housing conditions were set as: temperature 68–72 F, humidity 40–60%, dark/light cycle 12:12 hours. No experimental manipulations were performed on these animals.

**Wild animals**

This study did not involve wild animals.

**Reporting on sex**

Mice of equally mixed genders were used for the isolation of native GABAA receptors.

**Field-collected samples**

This study did not use any field-collected samples.

**Ethics oversight**

All mice were euthanized under the OHSU Institutional Animal Care and Use Committe (IACUC) protocol — TR03_IP00000905, consistent with the recommendations of the panel on euthanasia of the American Veterinary Medical Association (AVMA) carried out only by members of Dr. Gouaux's lab approved on the IACUC protocol.

Note that full information on the approval of the study protocol must also be provided in the manuscript.

