## [Peer Review File · Nature]

Manuscript Title: Cryo-EM structures reveal native GABAA receptor assemblies and pharmacology

Reviewer Comments & Author Rebuttals

Reviewer Reports on the Initial Version:

Referees' comments:

Referee #1 (Remarks to the Author):

This paper from Sun et al presents the structural characterization of benzodiazepine and neurosteroid modulation of native GABAA receptors. GABAA receptors are ion channels expressed in the nervous system. They are modulated by a large number of molecules including some used in therapeutics such as flurazepam, zolpidem and allopregnanolone, a lipophilic neurosteroid. In the study, the authors prepared and purified native $\alpha 1$ -containing GABAA receptors (na1GABARs) from mouse brain extracts and solve their atomic structures using cryo-EM in the presence of an analog of flurazepam (DID), zolpidem (ZOL) or allopregnanolone (ALP). Despite the native origin of the sample, the cryo-EM work is rich and insightful with the resolution of three main conformations in each of the three datasets with resolution ranging from 2.5 to 4 Å. The experimental plan is highly original and brilliantly conducted: the sample preparation is outstanding and the strategies used in data processing and model building, while clearly described, are technically remarkable.

A first part of the paper is dedicated to the composition of na1GABARs that are extracted from tissue samples as compared to the consensus found in the literature obtained by pull-downs or recombinant expression of GABAR. Here, the authors can identify the number of $\alpha 1$ subunits they find in their pentamers on the purified sample using a Fab fragment specific to this subunit and photobleaching experiments. Surprisingly, they find a higher number of assemblies containing a single $\alpha 1$ subunit than previously expected (~50% in photo-bleaching experiment), the rest of the pentamers presenting two $\alpha 1$ subunits. This number compares well with the proportions found in the data collected in cryo-EM and determined early in data processing by the alignment of the Fab fragments.

To determine the precise composition of the na1GABAR, the authors further discriminate between assemblies at the 3D classification steps using the Fab to locate $\alpha 1$ and using the glycosylation features to differentiate a, b and g subunits. They this way determine that their sample is mainly composed of 2a/2b/1g pentamers, but never with a b-b interface as it was described in recombinant systems. They distinguish three populations in all three datasets: a GABAR with two $\alpha 1$ ("two-Fab"), one with one $\alpha 1$ next to the $\alpha 2$ ("ortho-one-Fab") and one with one $\alpha 1$ in meta- of the $\alpha 2$ ("meta-one-Fab"). They then use the highest resolution dataset (ALP) to finalize model building and identify the subunits, using again the glycosylation patterns but also the density of the side chains, even though in some instances they cannot differentiate between highly similar subunits ($\alpha 2/\alpha 3/\alpha 5$ or b1/b2), a precaution that indicates the authors do not over-interpret the densities. Finally, they analyze the conformational differences associated with this differential subunit composition.

In the second part of the paper, the authors discuss the binding of ALP, a lipophilic neurosteroid used in the treatment of postpartum depression and that potentiates GABA-response of GABAARs. In all three sub-populations, two ALP molecules are found at a b/a interface in the inner leaflet part of the transmembrane domain. Interestingly, the binding pocket is partly lined by lipids, which possibly participates to the mode of action of ALP by changing its association/dissociation kinetics. Binding of ALP triggers a conformational change as compared to previously published GABA-bound structure, with a tilt of M2 helices enlarging the ionic pore and a rotation in the ECD

both compatible with allosteric potentiation.

In a final part, the article focuses on the binding of DID and ZOL on the $\alpha 1$ GABARs, with details of the binding pockets and comparison with a recently published structure of recombinant GABAR in complex with ZOL. The authors notably describe the presence of a density attributed to endogenous ALP co-purified with the receptors and responsible for the absence of binding of ZOL to the transmembrane domain. Interestingly, and contrary to ZOL and ALP-datasets, despite the fact that they solved the DID-structures without adding GABA to the sample, they still observe an apparent partially open-pore conformation of the receptor when DID is bound, consistent with the agonist-like properties of benzodiazepines at high concentrations.

Overall, the data are very original and important for the description of the mode of action of highly prescribed drugs in humans. Furthermore, the effort done by the authors to obtain samples as close as possible from native proteins is rewarding with the observation of native GABAR stoichiometries that differs from the paradigm obtained from recombinant systems data.

The work is rich and appears to have been performed rigorously, I do not have any major comments or concerns. Yet, we have some questions or comments that could improve the clarity and impact of the work:

- Title: Even if the authors are very close to in vivo conditions, I would nuance the title of the article
- Methods:
 - o The data processing section is rather clear, but it could be easier for the reader to know which dataset were processed with strategy 1 and which were with strategy 2
 - o line 740 : in what aspect the setup to prepare grids was "specific" and for what purpose ?
- Model building section: for the precise subunit identification, it is not clear whether the authors systematically looked at all the positions that could be indicative of the identity of the subunit and could not discriminate because of lack of clear density or whether they think they have a mixture of populations creating hybrid densities. A supplementary element (Figure or table) describing all the positions that were investigated and their possible assignment would be insightful.
- There is no legend description for Extended Data Figures 2-4 and 6
- line 110-114 Could the absence of a b-b interface stoichiometry in the dataset could be due to the elimination of particles during early processing (i.e. those with bad TMD signal in strategy 1 or those with bad Fab signal in strategy 2) ? if uncertain, I will recommend to mitigate the conclusion that b-b interface stoichiometry is marginal in the brain.
- A figure showing the glycosylation described in lines 100-105 would be helpful to understand how the authors used them to identify subunits
- Rearrangements described in lines 148-155 will be best visualized in a figure / table summarizing the measures.
- line 165-171. A figure showing the densities (with a contour level) in which the lipids were modelled would be helpful. Overall, it would be good to legend all the densities with their contour level to help the reader assess the strength of the signal coming from the ligands.
- Similarly, in lines 233-238, the ALP assignment in the ZOL/GABA dataset should be better described with a separate supplementary figure to be more convincing.
- lines 290-296 and extended data tables: the tables described the DID dataset as containing "endogenous GABA" which is puzzling with the last paragraph of the results that described a potentially conductive state in absence of GABA. Did the authors found density for GABA in the GABA-binding pockets in this dataset ? This needs to be clarified.

Referee #2 (Remarks to the Author):

This article presents for the first-time high-resolution structures, 2.5-2.6 Å, of GABAA receptors from native mammalian tissue using single-particle cryo-EM, a truly heroic feat. As a result, the authors have been able to provide answers to a number of key questions that have lingered in the

field for decades. Understanding the structures underlying both function and pharmacology of GABAARs is highly significant as it is the main inhibitory receptor in the brain and, is of immense clinical importance since many pharmaceuticals used to reduce activity in the nervous system work at the GABAARs.

The authors isolated the receptors with alpha1 subunit selective monoclonal Fab fragment that they had previously proven did not affect the structure of recombinant GABAARs. They then were able to count the number of alpha1 subunit in the structures isolated from brains of mice (50 one month old mice per preparation, enormous task!) using single-molecule fluorescence bleaching experiments and then, assessed protein composition by mass spectrometry.

Amazingly, despite 19 GABAAR subunits potentially available to form pentameric channels in neurons, only three major types were identified, based on location and number of alpha1 subunit in the receptors and then glycosylation patterns of other subunit-types, much fewer receptor-types than are commonly assumed to exist! In addition, of the alpha1 GABAAR complexes, ca 50% contained two alpha1s but the rest contained alpha1 together with alpha2, 3 or 5, another surprise for many in the field and with important physiological and pharmacological consequences, highly relevant for drug design efforts.

Despite the very high homology between alpha subunits, the structures containing the mixed alphas were more compact than those with two alpha1-subunit in the receptors. Again, this was unexpected but with potential consequences for domains interactions and thus function/pharmacology.

Although genes expressed in recombinant eucaryotic cell-based systems or bacterial systems are expected to faithfully represent the ion channel structure across species, this study demonstrates that the GABAAR isolated from native mouse brain incorporates features lacking in other systems, namely incorporating the neurosteroid ALP in the structure that then significantly alters conformations of the receptor complex. Native ALP was detected in the ZOL/GABA dataset where no ALP had been added and the TMD structures were almost identical. These results identify perhaps the most significant function of neurosteroids, namely modification of the ion channel pore/function and the mechanism of "native neuronal" allosteric potentiation of the GABAAR that notable is absent in other tissues/species/expression systems that may express GABAAR but do not produce neurosteroids.

The authors also identified and described molecular structure and location of two ALP binding sites in the TMD and the allosteric mechanism underlying potentiation of the GABAAR function. The residues involved in binding ALP are conserved in all alpha and beta subunits and some have been identified by structure-function and mutagenesis analysis before. Here they identified how the slight difference in the binding sites could explain difference in the affinity for ALP. Their data clearly demonstrates the specificity of ALP potentiation of the receptor allowing rejection of the idea that neurosteroids interaction is nonspecific and due to perturbation of the lipid bilayer. Further, the authors can explain and resolve the debate on the nature of the slow on/off rate of ALP which they show is due to the nature of the binding pockets which lipids participate in forming and where the lipids regulate the entry/exit of ALP.

The authors continue to map the binding pockets of ZOL and can explain how observed ZOL difference in GABAAR potency alpha1 > alpha2-3 > alpha5 GABAAR comes about based on whether or not hydrogen bonds are formed between ZOL and alpha amino acids or not. Importantly, the authors also highlight the difference between their native structure with ZOL and a recently published recombinant structure where the third binding site for ZOL was observed but one that probably does not exist in native receptors in brain tissue!

And finally, an important discovery where the authors capture the receptor complex in the open non-desensitized conformation, not only with the 9'L gate open but also the 2' gate open and the

dimensions are such that a dehydrated Cl⁻ can pass through. Is the 2' gate, a selectivity filter in the GABAAR in the non-desensitized receptor?

This is a seminal article that describes for the first time, GABAAR structures isolated from mammalian brains and their general and specific characteristics.

An outstanding paper presenting novel, important new data that address a number of fundamental gaps in the knowledge of how structure relates to function and pharmacology of native neuronal GABAARs. The methodology is robust and the use of statistics appropriate. The paper is clearly written and appropriate credit is given to previous work

No additional experiments are required and the Figures are generally of high quality.

Minor comments:

Line 76, a word appears missing: ... enable release from the affinity resin..., delete "it"

Line 582, A word appears missing: ... obtained from one mouse brain preparation (that includes brains from 50 mice)

Figure 4, labelling "neurosteroid" in TMD is unclear

Extended Figure 2,3,4, labelling of subunits is very difficult to read – perhaps having the labelling "in bold" might make them clearer.

Extended Figure 2, labelling of subunits in ortho-one-Fab appears wrong with two betas next to one another.

Referee #3 (Remarks to the Author):

A: Sun and colleagues from the Gouaux lab present the first cryo- EM structures of alpha1- GABA-A receptor populations that were isolated from native mouse brain – a true breakthrough. They studied a fraction of these receptors, in complex with relevant drugs.

B: Highly significant, novel work.

C: approach valid, data great, quality of presentation can be improved as detailed below

D: statistics appear fine. Uncertainties concerning the precise determination of major and small pools of the receptors in the workflow are not discussed well.

E: Apart from missing conclusions on the mass spec data (see below), the conclusions are robust, valid and reliable.

F: (suggested improvements) The data presentation and discussion has some weaknesses that need to be addressed in order for the data to be as valuable as it could be: Very interesting conclusions about expected receptor assemblies were possible, but were not followed through the data in a consistent manner. No reasoning is offered why e.g. the delta- containing population was not studied further. Binding modes of drugs are covered exhaustively (which is fine), but the data dealing with beta3, gamma1, delta, alpha4 and alpha6 containing pentamers is not reflected in the manuscript at all, a huge loss. This major point and several minor points are discussed in detail below in order to guide the revision process – thorough rewriting will fully address most points, with the exception of a suggested additional radioligand experiment that might help to further quantify receptor population sizes. Details below.

G: Key papers dealing with receptors that contain two different alpha isoforms in a single pentamers are completely missing, this must be corrected.

H: As indicated in the detailed points below, the clarity of the data presentation and the abstract can be improved considerably.

Detailed input:

Major points:

The authors state already in the abstract that receptors with two alpha isoforms in the pentamer, one of which is alpha1, was "unexpected" for them – this is NOT at all the case. Many groups have

indicated, as early as in the 1990-ies, that two different alpha isoforms can co- assemble in pentamers. This is less prominent for $\alpha 1$ - containing ones. An estimation based on various knock in mice by the Möhler/ Rudolph work (maybe Benke et al) found that at least 20% of alpha1- containing pentamers contains a second alpha isoform. For other alpha isoforms, it is quite clear that they co-assemble with $\alpha 1$ – see e.g. Bencsits 1999, Scholze et al. 2022 PMID: 36279694 and many works by other groups. A re-balanced view of the past findings is urgently needed here.

Lines 67-68: It is incorrect to state that ALP, DID and ZOL “target $\alpha 1$ - containing receptors”. DID is likely as unselective as its parent flurazepam, ALP is an unselective steroid, and ZOL targets the zolpidem sensitive $\alpha 1$, $\alpha 2$, $\alpha 3$ / $\gamma 2$ sites with high affinity. The statement is at best misleading, or can be read as wrong.

There is a major discrepancy between statements in the conclusion, and the contents in lines 87-89: “ analysis of the purified native receptor complexes by mass spectrometry identified all α and β subunits as well as the $\gamma 1$, $\gamma 2$, and δ subunits, demonstrating that $\alpha 1$ -dependent isolation captured receptors containing most of the 19 GABA A R subunits.” – in contrast – “ $\alpha 1$ GABAA Rs comprise three structural populations: tri- heteromeric $\alpha 1 \beta 2 \gamma 2$ receptors that constitute half the total population and two distinct assemblies containing one $\alpha 1$ subunit and one $\alpha 2/3/5$ subunit.” There is no tabular overview summarizing the mass spec findings in the MS (only in the supporting Figure 1), and no explanation why only gamma2- containing receptor species were studied further. Please explain and add appropriate material to the MS. Supporting Figure 1 panel i summarizes the mass spec findings, and shows that the gamma1 plus delta- populations added together are larger than the gamma2- population. In order to fully credit previous work and to provide a balanced view onto $\alpha 1$ - containing receptor species, a table with compositions that are suggested by the mass spec data and by the literature, in addition to those that are presented with the cryo- EM data, would vastly improve the presentation and interpretation of the findings.

The reader cannot follow where the beta3 subunit was “lost” – in Figure 1, only beta1 and beta2 are mentioned. The mass spec data gives b1/b2/b3 to be 14/26/20 %. Did you find that gamma2- alpha1- containing pentamers never contain beta 3? This would be a remarkable finding, if I interpret Figure 1 well. If this is not what you imply, clear writing (and the above suggested table) would greatly help the reader to follow through your data.

The title indicates that the work deals with “regulated assembly”, and yet the majority of figures deals with drug binding modes and conformations. As stated above, much of the mass spec data was not interpreted, nor used to reach conclusions about assemblies with beta3, gamma1, delta or alpha4 subunits. Thus, it is suggested that the authors rebalance the data presentation in such a way, that all findings from the mass spec work are presented and properly interpreted in the MS, and reduce a bit of the detail on binding modes and conformational changes to make the most of the data in terms of assembly regulation and pools of alpha1- containing receptors.

Minor points:

IUPHAR recommendation discourages the use of subscripts for subunit isoforms – subscripts should be reserved for stoichiometry, thus, the authors are encouraged to use IUPHAR recommended terminology throughout.

Why was the cerebellum excluded from the analysis? Please provide the rationale for the study design.

Line 210: “disassociated” – should read dissociated?

Congrats to this fascinating piece of work, I hope you can appreciate my input as constructive –

looking forward to the final version and to citing the paper,
bw
margot ernst

Referee #4 (Remarks to the Author):

Review on the manuscript "Regulated assembly and neurosteroid modulation constrain GABAA receptor pharmacology in vivo"

The authors of this manuscript impressively describe the long-desired composition and stoichiometry of native GABAA receptors that contain the $\alpha 1$ subunit for the first time. They provide high resolution models (up to 2.5-2.6Å) of those receptors in the presence of various endogenous and therapeutic molecules by utilizing a collection of state-of-the-art techniques. This is the first time for cryo-EM structures that two conditions are met, namely receptors harboring an intracellular domain and the presence of physiological membrane lipids. This study demonstrates that the majority of receptors in the mouse brain contain $\alpha 1\beta 2\gamma 2$ subunits, in perfect agreement with the literature. Interestingly, half of the receptors from the pool of $\alpha 1$ -containing GABAA receptors that were resolved, contained an additional alpha subunit and one or two distinct beta isoforms – thus, complex stoichiometry with up to five different subunits in one pentamer. This structural evidence for non-classical native GABAA receptors will likely accelerate structure-based drug development. Furthermore, this work provides significant evidence for the prevalence of two neurosteroids at their "canonical" transmembrane sites ($\beta +/\alpha -$) at native receptors that might be held in place by lipids. Similarly, as demonstrated for the GABA binding sites in the ECD, the neurosteroid (NS) binding sites were not equivalent, despite same architecture. It is proposed that NS bound receptors shift protein conformation and therefore should be taken into consideration when interpreting results from heterologous expression systems, without physiological concentrations of neurosteroids present. There was no evidence for additional intra-subunit neurosteroid binding sites at native $\alpha 1$ -containing receptors, as suggested from previous studies combining photolabeling and mass spectrometry studies. This study also presents data for the binding of two hypnotics, zolpidem and a flurazepam metabolite, further expanding our understanding of the physiological and pharmacological role of GABAA receptors. The authors claim that the DID-flurazepam bound structure without GABA might capture the first GABAA receptor in a conductive state that is published to this reviewer's knowledge in the cryo-EM field. In summary, this work provides novel insights into the assembly, stoichiometry, and function of native GABAA receptors and some of their modulators, interesting for a broad range of audience. Therefore, this reviewer advocates for the acceptance of a substantially revised version of the manuscript, if all the reviewers' remarks are addressed carefully, without additional wet-lab work. Thus, this reviewer thinks that the manuscript quality can benefit greatly from substantial revision of the text and figures.

Minor comments

- 1) Line 43 "Ref.5" is put in parenthesis - please cite the paper according to the style of the remaining.
- 2) Line 49: The authors write "In addition, ganaxalone, a synthetic mimetic of allopregnanolone..." - Please indicate, that ganaxalone is a synthetic derivative of allopregnanolone (3 β -methylated ALP). The synthetic version of allopregnanolone is called brexanolone and the FDA approved treatment for postpartum depression since 2019. However, that formulation additionally includes cyclodextrines for solubility reasons. In 2022 the FDA approved ganaxalone for the rare condition of CDKL-5 deficiency dependent seizures.
- 3) Line 76: please rephrase "release it the affinity resin"
- 4) Lines 95-97: Similar observations have been made with cerebellar $\alpha 1/\alpha 6$ -containing receptors, e.g. (Scholze et al., 2020)

5) Line 161: To this reviewer's knowledge, the effects of neurosteroids are not selective to GABA A receptors, thus please rephrase or delete that statement accordingly.

6) Line 164: The text refers to Figure 2B, not to Figure 2A.

7) Line 166-169: Please acknowledge the previous work done: (Chen et al., 2019; Lavery et al., 2017; Sugawara et al., 2019, 2020).

8) Line 198-201: Do the authors have any theory on how the same molecule (ALP) could open and modulate the receptors simultaneously via the same binding sites? On a single receptor level, when the exact same binding sites are involved into agonistic and allosteric effects of ALP, and both of its binding sites are already occupied with ALP molecules at low concentrations (allosteric effects), how can the receptor even sense higher concentrations of ALP and react differently (agonistic effect) to it? Additionally in the introduction the authors claim agonistic effects above 100nM of ALP, whereas the HPLC demonstrates >300nM ALP concentration in their preparation. However, only allosteric effects are observed. How could that be explained? Could additional unresolved sites exist within the TMD/ECD interface or within the ICD?

9) Line 246: See major comment 2) on the main metabolite of flurazepam.

10) Line 258: Which imidazole nitrogen is meant?

11) Line 308: Classifying GABAA receptors based on their alpha subunits only is a new concept to this reviewer, please clarify. The community used to seek for specific modulators that would spare certain α -containing receptor populations and be effective at others to generate molecules with improved pharmacological profiles (such as the search for anxi-selective drugs which spare $\alpha 1$ -containing receptors (meant were $\alpha 1\beta\gamma 2$ receptors) while remaining activity at $\alpha 2$ -containing receptors - a concept that turned out not to translate into humans, see Skolnick 2012. However, entire receptors have not been categorized according to their α -subunits solely. Receptor classification has been based on subunit composition (e.g. $\alpha 1\beta 2\gamma 2$) and interfaces they included, e.g high affinity benzo sites have been classified based on the respective α and γ - subunit that form it.

12) Line 310: For clarity reasons, it should be mentioned again, that allopregnanolone is the most abundant endogenous neurosteroid, not exclusively a medication for postpartum depression.

13) ALP is often referred to as high affinity benzo site ligand alprazolam, which can be confusing. AloP is commonly used in the literature to refer to allopregnanolone.

Minor comments to figures:

Figure 1:

- Asterisks should be explained in figure legends again for clarity.
- The legend should also include the clockwise orientation for intuitive interpretation for an expert reader (who usually sees subunit arrangement given in counterclockwise manner)
- It should be stated in the figure legend that cerebellum was excluded.
- Please comment on $n=2$ for the cryo-EM experiments in Panel 2B

Figure 2:

- The bottom 2D-chemical structure of ALP (allopregnanolone) needs rework and additional explanation of the color code used in the figure legends. ALP is drawn with a 3 α -hydroxymethyl (black line indicating the existence of a carbon atom, red line indicating a hydroxy group), which is incorrect. The functional group at C20 is represented the same way as the one on position 3, however it is a carbonyl group, thus a double bond should be drawn here.
- Again, asterisks should be mentioned in the legend text.
- It should be stated in the figure legend for panel b that two-Fab receptors are shown for easier processing of the figure.

Figure 3:

- Ball and stick representation of the ligands makes the appreciation of interactions for both molecules extremely difficult. Different rendering should be used.

Major comments

1)

Which of the proposed splice variants of the α -subunit (short, long) was observed in the mass spectrometry studies and Cryo-EM experiments? This finding should also be discussed.

2)

The authors of the study assert that they observed native receptors in combination with three clinically relevant drugs: allopregnanolone, flurazepam, and diazepam. It should be noted however that flurazepam is a prodrug that quickly converts to active metabolites, with various studies identifying multiple metabolites in human venous blood, including monodesethyl-flurazepam, didesethyl-flurazepam, flurazepam-aldehyde, hydroxyethyl-flurazepam, and desalkyl-flurazepam. In contrast to the claim in line 246 about DID-flurazepam being the main flurazepam metabolite and the presence of DID after flurazepam administration, N-desalkyl-flurazepam is generally recognized as the primary active metabolite, with a half-life of up to 100 hours, which is consistent with increased flurazepam efficacy following repeated administration due to accumulation. N-desalkyl-flurazepam is also a metabolite of various other benzodiazepines, such as midazolam, quazepam, ethyl loflazepate and others, further increasing its relevance. The reasoning behind the use of DID-flurazepam (Didesethyl-flurazepam) in the study and the clinical relevance in comparison to other recognized flurazepam metabolites was not explained properly. Please rewrite for clarity. Thus, it is important to note that while flurazepam and its metabolites have clinical relevance, the clinical aspect of this finding should be presented in a balanced and appropriate manner, without overstating its significance and should be discussed in the manuscript for clarity.

3)

Lines 298-306: Kindly provide a summary of the authors' interpretation of their findings regarding the proposed receptor stoichiometries and arrangements for α 1-subunit containing assemblies, such as α 1- β 2- α 1- γ 2- β 2 / α 1- β x- α x- γ 2- β x / α x- β x- α 1- γ 2- β x. Why do the authors exclude other native α 1-containing receptors than those? How do their findings contradict the existence of β - β interfaces in α 1-containing populations, which is in seeming contradiction to the literature? In their mass spectrometry experiments, the authors claim to have identified all alpha subunits, including α 4 and α 6, all β -isoforms, including β 3, as well as γ 1 and δ in α 1-containing pentamers. However, none of these assemblies are discussed further. Please explain. Is it possible that β - β interfaces could exist in a small fraction of assemblies, but they could not be detected due to their low abundance? Can you really exclude α 1- β x- α x- β x- β x or α 1- β x- α x- δ - β x receptors (which are considered to be a minority of all α 1-containing receptors, see (Bencsits et al., 1999)). Similarly, why could models of receptors containing α 4/ α 6 γ 1, and δ not be built? Their existence appears to be strongly suggested by the presented mass spectrometry data. Which explanation for the absence of β 3-subunits in the cryo-EM experiments can be made, despite their high abundance in neuronal cells? Please discuss on the limitations of the method to resolve native receptors for a non-expert readership. The authors ought to address these questions and moderate their claims about the potential existence of α 1-containing native GABA_A receptors accordingly.

4)

Lines 159-201; Panel 2d,e; extended Figure 7 and 8: Despite the nice conformational change that is indicated due to ALP binding in Figure 2D ext. Fig 7 and 8, this reviewer sees open questions and discrepancies between the claims and the cited sources: Please explain why 6I53 was used as a comparison to the ALP bound structures in this work. To this reviewer's knowledge, 6I53 is not in complex with GABA (as stated) but only with a megabody38, thus the comparison completely lacks plausibility. Additionally, please state why 6X3Z was not used instead – since it is a GABA-bound structure of the same protein complex (a1b2g2 rather than a1b3g2). The conformational changes induced by ALP have to be compared to the existing GABA bound a1b2g2 structure 6X3Z, despite missing ICD. If differences are found, they should be discussed properly in the main text, which should be rewritten accordingly. Those questions need to be addressed carefully prior to publishing. Furthermore, the possible impact of different lipids, cell types and other conditions should be addressed as well. Additionally, how does binding of ALP interfere with the organization of both Apo-GABA binding sites locally (besides ECD rotation) and the high affinity zolpidem/DID-Flurazepam binding site within the pentamer in comparison to 6X3Z; are there any conclusions to draw from this?

Major comments to figures:

Figure 3 panel f

- The authors claim that the DID-bound structure is potentially conducting due to an overall pore radius above 1.81 Angstrom. However, GABA was omitted in the preparation and no density for neurosteroids could be resolved. An agonistic effect of a benzodiazepine has never been observed before profoundly to this reviewer's knowledge. Thus, if the authors claim the depicted conformation might reflect a conductive state, then please provide electrophysiological data with DID-flurazepam agonistic effects in absence of allopregnanolone.
- If neurosteroids were present in agonistic concentrations but could not be resolved (as given as possible explanation in the main text), how does this theory fit with the observations made in comparison of ALP/GABA vs zolpidem/GABA bound states from extended data figure 8j (0.25 Angstrom difference). Furthermore, if neurosteroids were present in that structure, would it be expected then, that DID-flurazepam binding is reminiscent of diazepam/alprazolam binding in earlier studies? Please reevaluate the proposed conformations of given structures based on other GABAA receptor publications (such as the histamine-bound 7QN9 from Sente et al., 2022, Aricescu lab, who claim a pre-open receptor conformation or binary ab receptor conformations as from Miller et al, 2022, and other a1b2g2 receptors such as 6X3X / 6X3U from Hibbs lab).

Figure 4

- Line 546: The effect of neurosteroids, including endogenous neurosteroids cannot be generalized that way. Endogenous neurosteroids with excitatory function and negative allosteric modulation at GABAA receptor subtypes are known, such as pregnenolone sulfate (PS) or dehydroepiandrosterone (DHEA). They certainly do not always increase Cl⁻ influx through GABAA receptors. Please rephrase accordingly also in the main text.

Author Rebuttals to Initial Comments:

REFeree #1

Comment. Title: Even if the authors are very close to in vivo conditions, I would nuance the title of the article

Reply. *We thank the reviewer for the suggestion. We have removed "in vivo" from the title and rephrased it to "Cryo-EM structures reveal native GABA_A receptor assemblies and pharmacology".*

Comment. Methods:

The data processing section is rather clear, but it could be easier for the reader to know which dataset were processed with strategy 1 and which were with strategy 2

Reply. *Strategy #1 was used for the DID dataset and strategy #2 was used for the ZOL/GABA and the APG/GABA (we changed the abbreviation of allopregnanolone from ALP to APG in response to the kind suggestion by another reviewer) datasets. We have added the clarification to line 739.*

Comment. line 740 : in what aspect the setup to prepare grids was "specific" and for what purpose ?

Reply. *In the early phase of the project, we extensively screened ligands and buffers to find the best grid preparation conditions. We employed the "specific" PCR tube setup to streamline the preparation of different sample conditions. We used 200 μ L PCR tube strips to which we added pre-aliquoted additives so that screening different conditions only involved transferring the minimal amount of protein sample, brief pipetting, and loading onto the grid. We have edited the methods section to clarify this process. At lines 709–711, we have changed "to prepare" to "streamline the preparation" and changed "PCR tubes" to "a strip of 200 μ L microcentrifuge tubes"*

Comment. Model building section: for the precise subunit identification, it is not clear whether the authors systematically looked at all the positions that could be indicative of the identity of the subunit and could not discriminate because of lack of clear density or whether they think they have a mixture of populations creating hybrid densities. A supplementary element (Figure or table) describing all the positions that were investigated and their possible assignment would be insightful.

Reply. *In the Methods section (lines 782–784, we mentioned that "we carefully examined residues where the sidechain can be unambiguously assigned and where there is a*

*difference of more than 3 carbon atoms or one sulfur atom between respective residues of the different receptor subunits.” We agree with the reviewer that the subunit identification from density maps needs more elaboration. To this end, we augmented the legend of **Extended Data Figure 6**, a figure which shows representative examples of how differences in density features guided the definition of subunit identity. We also prepared an additional figure of sequence alignment of key residues where N-glycosylation was used for subunit identification (**Supplementary Figure 1**). We have also included a thorough description in the **Supplementary Information** to help the readers follow our process of subunit identification.*

Comment. There is no legend description for Extended Data Figures 2-4 and 6

Reply. *We have added descriptions for these Extended Data Figures.*

At lines 852, 855, 858.

Refer to the “cryo-EM data analysis” method section for more details.

At line 869.

*Cryo-EM maps from the APG/GABA dataset were analyzed alongside sequences and atomic models to identify subunits that are not bound to the antibody fragment. The figure’s top section displays schematics of the three identified populations, with the five subunits labeled by chain IDs and color-coded. In each row, aligned sequence segments are presented on the left, with the distinguishing residues encased in boxes. Following this, cryo-EM densities contoured at the same level from these three populations (separated by a vertical bar) surround these residues are shown, with arrows pointing to the distinguishing residues. By comparing the models, cryo-EM densities and respective amino acid sequences, subunits can be assigned or excluded (detailed description in the **Supplementary Information**). The final row showcases the luminal N-linked glycosylations from the α subunits instead of amino acid residues, with arrows pointing to the positions of a fucose moiety (absent in $\alpha 1$ subunit) linked to the first N-acetylglucosamine of $\alpha 2/\alpha 3$ subunits.*

*We have also re-arranged **Extended Data Figure 6** to make it easier to read.*

Comment. line 110-114 Could the absence of a b-b interface stoichiometry in the dataset could be due to the elimination of particles during early processing (i.e. those with bad TMD signal in strategy 1 or those with bad Fab signal in strategy 2) ? if uncertain, I will recommend to mitigate the conclusion that b-b interface stoichiometry is marginal in the brain.

Reply. We appreciate this comment and agree that it is possible that some particles with a β - β interface could have been eliminated early in the processing. Nevertheless, we note that in both the zolpidem and the allopregnanolone datasets, all the 3D classes can be classified as α - β - α - β - γ , including the “bad Fab” classes. In addition, we carried out 3D classifications with masks focused at the glycosylation sites, and found no classes with a β - β interface (data not shown). While N-glycosylation features are smaller compared to Fab fragments, they are distinct across GABA_A receptor subunits and are visible at medium resolution (~ 8 Å). We thus estimate that our 3D classification should capture receptors with β - β interface if they are present with a $\sim 5\%$ or greater abundance.

Nevertheless, we have ‘toned down’ our comments related to receptors with a β - β interface, as recommended. At line 113, change “we found no evidence for receptors with a β - β interface” to “our cryo-EM analysis did not reveal receptors with a β - β interface”.

Comment. A figure showing the glycosylation described in lines 100-105 would be helpful to understand how the authors used them to identify subunits

Reply. The glycosylation densities are shown in **Extended Data Figure 6** and described in the accompanying figure legend. In addition, we have prepared a supplementary figure (**Supplementary Figure 1**) detailing the differential N-linked glycosylation sites across GABA_A receptor subunits ($\alpha/\beta/\gamma/\delta$).

Comment. Rearrangements described in lines 148-155 will be best visualized in a figure / table summarizing the measures.

Reply. We have prepared a figure (**Supplementary Figure 2**) to highlight the domain rearrangements described in lines 153–161.

Comment. line 165-171. A figure showing the densities (with a contour level) in which the lipids were modelled would be helpful. Overall, it would be good to legend all the densities with their contour level to help the reader assess the strength of the signal coming from the ligands.

Reply. We agree with the reviewer that a figure showing the lipid density and the contour level would be informative. We have prepared a figure (**Supplementary Figure 3**) to show the lipid-like density in the allopregnanolone binding pocket and have defined the contour level.

Comment. Similarly, in lines 233-238, the ALP assignment in the ZOL/GABA dataset should be better described with a separate supplementary figure to be more convincing.

Reply. *Densities for neurosteroid in the ZOL/GABA dataset are shown in **Extended Data Figure 9h** and **9i** (extended data figures are reordered in response to the suggestion by another reviewer). To enhance figure clarity and presentation, we have added contour levels to the figure legend. To be consistent, we have added contour levels to other figure legends where densities for ligand molecules are shown.*

Figure 2, at line 541.

Cryo-EM density around APG is contoured at 6.6 σ .

(We changed the abbreviation of allopregnanolone from ALP to APG in response to the suggestion by another reviewer)

Figure 3, at line 557.

Cryo-EM density around DID and ZOL is contoured at 5.7 σ and 5.1 σ , respectively.

Extended Data Figure 9

At line 915.

Cryo-EM density around APG is contoured at 6.6 σ .

At line 928.

Cryo-EM density around APG is contoured at 5.1 σ .

Comment. lines 290-296 and extended data tables: the tables described the DID dataset as containing "endogenous GABA" which is puzzling with the last paragraph of the results that described a potentially conductive state in absence of GABA. Did the authors found density for GABA in the GABA-binding pockets in this dataset ? This needs to be clarified.

Reply. *The DID dataset does not have GABA added during receptor isolation or cryo-EM grid preparation, yet we have found density in the orthosteric binding pocket that is well fit by GABA, having copurified with the receptor. We have therefore described this dataset as containing "endogenous GABA" and have prepared an additional figure (**Supplementary Figure 6**) on the binding pocket with "endogenous GABA."*

REFeree #2

Minor comments:

Comment. Line 76, a word appears missing: ... enable release from the affinity resin..., delete "it"

Reply. *We appreciate the reviewer's identification of this typo. We have corrected it accordingly.*

Comment. Line 582, A word appears missing: ... obtained from one mouse brain preparation (that includes brains from 50 mice).

Reply. *We did use brains from 50 mice for each native receptor preparation. The reported masses are the average weight per mouse brain. We edited this line to make the meaning clearer.*

Comment. Figure 4, labelling "neurosteroid" in TMD is unclear

Reply. *We have increased the font for "neurosteroid" along with "GABA" from 5 to 6 in Figure 4 to make the labelling more legible.*

Comment. Extended Figure 2,3,4, labelling of subunits is very difficult to read – perhaps having the labelling "in bold" might make them clearer.

Reply. *We have changed the labeling to bold in these three Extended Figures, improving label clarity.*

Comment. Extended Figure 2, labelling of subunits in ortho-one-Fab appears wrong with two betas next to one another.

Reply. *We thank the viewer for catching this mistake. We have corrected the mislabels in Extended Data Figure 2.*

REFeree #3

Major points

Comment. The authors state already in the abstract that receptors with two alpha isoforms in the pentamer, one of which is $\alpha 1$, was "unexpected" for them – this is NOT at all the case. Many groups have indicated, as early as in the 1990-ies, that two different alpha isoforms can co- assemble in pentamers. This is less prominent for $\alpha 1$ - containing ones. An estimation based on various knock in mice by the Möhler/ Rudolph work (maybe Benke et

al) found that at least 20% of alpha1- containing pentamers contains a second alpha isoform. For other alpha isoforms, it is quite clear that they co-assemble with a1 – see e.g. Bencsits 1999, Scholze et al. 2022 PMID: 36279694 and many works by other groups. A re-balanced view of the past findings is urgently needed here.

Reply. *We appreciate this comment and have revised the text and cited the aforementioned papers accordingly.*

Comment. Lines 67-68: It is incorrect to state that ALP, DID and ZOL “target a1- containing receptors”. DID is likely as unselective as its parent flurazepam, ALP is an unselective steroid, and ZOL targets the zolpidem sensitive a1, a2, a3/ g2 sites with high affinity. The statement is at best misleading, or can be read as wrong.

Reply. *We thank the reviewer for this comment and have also revised the manuscript accordingly. At line 66, change “all of which target α 1-containing receptors” to “in the context of native receptors.”*

Comment. There is a major discrepancy between statements in the conclusion, and the contents in lines 87-89: “ analysis of the purified native receptor complexes by mass spectrometry identified all α and β subunits as well as the γ 1 , γ 2 , and δ subunits, demonstrating that α 1 –dependent isolation captured receptors containing most of the 19 GABA A R subunits.” – in contrast – “ na1 GABAA Rs comprise three structural populations: tri-heteromeric α 1 β 2 γ 2 receptors that constitute half the total population and two distinct assemblies containing one α 1 subunit and one α 2/3/5 subunit.” There is no tabular overview summarizing the mass spec findings in the MS (only in the supporting Figure 1), and no explanation why only gamma2- containing receptor species were studied further. Please explain and add appropriate material to the MS. Supporting Figure 1 panel I summarizes the mass spec findings, and shows that the gamma1 plus delta- populations added together are larger than the gamma2- population. In order to fully credit previous work and to provide a balanced view onto a1- containing receptor species, a table with compositions that are suggested by the mass spec data and by the literature, in addition to those that are presented with the cryo- EM data, would vastly improve the presentation and interpretation of the findings.

Reply. *We sincerely appreciate this comment. Indeed, the apparent discrepancy between the mass-spec data and our cryo-EM analysis lies in their different sensitivities in detecting protein subunits. While the mass-spec approach is highly sensitive and able to detect small amounts of different subunits, the cryo-EM methods are relatively less sensitive. Because the major focus of work is on the structures of subunit complexes, we feel it is appropriate to limit our discussion to the major population of subunit complexes that are supported by*

*both techniques. In response to the reviewer's comments, we have provided **Supplementary Tables** detailing the mass-spec data, which include two independent mass-spec analysis of our native receptor preparations. We have also updated the mass-spec method section (lines 650–658) to reflect the inclusion of additional data.*

Comment. The reader cannot follow where the beta3 subunit was “lost” – in Figure 1, only beta1 and beta2 are mentioned. The mass spec data gives b1/b2/b3 to be 14/26/20 %. Did you find that gamma2-alpha1- containing pentamers never contain beta 3? This would be a remarkable finding, if I interpret Figure 1 well. If this is not what you imply, clear writing (and the above suggested table) would greatly help the reader to follow through your data.

Reply. *We appreciate the reviewer's critique on the issue of the β 3 subunit. On one hand, the mass spec data support the presence of the β 3 subunit. However, while it is tempting to use the mass spec data to determine the relative proportion of different subunits, a more sophisticated mass spec approach is required, using isotopically labelled peptides, and this deeply involved experiment is simply beyond the scope of the present work. On the other hand, our cryo-EM reconstruction shows that that the β subunit in the two-Fab receptor is mostly β 2, and the β subunit in the one-Fab receptor is mostly β 1/ β 2, based on the side chain densities (Refer to the **Extended Data Figure 6** and the **Supplementary Information**). That being said, if the abundance of β 3 subunits at these β positions is small, such as 20% or less, their sidechain features may simply be obscured by those from the more abundant β 1/ β 2 subunits.*

*In addition, while examining our subunit assignment, we found that the non- α 1 subunit should be better described as α 2/ α 3, rather than α 2/ α 3/ α 5. The reasoning is included in the **Supplementary Information**. We have updated the figure and text accordingly.*

To reflect the caveat in our analysis, we have added the following statement in the manuscript. At line 130, after “pentameric assembly,” add “These subunit assignments only represent the most abundant subunits based on the cryo-EM maps. It is possible that other subunits are present, such as the β 3 subunit at the β positions, if their relative abundance is ~20% or less.”

Comment. The title indicates that the work deals with “regulated assembly”, and yet the majority of figures deals with drug binding modes and conformations. As stated above, much of the mass spec data was not interpreted, nor used to reach conclusions about assemblies with beta3, gamma1, delta or alpha4 subunits. Thus, it is suggested that the authors rebalance the data presentation in such a way, that all findings from the mass spec work are presented and properly interpreted in the MS, and reduce a bit of the detail on

binding modes and conformational changes to make the most of the data in terms of assembly regulation and pools of alpha1- containing receptors.

Reply. *We appreciate this concern and, first, have revised the title. Though we would like to take as great of advantage of the mass spec data as the reviewer suggests, we are constrained by the experimental fact that the mass spec data only tells us if a peptide, and correspondingly a subunit, is present. We do not learn about molar ratios, an observation that would be required to carry out the interpretation the reviewer is suggesting. While we agree that it would be ideal to have molar ratios of subunits from mass spec data, that type of a mass spec experiment requires custom synthesized and isotopically labeled peptides for each receptor subunit and, in addition, a great deal of mass spec experimentation and analysis, which is deeply involved and simply beyond the scope of the present study. Nevertheless, we have re-analyzed our mass spectrometry data and found evidence to support the presence of both proposed splicing isoforms of the $\gamma 2$ subunits (Whiting 1990) at the protein level, which is difficult to achieve with the cryo-EM and highlights the particular strength of mass-spec in protein subunit identification. We have included that finding in the manuscript at line 89. In addition, we have presented mass spectrometry data including the peptide information as **Supplementary Tables**.*

Minor points

Comment. IUPHAR recommendation discourages the use of subscripts for subunit isoforms – subscripts should be reserved for stoichiometry, thus, the authors are encouraged to use IUPHAR recommended terminology throughout.

Reply. *We thank the reviewer for this suggestion. We have removed the subscripts from subunit isoforms throughout the manuscript text and figures.*

Comment. Why was the cerebellum excluded from the analysis? Please provide the rationale for the study design.

Reply. *Previous research by Yamasaki (2017, PMID 28279354) showed that that GABA_A receptors in cerebellum form a tripartite complex with LHFPL4 and neuroligin 2. In order to make the best use of the animals and the resulting tissue, we saved the cerebellum tissue for future studies directed toward capturing the tripartite complex.*

Comment. Line 210: “disassociated” – should read dissociated?

Reply. *We have incorporated the suggested edit.*

REFEREE #4

Minor comments

Comment 1) Line 43 "Ref.5" is put in parenthesis - please cite the paper according to the style of the remaining.

Reply. In accord with journal convention, the citation was written as such to prevent the confusing letters/symbols "Zr²⁺⁵".

Comment 2) Line 49: The authors write "In addition, ganaxolone, a synthetic mimetic of allopregnanolone..." - Please indicate, that ganaxalone is a synthetic derivative of allopregnanolone (3b-methylated ALP). The synthetic version of allopregnanolone is called brexanolone and the FDA approved treatment for postpartum depression since 2019. However, that formulation additionally includes cyclodextrines for solubility reasons. In 2022 the FDA approved ganaxalone for the rare condition of CDKL-5 deficiency dependent seizures.

Reply. We appreciate these comments. At line 48, we revised "a synthetic mimetic of" to "a synthetic derivative of."

Comment 3) Line 76: please rephrase "release it the affinity resin"

Reply. We have made the following change to correct this sentence. At line 75, change "it" to "from."

Comment 4) Lines 95-97: Similar observations have been made with cerebellar $\alpha 1/\alpha 6$ -containing receptors, e.g. (Scholze et al., 2020)

Reply. We appreciate this comment and recognize that there are previous reports supporting the existence of GABA receptors with mixed α subunits in the brain tissue. Here, however, we are limiting ourselves to non-cerebellar receptors – we did not include cerebellar tissue in our preparations.

Comment 5) Line 161: To this reviewer's knowledge, the effects of neurosteroids are not selective to GABA A receptors, thus please rephrase or delete that statement accordingly.

Reply. We have rephrased this sentence to read "...properties by potentiating the activity of GABA_ARs."

Comment 6) Line 164: The text refers to Figure 2B, not to Figure 2A.

Reply. The figure call was meant to refer to the chemical structure of ALP in Figure 2A. We agree that this figure call may be confusing and we have moved it to line 165.

Comment 7) Line 166-169: Please acknowledge the previous work done: (Chen et al., 2019; Laverty et al., 2017; Sugasawa et al., 2019, 2020).

Reply. We have added the following statement to credit previous work. At line 175, we have included the statement, "These key residues identified are consistent with previous studies on recombinant GABA_ARs (Chen 2019, Sugasawa 2020) and GABA_AR-chimeras (Laverty 2017, Miller 2017, Chen 2018)." In order to keep the number of references to 50 or fewer, as required by the journal, we have accordingly removed several other references.

Comment 8) Line 198-201: Do the authors have any theory on how the same molecule (ALP) could open and modulate the receptors simultaneously via the same binding sites? On a single receptor level, when the exact same binding sites are involved into agonistic and allosteric effects of ALP, and both of its binding sites are already occupied with ALP molecules at low concentrations (allosteric effects), how can the receptor even sense higher concentrations of ALP and react differently (agonistic effect) to it? Additionally in the introduction the authors claim agonistic effects above 100nM of ALP, whereas the HPLC demonstrates >300nM ALP concentration in their preparation. However, only allosteric effects are observed. How could that be explained? Could additional unresolved sites exist within the TMD/ECD interface or within the ICD?

Reply. Our speculation is that the allosteric and orthosteric sites are coupled and that the affinity of allopregnanolone towards its binding site, and its resulting efficacy, depends on the state of the receptor, for example, by the concentration of GABA and the resulting occupancy of the orthosteric site. The agonistic effect of allopregnanolone, by definition, has been measured (Cottrell 1987, Callacha 1987) in the absence of GABA; while the potentiating effects of allopregnanolone have been measured with GABA concentrations between 1 μ M and 10 μ M. We hypothesize that without GABA, the closed state of the receptor is not saturated with 100 nM of allopregnanolone and can transition to the open state in response to higher concentrations of allopregnanolone. In our ZOL/GABA sample,

we indeed found more than 300 nM allopregnanolone with mass-spec, but 1 mM GABA was also present. This combined ligand condition is in accord with observing the potentiating effect of allopregnanolone. Regarding other potential sites, we cannot find any other sites within the TMD/ECD interface in our cryo-EM density map, although there may be sites that we cannot resolve or indirect effects on the receptor via perturbations in the lipids/membrane. We also have not found any binding sites within the ICD.

Comment 9) Line 246: See major comment 2) on the main metabolite of flurazepam.

Reply. *Please refer to our response to the major comment 2.*

Comment 10) Line 258: Which imidazole nitrogen is meant?

Reply. *"By imidazole nitrogen" we meant the nitrogen atom that is only part of the imidazo moiety. We have changed the wording to read "...and the imidazo nitrogen..." at line 267.*

Comment 11) Line 308: Classifying GABAA receptors based on their alpha subunits only is a new concept to this reviewer, please clarify. The community used to seek for specific modulators that would spare certain α -containing receptor populations and be effective at others to generate molecules with improved pharmacological profiles (such as the search for anxi- selective drugs which spare $\alpha 1$ -containing receptors (meant were $\alpha 1\beta\gamma 2$ receptors) while remaining activity at $\alpha 2$ -containing receptors - a concept that turned out not to translate into humans, see Skolnick 2012. However, entire receptors have not been categorized according to their α -subunits solely. Receptor classification has been based on subunit composition (e.g. $\alpha 1\beta 2\gamma 2$) and interfaces they included, e.g high affinity benzo sites have been classified based on the respective α and γ - subunit that form it.

Reply. *it is a common or even prevalent practice to distinguish receptors using their α subunits in many pharmacological studies where the same β and γ subunits are kept constant while changing the α subunits. We agree with the reviewer that classifying GABA_A receptors based on their α subunits appears simplistic and we have classified receptors based on subunit composition and arrangement.*

Comment 12) Line 310: For clarity reasons, it should be mentioned again, that allopregnanolone is the most abundant endogenous neurosteroid, not exclusively a medication for postpartum depression.

Reply. We thank the reviewer thank for this suggestion and have rephrased the sentence to highlight this information. At line 319, we changed the text from “poses of the postpartum depression medication, allopregnanolone,” to “poses of allopregnanolone, an endogenous neurosteroid that is also used for treating postpartum depression,”

Comment 13) ALP is often referred to as high affinity benzo site ligand alprazolam, which can be confusing. AloP is commonly used in the literature to refer to allopregnanolone.

Reply. We thank the reviewer for the suggestion. To be consistent with the naming of other ligands in the manuscript, we have referred allopregnanolone as APG throughout the manuscript.

Minor comments to figures

Figure 1:

Comment. Asterisks should be explained in figure legends again for clarity.

Reply. We appreciate the reviewer’s suggestion. We have addressed it in together with the next comment.

Comment. The legend should also include the clockwise orientation for intuitive interpretation for an expert reader (who usually sees subunit arrangement given in counterclockwise manner)

Reply. We thank the reviewer and have noted that the subunits are counted clockwise in the figure legend. At line 523, we have added “(* denotes that the subunit is next to the $\gamma 2$ subunit, and subunits are counted clockwise when viewed from the extracellular space)”

Comment. It should be stated in the figure legend that cerebellum was excluded.

Reply. We have mentioned that the cerebellum was excluded in the figure legend. At line 526, add “(cerebellum excluded)” after “from mouse brain”

Comment. Please comment on n=2 for the cryo-EM experiments in Panel 2B

Reply. We have included the following statements to address why n=2 for the cryo-EM experiments.
At line 532, we added “The DID cryo-EM dataset was excluded from the analysis because it

was not purified the same way as the TIRF samples and contained a smaller particle count than the other two cryo-EM datasets."

Figure 2:

Comment. The bottom 2D-chemical structure of ALP (allopregnanolone) needs rework and additional explanation of the color code used in the figure legends. ALP is drawn with a 3 α -hydroxymethyl (black line indicating the existence of a carbon atom, red line indicating a hydroxy group), which is incorrect. The functional group at C20 is represented the same way as the one on position 3, however it is a carbonyl group, thus a double bond should be drawn here.

Reply. *We thank the reviewer for the comments. We have updated the 2D chemical structure to better represent the carbonyl group.*

Comment. Again, asterisks should be mentioned in the legend text.

Reply. *We have explained the notation in the legend at line 538.*

Comment. It should be stated in the figure legend for panel b that two-Fab receptors are shown for easier processing of the figure.

Reply. *We have stated the nature of the receptor for panel b in the legend.*

Figure 3:

Comment. Ball and stick representation of the ligands makes the appreciation of interactions for both molecules extremely difficult. Different rendering should be used.

Reply. *We thank the reviewer for his feedback. We have changed the style of ligand from CPK to stick.*

Major comments

Comment 1. Which of the proposed splice variants of the g-subunit (short, long) was observed in the mass spectrometry studies and Cryo-EM experiments? This finding should also be discussed.

Reply. *We thank the reviewer for this important question. Out of ~500 residues, the two proposed spliced forms of the γ 2 subunits (short and long), only differ in the M3-M4 loop*

by 8 amino acid, which is underlined in the $\gamma 2$ -L sequence "KKKKNPLLRMFSFKAPTIDIRPRSATIOQMNATHLQER". This relatively small difference makes it challenging to differentiate them, even by with mass spectrometry. Fortunately in the mass spec data, there were two detected peptides "APTIDIRPR" and "APTIDIRPRSATIOQMNATHLQER" that can only be derived from the long isoform, unambiguously supporting its presence in our native receptor preparation. In addition, the peptide "KNPAPTIDIRPR" that is unique to the short isoform was also detected with mass spectrometry. Because we did not observe densities for the differentiating region in the cryo-EM reconstruction, we would use the protein mass-spec data only to determine the splice isoform of the $\gamma 2$ subunit. In summary, our data support the presence of both splice variants at the protein/receptor level.

We have supplied additional mass-spec data as **Supplementary Tables** and recapitulated the discussion above in the manuscript.

At line 88, we have added the following statement:

"Our mass spectrometry data detected peptides unique to both isoforms of the $\gamma 2$ ($\gamma 2$ -L and $\gamma 2$ -S) subunit (Whiting 1990), thus supporting their presence at the protein level in the brain."

Comment 2. The authors of the study assert that they observed native receptors in combination with three clinically relevant drugs: allopregnanolone, flurazepam, and diazepam. It should be noted however that flurazepam is a prodrug that quickly converts to active metabolites, with various studies identifying multiple metabolites in human venous blood, including monodesethyl-flurazepam, didesethyl-flurazepam, flurazepam-aldehyde, hydroxyethyl-flurazepam, and desalkyl-flurazepam. In contrast to the claim in line 246 about DID-flurazepam being the main flurazepam metabolite and the presence of DID after flurazepam administration, N-desalkyl-flurazepam is generally recognized as the primary active metabolite, with a half-life of up to 100 hours, which is consistent with increased flurazepam efficacy following repeated administration due to accumulation. N-desalkyl-flurazepam is also a metabolite of various other benzodiazepines, such as midazolam, quazepam, ethyl loflazepate and others, further increasing its relevance. The reasoning behind the use of DID-flurazepam (Didesethyl-flurazepam) in the study and the clinical relevance in comparison to other recognized flurazepam metabolites was not explained properly. Please rewrite for clarity. Thus, it is important to note that while flurazepam and its metabolites have clinical relevance, the clinical aspect of this finding should be presented in a balanced and appropriate manner, without overstating its significance and should be discussed in the manuscript for clarity.

Reply. We are grateful to the reviewer for providing us with insightful details on flurazepam pharmacokinetics. According to the classic study by Schwartz and Postma

(1970), the initial steps of flurazepam metabolism involve serial N-dealkylation to form monodesethyl-flurazepam and didesethyl-flurazepam. Therefore, we consider it reasonable to refer to didesethyl-flurazepam as one of the major metabolites of the prodrug. While we agree that N-desalkyl-flurazepam has a long half-life and is likely the active metabolite during prolonged use of flurazepam, we believe that the technical aspects of clinical relevance are better suited to specialized journal articles that have already been published. Additionally, because the molecular interactions we captured in our drug-receptor structure are limited to the core moiety of flurazepam, we expect these interactions to be conserved across the prodrug, didesethyl-flurazepam, and N-desalkyl-flurazepam. Therefore, we feel that discussing the structural differences that are unlikely to affect drug binding modes would detract from the manuscript's focus. In line with this, we have revised the manuscript to refer to didesethylflurazepam as "one of the major metabolites" to avoid any confusion.

Comment 3. Lines 298-306: Kindly provide a summary of the authors' interpretation of their findings regarding the proposed receptor stoichiometries and arrangements for a1-subunit containing assemblies, such as a1-b2-a1-g2-b2 / a1-bx-ax-g2-bx / ax-bx-a1-g2-bx. Why do the authors exclude other native a1- containing receptors than those? How do their findings contradict the existence of b-b interfaces in a1-containing populations, which is in seeming contradiction to the literature? In their mass spectrometry experiments, the authors claim to have identified all alpha subunits, including a4 and a6, all b-isoforms, including b3, as well as g1 and d in a1-containing pentamers. However, none of these assemblies are discussed further. Please explain. Is it possible that beta-beta interfaces could exist in a small fraction of assemblies, but they could not be detected due to their low abundance? Can you really exclude a1-bx-ax-bx-bx or a1-bx-ax-d-bx receptors (which are considered to be a minority of all a1-containing receptors, see (Bencsits et al., 1999)). Similarly, why could models of receptors containing a4/a6 g1, and delta not be built? Their existence appears to be strongly suggested by the presented mass spectrometry data. Which explanation for the absence of b3-subunits in the cryo-EM experiments can be made, despite their high abundance in neuronal cells? Please discuss on the limitations of the method to resolve native receptors for a non-expert readership. The authors ought to address these questions and moderate their claims about the potential existence of a1-containing native GABAA receptors accordingly.

As the comment is lengthy, we divide it into smaller parts and address each of them in sequence.

Comment 3.1. Lines 298-306: Kindly provide a summary of the authors' interpretation of their findings regarding the proposed receptor stoichiometries and arrangements for a1-subunit containing assemblies, such as a1-b2-a1-g2-b2 / a1-bx-ax-g2-bx / ax-bx-a1-g2-bx. Why do the authors exclude other native a1- containing receptors than those?

Reply. *The interpretation of the cryo-EM data led to the identification of the receptor assemblies representing the major populations of α 1-containing receptors. If the reviewer is asking why only α - β - α - β - γ assemblies are discussed, it is because we only observe this arrangement in our data analysis. If the reviewer is asking why certain subunits are omitted, it is because we cannot find protein sidechain density that support their presence in the cryo-EM reconstruction. We have modified the **Extended Data Figure 6** and included additional **supplementary information** to document our subunit interpretation and assignment process.*

Comment 3.2. How do their findings contradict the existence of b-b interfaces in α 1-containing populations, which is in seeming contradiction to the literature?

Reply. *Several studies, including the structural work from Kasaragod (2022), used α 1 containing receptors harboring a β - β interface. In our cryo-EM analysis of the native α 1-containing receptor pool, we simply do not observe any receptors with a β - β interface. As previously noted, however, receptors with a β - β interface may be present, yet of such low abundance (<5%) as to not be detectable by cryo-EM.*

Comment 3.3. In their mass spectrometry experiments, the authors claim to have identified all alpha subunits, including α 4 and α 6, all b-isoforms, including b3, as well as g1 and d in α 1-containing pentamers. However, none of these assemblies are discussed further. Please explain.

Reply. *We did not discuss these assemblies because they do not sufficiently populated to show up in the cryo-EM data analysis. Mass spectrometry can identify a protein subunit based on the detection of a single peptide fragment, and is substantially more sensitive than cryo-EM.*

Comment 3.4. Is it possible that beta-beta interfaces could exist in a small fraction of assemblies, but they could not be detected due to their low abundance?

Reply. *Yes. We estimate that if receptors with β - β interfaces are present in less than ~5% of the particle population, we will have a difficult time finding them in the cryo-EM analysis.*

Comment 3.5. Can you really exclude α 1-bx-ax-bx-bx or α 1-bx-ax-d-bx receptors (which are considered to be a minority of all α 1-containing receptors, see (Bencsits et al., 1999).

Reply. *It is possible that the binary α - β - α - β - β receptor and the α - β - α - β - δ receptor are present in the α 1-containing receptor pool. We estimate that their abundance needs to be less than ~5% for us to not find them in the 3D classification.*

Comment 3.6. Similarly, why could models of receptors containing α 4/ α 6, γ 1, and δ not be built? Their existence appears to be strongly suggested by the presented mass spectrometry data. Which explanation for the absence of β 3-subunits in the cryo-EM experiments can be made, despite their high abundance in neuronal cells?

Reply. *We cannot reconstruct the 3D structures for these receptor assemblies; hence their molecular models were not built. The reconciliation between cryo-EM analysis and mass-spec data lies in their different sensitivities in detecting protein subunits. We cannot absolutely exclude β 3 from our proposed assemblies, but it is less populated compared to β 1 and β 2 in the α 1-containing receptor pool. This is not necessarily in contradiction with their high abundance in neuronal cells because they may associate more frequently with subunits other than the α 1 subunit.*

Comment 3.7. Please discuss on the limitations of the method to resolve native receptors for a non-expert readership. The authors ought to address these questions and moderate their claims about the potential existence of α 1-containing native GABAA receptors accordingly.

Reply. *Yes, we have expanded our discussion to resolve the apparent conflict between cryo-EM and mass-spec in the manuscript at lines 130–133.*

Comment 4. Lines 159-201; Panel 2d,e; extended Figure 7 and 8: Despite the nice conformational change that is indicated due to ALP binding in Figure 2D ext. Fig 7 and 8, this reviewer sees open questions and discrepancies between the claims and the cited sources: Please explain why 6I53 was used as a comparison to the ALP bound structures in this work. To this reviewer's knowledge, 6I53 is not in complex with GABA (as stated) but only with a megabody38, thus the comparison completely lacks plausibility. Additionally, please state why 6X3Z was not used instead – since it is a GABA-bound structure of the same protein complex (a1b2g2 rather than a1b3g2). The conformational changes induced by ALP have to be compared to the existing GABA bound a1b2g2 structure 6X3Z, despite missing ICD. If differences are found, they should be discussed properly in the main text, which should be rewritten accordingly. Those questions need to be addressed carefully prior to publishing. Furthermore, the possible impact of different lipids, cell types and other conditions should be addressed as well. Additionally, how does binding of ALP interfere with the organization of both Apo-GABA binding sites locally (besides ECD rotation) and the high

affinity zolpidem/DID-Flurazepam binding site within the pentamer in comparison to 6X3Z; are there any conclusions to draw from this?

As the comment is lengthy, we divide it into smaller parts and deal with each of them in sequence.

Comment 4.1. Lines 159-201; Panel 2d,e; extended Figure 7 and 8: Despite the nice conformational change that is indicated due to ALP binding in Figure 2D ext. Fig 7 and 8, this reviewer sees open questions and discrepancies between the claims and the cited sources: Please explain why 6I53 was used as a comparison to the ALP bound structures in this work. To this reviewer's knowledge, 6I53 is not in complex with GABA (as stated) but only with a megabody38, thus the comparison completely lacks plausibility.

Reply. *The reason we used 6I53 for the structure comparison is because it is the only full-length triheteromeric GABAAR structure that is in complex with GABA, but not other drug ligands, making it a good reference to evaluate the effect of the neurosteroid. Despite the lack of an explicit GABA label in the PDB deposition, the authors themselves confirmed the GABA-bound nature of this structure in the paper, "Although our structure was solved in the absence of exogenous GABA, we observe serendipitous non-protein densities at the agonist-binding sites of both β^+/α^- interfaces (Extended Data Fig. 9b,c)." Consistent with it being a GABA-bound complex, the 6I53 structure has a complete loop C closure. Additionally, we have compared the GABA binding pockets of 6I53 with 6HUO (stated as in complex with Xanax and GABA) and found that they are similar. Between these two structures, the β^+/α^* (* denotes the subunit is adjacent to the $\gamma 2$ subunit) pocket (residues that are less than 5 angstroms away from the bound GABA; including 11 residues $\alpha 65F$, $\alpha 67R$, $\alpha 118L$, $\alpha 120T$, $\beta 97Y$, $\beta 155E$, $\beta 156S$, $\beta 157Y$, $\beta 200F$, $\beta 202T$, $\beta 205Y$) has a backbone RMSD of 0.23 Å and an all-atom RMSD of 0.37 Å; while the β^*/α^- pocket has a backbone RMSD of 0.23 Å and an all-atom RMSD of 0.37 Å. As a result, we think it is valid to use the 6I53 structure as the GABA-bound reference model for the structural comparison.*

Comment 4.2. Additionally, please state why 6X3Z was not used instead – since it is a GABA-bound structure of the same protein complex (a1b2g2 rather than a1b3g2). The conformational changes induced by ALP have to be compared to the existing GABA bound a1b2g2 structure 6X3Z, despite missing ICD.

Reply. *We did not use 6X3Z for the structural comparison mostly because the missing ICD, which has been implicated to be crucial for the neurosteroid modulation of GABA receptors in previous studies. Most notably, it has been shown that the allosteric modulation of GABA receptors depends on the kinase and phosphatase activity (Fáncsik 2000) and the phosphorylation sites of GABA receptor subunits are located at the ICD. In addition, the omission of ICD in 6X3Z may result in structure perturbations in/around the M3/M4 helices*

near the cytoplasmic side, which is close to identified allopregnanolone binding sites. Nevertheless, we have prepared an additional page of figures, numbered as **Extended Data Figure 8**, dedicated to address this issue. Here, we have compared the TMD conformations between 6X3Z and 6I53, and included all the structural comparison with 6X3Z that are in parallel to what we did with 6I53. All the findings are described in the figure legend.

To accommodate these changes, we have renumbered the Extended Data Figures and moved the previous **Extended Data Figure 9** into the supplementary information as **Supplementary Figure 4** due to the figure number limit.

Comment 4.3. If differences are found, they should be discussed properly in the main text, which should be rewritten accordingly. Those questions need to be addressed carefully prior to publishing.

Reply. We have found similar structural rearrangements using the structure 6X3Z as the reference. At line 188, after “pharmacology”, add “Structural comparison with the GABA-bound recombinant $\alpha 1\beta 2\gamma 2$ structure (PDB 6X3Z) revealed similar TMD rearrangements near the APG binding pockets (**Extended Data Figure 8**).”

Major comments to figures Figure 3 panel f

Comment. The authors claim that the DID-bound structure is potentially conducting due to an overall pore radius above 1.81 Angstrom. However, GABA was omitted in the preparation and no density for neurosteroids could be resolved. An agonistic effect of a benzodiazepine has never been observed before profoundly to this reviewer’s knowledge. Thus, if the authors claim the depicted conformation might reflect a conductive state, then please provide electrophysiological data with DID-flurazepam agonistic effects in absence of allopregnanolone.

Reply. We thank the reviewer for this comment and have revised the comments related to the DID-bound structure as “possibly representing a partially open state” at line 300.

Comment. If neurosteroids were present in agonistic concentrations but could not be resolved (as given as possible explanation in the main text), how does this theory fit with the observations made in comparison of ALP/GABA vs zolpidem/GABA bound states from extended data figure 8j (0.25 Angstrom difference).

Reply. We understand the structural similarity between the APG/GABA structure and the ZOL/GABA as follows. In the ZOL/GABA structure, APG is also bound to the same sites, thus

the main difference between ZOL/GABA and APG/GABA is the presence of ZOL in the former. Previous studies (Masiulis 2019, Kim 2020) showed that benzodiazepines caused minimal changes when the receptor is bound with GABA, we think, in a similar fashion, ZOL causes minimal changes when the receptor is bound with saturating amount of GABA in the desensitized state.

Comment. Furthermore, if neurosteroids were present in that structure, would it be expected then, that DID-flurazepam binding is reminiscent of diazepam/alprazolam binding in earlier studies?

Reply. *We cannot confidently assess the effect of the unresolved neurosteroid on DID binding. It should be noted we have not measured the concentration of endogenous neurosteroid in the DID sample. That being said, the DID binding is reminiscent of previous structural studies with other benzodiazepines, which is expected. If the reviewer is alluding to why the DID structure has a distinct TMD conformation from previous benzodiazepine structures that adopted the desensitized state, we believe the difference was the omission of a high concentration of GABA from the sample preparation. Although we can observe GABA density in the GABA pockets, the GABA pockets are different from previous GABA-bound structures, with the loop C incompletely closed (**Figure 3e**) and with GABA in a slightly different conformation (refer to the **Supplementary Figure 6** for more details). We speculate that, in the DID structure, we may have captured a structure that is between the closed and open states.*

Comment. Please reevaluate the proposed conformations of given structures based on other GABAA receptor publications (such as the histamine-bound 7QN9 from Sente et al., 2022, Aricescu lab, who claim a pre-open receptor conformation or binary ab receptor conformations as from Miller et al, 2022, and other $\alpha 1\beta 2\gamma 2$ receptors such as 6X3X / 6X3U from Hibbs lab).

Reply. *We have compared the pore conformation of the DID structure to these GABA_AR structures suggested by the reviewer as well as open/activated structures of channel from the cys-loop family in **Supplementary Figure 5**.*

Figure 4

Comment. Line 546: The effect of neurosteroids, including endogenous neurosteroids cannot be generalized that way. Endogenous neurosteroids with excitatory function and negative allosteric modulation at GABAA receptor subtypes are known, such as pregnenolone sulfate (PS) or dehydroepiandrosterone (DHEA). They certainly do not always

increase Cl⁻ influx through GABA_A receptors. Please rephrase accordingly also in the main text.

Reply. *We thank the reviewer for this suggestion and agree that certain neurosteroids may have unique properties that do not fit into this diagram. We have narrowed our scope of endogenous neurosteroids to positive modulators of GABA_A receptors at line 322 and revised the figure legend accordingly.*

Reviewer Reports on the First Revision:

Referees' comments:

Referee #3 (Remarks to the Author):

As this is a re-review, the comments will be provided without adhering to the questions.

I would like to emphasize once more that it is a fantastic accomplishment to present experimental structures of three populations of native mouse GABA-A receptors selected for the $\alpha 1$ - subunit, very worthy to be published at high impact – BUT – the authors have chosen to dismiss some key points from my previous review. I consider them sufficiently crucial to persevere as explained point by point below, because the prior research is not sufficiently acknowledged, and the findings are interpreted and presented in a very misleading fashion. The authors have identified a large panel of $\alpha 1$ - containing receptors, based on their mass- spec data. Only the largest three populations were suitable for the structure determination – which is fine. But, it is NOT fine to suppress the additional populations in the text. Every pentameric assembly that exists in a mammalian brain has a role, as can clearly be understood considering e.g. the severity of disease that can be observed with variants of e.g. the delta or even the pi subunit (seizure disorders). Thus, ALL key findings should be properly presented and discussed in the paper, not limited and restricted to the cryo-EM data. Once more, I offer specific and very easily implementable suggestions.

Major points:

In the Abstract the authors state “only three structural populations” – this is still misleading and should be re-phrased to state “three major populations”

The statement “two unanticipated assemblies containing one $\alpha 1$ and either an $\alpha 2$ or $\alpha 3$ subunit.”

Also is misleading, as it was pointed out previously in the initial review that a large body of literature pointed towards assemblies of $\alpha 1$ plus αx : Bencsits et al (doi:

10.1074/jbc.274.28.19613) demonstrated many years ago that $\alpha 4$ co- assembles with all alpha isoforms except alpha6. Co-assembly of $\alpha 6+\alpha 1$ is long acknowledged. Many additional papers out there on $\alpha 2+\alpha 3$, $\alpha 1+\alpha 5$, $\alpha 2+\alpha 1$ - Why do you call such long accepted assemblies “non- canonical”? Please acknowledge prior work and abstain from claiming “novelty” for long accepted facts as a sign of appreciation for other researchers.

Line 124: “consistent with genetic (22) ... evidence” Reference 22 is far from providing any genetic evidence for the $\alpha 1$ - $\beta 2$ - $\alpha 1$ - $\beta 2$ - $\gamma 2$ receptor. In fact, genetics cannot provide evidence for any particular subunit arrangement at all. Please reword accordingly. (I consider this a major point simply because high impact papers, by highly renowned researchers, should be educational beyond the methodical advances, and writing should reflect the limitations of the disciplines properly.)

Line 92 “ $\alpha 1$ -GABA A Rs comprise three structural populations” is misleading, as was pointed out previously. Your mass spec data indicates that there must be at least populations with $\alpha 1+\gamma 1$, $\alpha 1+\gamma 2$, $\alpha 1+d$ (!), $\alpha 1+ \alpha x$ ($x=2, 3, 4, 5, 6$) and $\alpha 1 +$ any one or two of $\beta 1, \beta 2, \beta 3$. Due to limitations of the method, you studied the LARGEST populations, which turned out, not surprisingly, $\gamma 2$ containing. This can and should be stated clearly. The other populations might be small – but they contribute to brain function and should not be ignored. As suggested in the original review, the mass spec data deserves a place in the MS (at least a simple table), and appropriate discussion.

Another point here: How high is the confidence with which you can distinguish $\gamma 2$ from $\gamma 1$?

Lines 153- 163 : “74% sequence similarity between $\alpha 1$ and $\alpha 3$ ” – up to here, the non- $\alpha 1$ alpha

subunit is denoted as $\alpha 2/3$, and now you go back to $\alpha 3$ only in the entire passage? This seems inconsistent. Please keep in mind: Readers from different disciplines need to understand the degree of faith with which entities are assigned to experimental evidence.

Line 307 "Our study reveals that $\alpha 1$ -GABA A Rs comprise three structural populations:" is still wrong. This study describes the MAIN populations in detail, and identified ADDITIONAL populations which occur in sufficiently low quantity to prevent downstream structural studies. Once more, these populations are OF MAJOR INTEREST and should not be ignored or suppressed, even if no structure determination was possible. They deserve to be presented. I truly feel it is a waste of money to generate the mass spec data and then to marginalize them in a high impact publication.

Minor points:

line 42, including Zn²⁺ (ref. 3) – ref 3 should be superscript like all other references

Referee #4 (Remarks to the Author):

Review of revised version of „Cryo-EM structures reveal native GABAA receptor assemblies and pharmacology“ for Nature by reviewer #4

I want to emphasize the importance of this study and the great achievement of the group to resolve native GABAA receptors, especially in presence of endogenous neurosteroids. This strengthens previous reports and provides additional new insights into GABAA receptor gating by various molecules. However, this reviewer thinks that even though some of the technical points raised were addressed well, major concerns were not addressed satisfactorily. Critically, important information to major reviewer concerns should not be placed in the supplement without discussing them further in the manuscript.

This study claims to allow drug development in a target-specific manner. However, once more I want to emphasize that ALL the results of the performed work need to be presented in a clear fashion so it can be interpreted correctly by a broad readership to benefit from this study. Additionally, I once more want to emphasize that the way the synergistic power of mass spectrometry data in terms of sensitivity (as also given as a reply by the group to various reviewer comments) and cryo-EM data need to be explained in more detail in the manuscript and are still misleading in the present version throughout the manuscript. Mass-spec data must not be ignored, as it is currently selectively done throughout the manuscript. As this manuscript has the potential to address a broad readership and to be cited a lot in the future, the way results are presented currently is predestined to cause the spread of incomplete and partly misleading information in the community. This reviewer will endorse publication only if ALL findings from this study are presented in a clear, non-misleading way, which requires a rigorous re-writing, rather than adapting a few sentences.

Major comments:

Question to reply to major comment 1:

It is great that the natural occurrence of both splice isoforms is addressed now in the manuscript. However, this reviewer does not fully understand the answer given by the group:

"...forms of the $\gamma 2$ subunits (short and long), only differ in the M3-M4 loop by 8 amino acids, which is underlined in the $\gamma 2$ -L sequence "KKKKNPLLRFMSFKAPTIDIRPRSATIQMNNATHLQER". This relatively small difference makes it challenging to differentiate them, even by with mass spectrometry. Fortunately, in the mass spec data, there were two detected peptides "APTIDIRPR"

and "APTIDIRPRSATIQMNNATHLQER" that can only be derived from the long isoform, unambiguously supporting its presence in our native receptor preparation."

Q: It is correct that both isoforms differ by the given 8 amino acids. However, the two given peptide fragments could be derived from both isoforms sequence-wise, since they start right after the insertion that is specific to the g2L version and thus are included by both versions. How can the native existence of the long isoform be proven with those fragments? If something was misunderstood, please explain.

Q to reply to major comment 4.1:

"the reason we used 6I53 for the structure comparison is because it is the only full-length triheteromeric GABAAR structure that is in complex with GABA, but not other drug ligands, making it a good reference to evaluate the effect of the neurosteroid. Despite the lack of an explicit GABA label in the PDB deposition, the authors themselves confirmed the GABA-bound nature of this structure in the paper, "Although our structure was solved in the absence of exogenous GABA, we observe serendipitous non-protein densities at the agonist-binding sites of both β +/ α - interfaces (Extended Data Fig. 9b,c)." Consistent with it being a GABA-bound complex, the 6I53 structure has a complete loop C closure."

REMARK: The authors that published 6I53 further state "Although we cannot reliably determine the identity of the contaminating ligand, we carried out in silico molecular docking of GABA within these sites". Thus, GABA could not be reliably identified in a probe that was prepared without GABA, but with high mM concentrations of contaminants. Therefore, it is great that the authors compared their structures to the proposed 6HUO and included their findings into the reply letter, however, the comparison between 6I53, 6HUO, 6X3X and novel structures should be accessible in form of a figure and be discussed. This could have been included or been in addition to the new supplementary figure 8.

Major comment 3

As suggested by the mass-spec data in this study, α 1- subunit-containing GABA_A receptors form hetero-pentamers with a wide variety of GABA_A receptor subunits – far more than those that were identified in the resolved cryo-EM structures by the group. Those include α 1-containing receptors with ALL other alpha isoforms, ALL beta isoforms, ALL gamma isoforms and the delta subunit. As stated in the reply letter by the group, receptors with low abundance (less than 5%) cannot be resolved by cryo-EM experiments, despite their undoubtable existence! The existence of these smaller populations must be fully addressed in the manuscript before publication.

It is long known, that α 1 β 2 γ 2-containing receptors represent the major GABA_AR population in the brain. It is also suggested for more than 20 years that α 1-containing receptors can have multiple α - or β - isoforms incorporated -as is now addressed by citation in the paper.

However, therefore it is also common knowledge that the other GABA_AR isoforms present a very small fraction in the brain. Since there is so many of them theoretically, it is a given that the majority of them can NOT be resolved in current cryo-EM experiments. That does not exclude their existence! The mass-spec data of this study in contrast to the claims made in the manuscript even prove their existence and as derived from the reply letters - the only reason they could not be resolved is due to inherent limitations of the cryo-EM experiments. Severe disease-causing variants in ALL the subunits identified in the mass-spec data essentially highlight their physiological importance and thus inclusion into functional pentamers further.

It is unacceptable to ignore or marginalize their existence and emphasize the existence of the proposed α 1-containing receptors exclusively, because no other population could be resolved with the main experimental technique.

I especially refer to:

Line 31-33: This statement in the abstract is strongly misleading considering the mass-spec data. The group resolved major populations of $\alpha 1$ -containing receptors, whereas other existing assemblies with less abundance could not be resolved in the cryo-EM experiments owe to technical limitations of the used method. In line with this, GABAA receptor assembly does NOT "yield a small number of structurally distinct complexes" but the rest of complexes could simply not be resolved, despite their existence - as shown by the more sensitive mass-spec data. Thus, any assumption on assembly regulation, stoichiometry and arrangement of less prevalent receptors can simply not be made and proper writing needs to reflect this fact.

Line 113-117:

Again, this is simply wrong. In light of the mass spec data, $\alpha 1$ -containing receptors do form receptors with the d-subunit and with b-subunits that could both not be resolved here. Heterologous expression might just enable the detection of those receptor population due to overexpression compared to neurons, thus enrichment of those populations. Any conclusion on heterologous expression systems is highly speculative based on the mass-spec data. Possible differences in results, however, should again be discussed!

Line 307-309

This is both, wrong and strongly misleading. This study does not reveal that murine $\alpha 1$ - containing receptors do form three structural populations as stated, but it only resolved the three most abundant assemblies of $\alpha 1$ -containing receptors could be resolved, with no information about the rest. Yes, identifying three major native receptor assemblies in cry-EM experiments speaks for regulation of $\alpha 1\beta 2\gamma 2$ receptor assembly in neurons, but nothing more than that.

Thus, the otherwise outstanding work will benefit greatly from appropriate framing and discussion of all $\alpha 1$ - assembly partners that were identified by all utilized methods, and this reviewer looks forward to the improved version.

Major points:

Comment. In the Abstract the authors state “only three structural populations” – this is still misleading and should be re-phrased to state “three major populations”

Reply. *We have edited the abstract to read “..three major structural populations...”.*

Comment. The statement “two unanticipated assemblies containing one α_1 and either an α_2 or α_3 subunit.”

Reply. *In accordance with the reviewer’s suggestion, we have changed “...two unanticipated assemblies...” to “...two assemblies...”.*

Comment. Also is misleading, as it was pointed out previously in the initial review that a large body of literature pointed towards assemblies of α_1 plus α_x : Bencsits et al (doi: 10.1074/jbc.274.28.19613) demonstrated many years ago that α_4 co- assembles with all alpha isoforms except α_6 . Co-assembly of $\alpha_6 + \alpha_1$ is long acknowledged. Many additional papers out there on $\alpha_2 + \alpha_3$, $\alpha_1 + \alpha_5$, $\alpha_2 + \alpha_1$ - Why do you call such long accepted assemblies “non- canonical”? Please acknowledge prior work and abstain from claiming “novelty” for long accepted facts as a sign of appreciation for other researchers.

Reply. *We appreciate this comment and sincerely endeavor to acknowledge previous studies. To this end, we have cited the following papers: Duggen et al. (1991) and Benke et al. (2004) for their research on α_1/α_2 , α_1/α_3 , and α_2/α_3 receptors, Bencsits et al. (1999) concerning α_4/α_x receptors, Araujo et al. (1999) regarding α_1/α_5 receptors, Scholze et al. (2020) relating to α_1/α_6 receptors, and the review by Smart and Stephenson (2019), which touches on the ensembles of receptors in general. We would very much like to cite additional papers but are limited in the number of citations we can make due to journal restrictions.*

Comment. Line 124: “consistent with genetic (22) ... evidence” Reference 22 is far from providing any genetic evidence for the α_1 - β_2 - α_1 - β_2 - γ_2 receptor. In fact, genetics cannot provide evidence for any particular subunit arrangement at all. Please reword accordingly. (I consider this a major point simply because high impact papers, by highly renowned researchers, should be educational beyond the methodical advances, and writing should reflect the limitations of the disciplines properly.)

Reply. We appreciate the reviewer's comment and have removed "genetic" and changed the sentence to read "...consistent with pharmacological...data...".

Comment. Line 92 "α1-GABA A Rs comprise three structural populations" is misleading, as was pointed out previously. Your mass spec data indicates that there must be at least populations with a1+g1, a1+g2, a1+d (!), a1+ ax (x=2, 3, 4, 5, 6) and a1 + any one or two of b1, b2, b3. Due to limitations of the method, you studied the LARGEST populations, which turned out, not surprisingly, gamma2 containing. This can and should be stated clearly. The other populations might be small – but they contribute to brain function and should not be ignored. As suggested in the original review, the mass spec data deserves a place in the MS (at least a simple table), and appropriate discussion.

Reply. We appreciate the reviewer's comment and agree on the presence and the importance of the minor receptor populations supported by the mass spectrometry data. We certainly did not intend to imply that only the three major populations resolved by the cryo-EM matter for brain function.

To be clear that the structural methods are limited to the largest native receptor populations, we changed "three structural populations" to "three major structural populations" at line 97.

The mass spec data are summarized in **Extended Data Figure 1** and provided as **Supplementary Tables 1–3**. All these data are referenced in the main text (line 93). We have also expanded our description and discussion on mass spec data at lines 85–96 and lines 307–320.

lines 85–96:

To validate that the 8E3-GFP Fab captured GABA_AR complexes, before carrying out structural studies, we performed mass spectrometry analysis of the purified native receptor complexes, cognizant of the experimental fact that mass spectrometry is a highly sensitive method that enables the identification of both the most abundant receptor subunits as well as those subunits that comprise only a small fraction of the total population. Indeed, in the subsequent mass spectrometric analysis we identified all α, β, γ subunits as well as the δ subunits (**Extended Data Figure 1, Supplementary Table 1–3**), demonstrating that α1-dependent isolation captured receptors containing most of the 19 GABA_AR subunits, consistent with decades of incisive experimental studies showing that the α1 subunit assembles with all other α subunits, and with the β, γ, δ subunits¹⁸⁻²³. Our mass spectrometry studies also detected peptides unique to the proposed short-splicing isoform of the γ2 subunit²⁴ (γ2-S) (**Supplementary Table 3**).

lines 307–320:

In summary, the cryo-EM analysis reveals three major structural populations of $\alpha 1$ -containing receptors, the single molecule TIRF experiments are consistent with a substantial fraction of receptors harboring a single $\alpha 1$ subunit, yet the mass spectrometry studies show that nearly all of the 19 GABA_AR subunits are present in the $\alpha 1$ -purified preparation, in harmony with the long-standing observation that multiple other subunits assemble with the $\alpha 1$ subunit¹⁸⁻²³. The apparent discrepancy between the cryo-EM and mass spectrometric experiments is simply grounded in their relative sensitivities. While the cryo-EM experiments enable the resolution of the major receptor species and their subunit arrangements, they only allow for the detection of the most highly populated receptor species. By contrast, the mass spectrometry studies allow for the detection of less abundant receptor subunits, but do not enable the determination of which subunits are in specific receptor assemblies. In combination, the two approaches define the major $\alpha 1$ receptor species present in the brain while at the same time supporting the presence of crucial, yet less abundant, $\alpha 1$ -containing receptor complexes^{17,19-22,28}. Future experiments will be required to obtain structural information on these less abundant species.

Comment. Another point here: How high is the confidence with which you can distinguish g2 from g1?

Reply. *Inspection of the density maps at position 115 show clear density for a phenylalanine, thus indicating that the major population of the subunit is $\gamma 2$. In contrast, the $\gamma 1$ subunit has an isoleucine at the equivalent position. Nevertheless, there may be potentially a small amount of $\gamma 1$ subunit (less than 20%) present at the γ positions in the native receptor population, as described in the Supplementary Information.*

We have included the discussion above at line 135–138.

Moreover, these subunit assignments only represent the most abundant subunits based on the cryo-EM maps, with the potential presence of other subunits, such as $\beta 3$ at the β positions or $\gamma 1$ at the γ position, if their relative abundance is ~20% or less.

Comment. Lines 153- 163 : “74% sequence similarity between $\alpha 1$ and $\alpha 3$ ” – up to here, the non- $\alpha 1$ alpha subunit is denoted as $\alpha 2/3$, and now you go back to $\alpha 3$ only in the entire passage? This seems inconsistent. Please keep in mind: Readers from different disciplines need to understand the degree of faith with which entities are assigned to experimental evidence.

Reply. *We appreciate the reviewer’s thoughtful critique. We have therefore modified the main text to refer to the non- $\alpha 1$ α subunit as $\alpha 2/3$ consistently and explained the choice of subunits in the method section.*

The quoted paragraph now reads as follows.

*Despite containing homologous subunits, where there are 80%/75% amino acid sequence identities between $\alpha 1$ and $\alpha 2/3$ subunits, respectively, we observed conformational differences between the two-Fab and one-Fab complexes. The extracellular domains (ECDs) and TMDs are almost identical in $\alpha 1$ and $\alpha 2/3$ subunits in equivalent positions, having backbone RMSDs of 0.45 Å and 0.35 Å, respectively. However, when aligned by the TMD, the RMSD of ECD increases to 1.05 Å (**Supplementary Figure 2**), suggesting significant inter-domain displacement between these $\alpha 1$ and $\alpha 2/3$ subunits. Furthermore, both meta-one-Fab and ortho-one-Fab exhibit markedly shorter separations between the ECD and TMD center of masses than two-Fab complexes, which is also apparent as a reduction of angles between the primary axes of the ECDs and the TMDs in one-Fab receptors (**Extended Data Figure 7**).*

We have added the following sentence in the method section at lines 808–813.

*Although the one-Fab population likely consists of a mixture of $\alpha 2/3$ subunits at the α positions and a mixture of $\beta 1/\beta 2$ subunits at the β positions, we decided to use the $\alpha 3$ subunit and the $\beta 2$ subunit for the modeling and subsequent structural comparison, based on our best interpretations of the density maps. We emphasize, nevertheless, that both $\alpha 2$ and $\beta 1$ models reasonably fit the density maps. Shown in **Extended Data Figure 6** are sequence relationships between the subunits at chosen regions.*

*We have also updated the subunit notation in **Extended Data Figure 7** and **Supplementary Figure 2** accordingly.*

Comment. Line 307 "Our study reveals that $\alpha 1$ -GABA A Rs comprise three structural populations:" is still wrong. This study describes the MAIN populations in detail, and identified ADDITIONAL populations which occur in sufficiently low quantity to prevent downstream structural studies. Once more, these populations are OF MAJOR INTEREST and should not be ignored or suppressed, even if no structure determination was possible. They deserve to be presented. I truly feel it is a waste of money to generate the mass spec data and then to marginalize them in a high impact publication.

Reply. *We agree with the reviewer's emphasis that the 'additional populations' of native receptors are important. We have expanded our discussion about these populations of the receptor at lines 307–320.*

*Additionally, to make the mass spec data more accessible and apparent, we have summarized mass spec data from both methods in **Extended Data Figure 1**, and clearly pointed to the figure and supplementary tables at line 91.*

Lastly, we further emphasize that because of space constraints, we are simply not able to discuss the mass spectrometry data in more detail.

Minor points:

Comment. line 42, including Zn 2+ (ref. 3) – ref 3 should be superscript like all other references

Reply. *In accord with journal convention, the citation was written as such to prevent the confusing letters/symbols "Zn²⁺³".*

REFEREE #4

Major comments:

Comment. Question to reply to major comment 1:

It is great that the natural occurrence of both splice isoforms is addressed now in the manuscript. However, this reviewer does not fully understand the answer given by the group:

"...forms of the $\gamma 2$ subunits (short and long), only differ in the M3-M4 loop by 8 amino acids, which is underlined in the $\gamma 2$ -L sequence "KKKKNPLLRMFSFKAPTIDIRPRSATIQMNNATHLQER". This relatively small difference makes it challenging to differentiate them, even by with mass spectrometry. Fortunately, in the mass spec data, there were two detected peptides "APTIDIRPR" and "APTIDIRPRSATIQMNNATHLQER" that can only be derived from the long isoform, unambiguously supporting its presence in our native receptor preparation."

Q: It is correct that both isoforms differ by the given 8 amino acids. However, the two given peptide fragments could be derived from both isoforms sequence-wise, since they start right after the insertion that is specific to the g2L version and thus are included by both versions. How can the native existence of the long isoform be proven with those fragments? If something was misunderstood, please explain.

Reply. *We thank the reviewer for this question. Because we used trypsin to digest the protein, we inferred that the only way that the "APT..." peptides could arise is if the long isoform were present, where the aforementioned peptide would be generated by trypsin cleavage between the 'K' and 'A', "...SFKAPT...", where the 'K' is present in the long isoform.*

The reviewer is completely correct that we cannot prove the presence of the long isoform directly from the observed peptides. Thus, we have rephrased our conclusion to state that the mass spec data is consistent with the presence of the short isoform. Further studies will need to be carried out to determine the presence of the long isoform.

*We have changed the conclusion regarding the isoform of $\gamma 2$ subunit at line 94–96. Our mass spectrometry studies also detected peptides unique to the proposed short splicing isoform of the $\gamma 2$ subunit²⁵ ($\gamma 2$ -S) (**Supplementary Table 3**).*

We have also updated the Supplementary Tables accordingly.

Comment. Q to reply to major comment 4.1:

“the reason we used 6I53 for the structure comparison is because it is the only full-length triheteromeric GABAAR structure that is in complex with GABA, but not other drug ligands, making it a good reference to evaluate the effect of the neurosteroid. Despite the lack of an explicit GABA label in the PDB deposition, the authors themselves confirmed the GABA-bound nature of this structure in the paper, “Although our structure was solved in the absence of exogenous GABA, we observe serendipitous non-protein densities at the agonist-binding sites of both $\beta +/\alpha -$ interfaces (Extended Data Fig. 9b,c).” Consistent with it being a GABA-bound complex, the 6I53 structure has a complete loop C closure.”

REMARK: The authors that published 6I53 further state “Although we cannot reliably determine the identity of the contaminating ligand, we carried out in silico molecular docking of GABA within these sites”. Thus, GABA could not be reliably identified in a probe that was prepared without GABA, but with high mM concentrations of contaminants. Therefore, it is great that the authors compared their structures to the proposed 6HUO and included their findings into the reply letter, however, the comparison between 6I53, 6HUO, 6X3X and novel structures should be accessible in form of a figure and be discussed. This could have been included or been in addition to the new supplementary figure 8.

Reply. *To enhance the accessibility of the structural comparison mentioned in the previous response, we have prepared an additional supplementary figure (**Supplementary Figure 3**) to illustrate the structural features of the 6I53 GABA binding site and we refer to this addition to the manuscript at line 181 of the main text.*

*As suggested by the reviewer, we have revised the new **Extended Data Figure 8** to include the structural comparison between 6I53, 6HUO, 6X3X and our neurosteroid bound structure. Panel a highlights the comparison of the extracellular domains and the GABA binding pockets, showing that the binding pockets are highly similar, panel b focuses on the transmembrane domains, and panel c compares the neurosteroid pockets. The figure*

legend provides a brief description and discussion of these structural comparisons. We would ideally like to include further discussion in the main text but are limited by space constraints.

Comment. As suggested by the mass-spec data in this study, $\alpha 1$ - subunit-containing GABAA receptors form hetero-pentamers with a wide variety of GABAA receptor subunits – far more than those that were identified in the resolved cryo-EM structures by the group. Those include $\alpha 1$ -containing receptors with ALL other alpha isoforms, ALL beta isoforms, ALL gamma isoforms and the delta subunit. As stated in the reply letter by the group, receptors with low abundance (less than 5%) cannot be resolved by cryo-EM experiments, despite their undoubtable existence! The existence of these smaller populations must be fully addressed in the manuscript before publication.

It is long known, that $\alpha 1\beta 2\gamma 2$ -containing receptors represent the major GABAAR population in the brain. It is also suggested for more than 20 years that $\alpha 1$ -containing receptors can have multiple α - or β - isoforms incorporated -as is now addressed by citation in the paper. However, therefore it is also common knowledge that the other GABAAR isoforms present a very small fraction in the brain. Since there is so many of them theoretically, it is a given that the majority of them can NOT be resolved in current cryo-EM experiments. That does not exclude their existence! The mass-spec data of this study in contrast to the claims made in the manuscript even prove their existence and as derived from the reply letters - the only reason they could not be resolved is due to inherent limitations of the cryo-EM experiments. Severe disease-causing variants in ALL the subunits identified in the mass-spec data essentially highlight their physiological importance and thus inclusion into functional pentamers further.

It is unacceptable to ignore or marginalize their existence and emphasize the existence of the proposed $\alpha 1$ -containing receptors exclusively, because no other population could be resolved with the main experimental technique.

Reply. *We agree that mass spectrometry data support the presence of other receptor populations than the three identified by the cryo-EM studies. To address these minor populations and to resolve the apparent discrepancy between the cryo-EM data and mass spectrometry data, we have included the following discussion at lines 307–320.*

In summary, the cryo-EM analysis reveals three major structural populations of $\alpha 1$ -containing receptors, the single molecule TIRF experiments are consistent with a substantial fraction of receptors harboring a single $\alpha 1$ subunit, yet the mass spectrometry studies show that nearly all of the 19 GABA_AR subunits are present in the $\alpha 1$ -purified preparation, in harmony with the long-standing observation that multiple other subunits assemble with the $\alpha 1$ subunit¹⁸⁻²³. The apparent discrepancy between the cryo-EM and mass spectrometric

experiments is simply grounded in their relative sensitivities. While the cryo-EM experiments enable the resolution of the major receptor species and their subunit arrangements, they only allow for the detection of the most highly populated receptor species. By contrast, the mass spectrometry studies allow for the detection of less abundant receptor subunits, but do not enable the determination of which subunits are in specific receptor assemblies. In combination, the two approaches define the major $\alpha 1$ receptor species present in the brain while at the same time supporting the presence of crucial, yet less abundant, $\alpha 1$ -containing receptor complexes^{17,19-22,28}. Future experiments will be required to obtain structural information on these less abundant species.

Comment. Line 31-33: This statement in the abstract is strongly misleading considering the mass-spec data.

The group resolved major populations of $\alpha 1$ -containing receptors, whereas other existing assemblies with less abundance could not be resolved in the cryo-EM experiments owe to technical limitations of the used method. In line with this, GABAA receptor assembly does NOT “yield a small number of structurally distinct complexes” but the rest of complexes could simply not be resolved, despite their existence - as shown by the more sensitive mass-spec data. Thus, any assumption on assembly regulation, stoichiometry and arrangement of less prevalent receptors can simply not be made and proper writing needs to reflect this fact.

Reply. *We agree with the reviewer and have reworked the sentence to read “Together, our data reveal the major $\alpha 1$ -containing GABA_AR assemblies, bound with endogenous neurosteroid, thus defining a structural landscape from which subtype-specific drugs can be developed.”*

Comment. Line 113-117:

Again, this is simply wrong. In light of the mass spec data, $\alpha 1$ -containing receptors do form receptors with the d-subunit and with b-subunits that could both not be resolved here. Heterologous expression might just enable the detection of those receptor population due to overexpression compared to neurons, thus enrichment of those populations. Any conclusion on heterologous expression systems is highly speculative based on the mass-spec data. Possible differences in results, however, should again be discussed!

Reply. *We agree with the reviewer that we should refrain from comparing our findings with native receptors to heterologous expression systems. We have deleted lines 113–117. We also removed the comparison to heterologous expression system from the conclusion section.*

Comment. Line 307-309

This is both, wrong and strongly misleading. This study does not reveal that murine $\alpha 1$ -containing receptors do form three structural populations as stated, but it only resolved the three most abundant assemblies of $\alpha 1$ -containing receptors could be resolved, with no information about the rest. Yes, identifying three major native receptor assemblies in cryo-EM experiments speaks for regulation of $\alpha 1\beta 2\gamma 2$ receptor assembly in neurons, but nothing more than that.

Thus, the otherwise outstanding work will benefit greatly from appropriate framing and discussion of all $\alpha 1$ - assembly partners that were identified by all utilized methods, and this reviewer looks forward to the improved version.

Reply. *We appreciate this comment and have entirely reworked the paragraph to read as follows:*

"In summary, the cryo-EM analysis reveals three major structural populations of $\alpha 1$ -containing receptors, the single molecule TIRF experiments are consistent with a substantial fraction of receptors harboring a single $\alpha 1$ subunit, yet the mass spectrometry studies show that nearly all of the 19 GABA_AR subunits are present in the $\alpha 1$ -purified preparation, in harmony with the long-standing observation that multiple other subunits assemble with the $\alpha 1$ subunit¹⁸⁻²³. The apparent discrepancy between the cryo-EM and mass spectrometric experiments is simply grounded in their relative sensitivities. While the cryo-EM experiments enable the resolution of the major receptor species and their subunit arrangements, they only allow for the detection of the most highly populated receptor species. By contrast, the mass spectrometry studies allow for the detection of less abundant receptor subunits, but do not enable the determination of which subunits are in specific receptor assemblies. In combination, the two approaches define the major $\alpha 1$ receptor species present in the brain while at the same time supporting the presence of crucial, yet less abundant, $\alpha 1$ -containing receptor complexes^{17,19-22,28}. Future experiments will be required to obtain structural information on these less abundant species."

Reviewer Reports on the Second Revision:

Referees' comments:

Referee #3 (Remarks to the Author):

In the re- revision, the authors have dealt with all concerns. I only weakly suggest one tiny change: lines 317-318:

Instead of "while at the same time supporting the presence of crucial, yet less abundant, α 1-containing receptor complexes"

it could be stated "while at the same time supporting the presence of MULTIPLE, yet less abundant, CRUCIAL α 1-containing receptor complexes".

But I am also fine with as it stands, the thoughtful reply and the changes to the text are received very well,

with collegiate best wishes from Vienna

margot ernst

Referee #4 (Remarks to the Author):

The reviewer expresses a strong appreciation for the modifications made to the manuscript and, therefore, recommends the publication of the current version of the manuscript.

Best regards,
Florian Daniel Vogel

Author Rebuttals to Second Revision:

REFEREE #3

Minor point: In the re- revision, the authors have dealt with all concerns. I only weakly suggest one tiny change: lines 317-318:

Instead of "while at the same time supporting the presence of crucial, yet less abundant, α 1-containing receptor complexes" it could be stated "while at the same time supporting the presence of MULTIPLE, yet less abundant, CRUCIAL α 1-containing receptor complexes".

Comment. We have revised the manuscript in accord with the reviewer's comment.